# How Over-Parameterization Slows Down Gradient Descent in Matrix Sensing: The Curses of Symmetry and Initialization

**Nuoya Xiong** [*]
nuoyaxiong@gmail.com

**Lijun Ding** [†]
lding@tamu.edu

**Simon S. Du** [‡]
ssdu@cs.washington.edu

## Abstract

This paper rigorously shows how over-parameterization dramatically changes the convergence behaviors of gradient descent (GD) for the matrix sensing problem, where the goal is to recover an unknown low-rank ground-truth matrix from near-isotropic linear measurements. First, we consider the symmetric setting with the symmetric parameterization where $M^* \in \mathbb{R}^{n \times n}$ is a positive semi-definite unknown matrix of rank $r \ll n$, and one uses a symmetric parameterization $XX^\top$ to learn $M^*$. Here, $X \in \mathbb{R}^{n \times k}$ with $k > r$ is the factor matrix. We give a novel $\Omega\left(1/T^2\right)$ *lower bound* of randomly initialized GD for the over-parameterized case $(k > r)$ where $T$ is the number of iterations. This is in stark contrast to the exact-parameterization scenario $(k = r)$ where the convergence rate is $\exp\left(-\Omega\left(T\right)\right)$. Next, we study asymmetric setting where $M^* \in \mathbb{R}^{n_1 \times n_2}$ is the unknown matrix of rank $r \ll \min\{n_1, n_2\}$, and one uses an asymmetric parameterization $FG^\top$ to learn $M^*$ where $F \in \mathbb{R}^{n_1 \times k}$ and $G \in \mathbb{R}^{n_2 \times k}$. Building on prior work, we give a global exact convergence result of randomly initialized GD for the exact-parameterization case $(k = r)$ with an $\exp\left(-\Omega\left(T\right)\right)$ rate. Furthermore, we give the first global exact convergence result for the over-parameterization case $(k > r)$ with an $\exp\left(-\Omega\left(\alpha^2 T\right)\right)$ rate where $\alpha$ is the initialization scale. This linear convergence result in the over-parameterization case is especially significant because one can apply the asymmetric parameterization to the symmetric setting to speed up from $\Omega\left(1/T^2\right)$ to linear convergence. Therefore, we identify a surprising phenomenon: *asymmetric parameterization can exponentially speed up convergence.* Equally surprising is our analysis that highlights the importance of *imbalance* between $F$ and $G$. This is in sharp contrast to prior works which emphasize balance. We further give an example showing the dependency on $\alpha$ in the convergence rate is unavoidable in the worst case. On the other hand, we propose a novel method that only modifies one step of GD and obtains a convergence rate independent of $\alpha$, recovering the rate in the exact-parameterization case. We provide empirical studies to verify our theoretical findings.

## 1 Introduction

A line of recent work showed over-parameterization plays a key role in optimization, especially for neural networks (Allen-Zhu et al., 2019; Du et al., 2018b; Jacot et al., 2018; Safran & Shamir, 2018; Chizat et al., 2019; Wei et al., 2019; Nguyen & Pham, 2020; Fang et al., 2021; Lu et al., 2020; Zou et al., 2020). However, our understanding of the impact of over-parameterization on optimization is far from complete. In this paper, we focus on the canonical matrix sensing problem and show that over-parameterization qualitatively changes the convergence behaviors of gradient descent (GD).

Matrix sensing aims to recover a low-rank unknown matrix $M^\star$ from $m$ linear measurements,

$$y_i = \mathcal{A}_i(M^\star) = \langle A_i, M^\star \rangle = \text{tr}(A_i^\top M^\star), \text{ for } i = 1, \ldots, m, \tag{1.1}$$

[*]IIIS, Tsinghua University. Part of the work was done while Nuoya Xiong was visiting the University of Washington

[†]Wm Michael Barnes '64 Department of Industrial & Systems Engineering, Texas A&M University.

[‡]Paul G. Allen School of Computer Science and Engineering, University of Washington

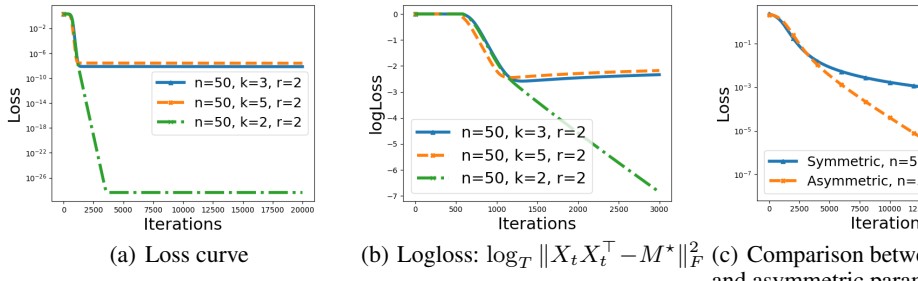

(a) Loss curve  (b) Logloss: $\log_T \|X_t X_t^\top - M^\star\|_F^2$  (c) Comparison between symmetric and asymmetric parameterization

Figure 1.1: Experiments on symmetric setting. The first two figures show that the convergence rate of symmetric matrix factorization in the over-parameterized setting is about $\Theta(1/T^2)$, while the rate of the exact-parameterized setting is linear. 1(c) shows that using asymmetric parameterization is exponentially faster than symmetric parameterization. See §H for experimental details.

where $\mathcal{A}_i$ is a linear measurement operator and $A_i$ is the measurement matrix of the same size as $M^\star$. This is a classical problem with numerous real-world applications, including signal processing (Weng & Wang, 2012) and face recognition (Chen et al., 2012), image reconstruction (Zhao et al., 2010; Peng et al., 2014). Moreover, this problem can serve as a test-bed of convergence behaviors in deep learning theory since it is non-convex and retains many key phenomena (Soltanolkotabi et al., 2023; Jin et al., 2023; Li et al., 2018; 2020; Arora et al., 2019). We primarily focus on the *over-parameterized* case where we use a model with rank larger than that of $M^\star$ in the learning process. This case is particularly relevant because rank($M^\star$) is usually unknown in practice.

## 1.1 SETTING 1: SYMMETRIC MATRIX SENSING WITH SYMMETRIC PARAMETERIZATION

We first consider the symmetric matrix sensing setting, where $M^\star \in \mathbb{R}^{n \times n}$ is a positive semi-definite matrix of rank $r \ll n$. A standard approach is to use a factored form $XX^\top$ to learn $M^\star$ where $X \in \mathbb{R}^{n \times k}$. We call this *symmetric parameterization* because $XX^\top$ is always symmetric and positive semi-definite. We will also introduce the *asymmetric* parameterization soon. We call the case when $k = r$ the *exact-parameterization* because the rank of $XX^\top$ matches that of $M^\star$. However, in practice, $r$ is often unknown, so one may choose some large enough $k > r$ to ensure the expressiveness of $XX^\top$, and we call this case *over-parameterization*.

We consider using gradient descent to minimize the standard $L_2$ loss for training: $L_{\text{tr}}(X) = \frac{1}{2m} \sum_{i=1}^m \left(y_i - \langle A_i, XX^\top \rangle\right)^2$. We use the Frobneius norm of the reconstruction error as the performance metric:

$$L(X) = \frac{1}{2}\|XX^\top - M^\star\|_F^2. \tag{1.2}$$

We note that $L(X)$ is also the *matrix factorization* loss and can be viewed as a special case of $L_{\text{tr}}$ when $\{A_i\}_{i=1}^m$ are random Gaussian matrices and the number of linear measurements go to infinity.

For the exact-parameterization case, one can combine the results by Stöger & Soltanolkotabi (2021) and Tu et al. (2016) to show that randomly initialized gradient descent enjoys an $\exp(-\Omega(T))$ convergence rate where $T$ is the number of iterations. For the over-parameterization case, one can combine the results by Stöger & Soltanolkotabi (2021) and Zhuo et al. (2021) to show an $O(1/T^2)$ convergence rate *upper bound*, which is exponentially worse. This behavior has been empirically observed (Zhang et al., 2021b; 2023; Zhuo et al., 2021) without a rigorous proof. See Figure 1.

**Contribution 1:** $\Omega(1/T^2)$ **Lower Bound for Symmetric Over-Parameterization**. Our first contribution is a rigorous exponential separation between the exact-parameterization and over-parameterization by proving an $\Omega(1/T^2)$ convergence rate *lower bound* for the symmetric setting with the symmetric over-parameterization.

**Theorem 1.1** (Informal). *Suppose we initialize $X$ with a Gaussian distribution with small enough variance that scales with $\alpha^2$, and use gradient descent with a small enough constant step size to optimize the matrix factorization loss (1.2). Let $X_t$ denote the factor matrix at the $t$-th iteration.*

*Then with high probability over the initialization, there exists $T^{(0)} > 0$ such that we have[1][2]*

$$\|X_t X_t^\top - M^\star\|_F^2 \geq \left(\frac{\alpha^2}{t}\right)^2, \forall t \geq T^{(0)}. \tag{1.3}$$

**Technical Insight:** We find the root cause of the slow convergence is from the *redundant space* in $XX^\top$, which converges to $0$ at much slower rate compared to the *signal space* which converges to $M^\star$ with a linear rate. To derive the lower bound, we construct a potential function and use some novel analyses of the updating rule to show that the potential function decreases slowly after a few rounds. See the precise theorem and more technical discussions in Section 3.

## 1.2 SETTING 2: SYMMETRIC AND ASYMMETRIC MATRIX SENSING WITH ASYMMETRIC PARAMETERIZATION

Next, we consider the more general asymmetric matrix sensing problem where the ground-truth $M^\star \in \mathbb{R}^{n_1 \times n_2}$ is asymmetric matrix of rank $r$. For this setting, we must use the *asymmetric parameterization*. Specifically, we use $FG^\top$ to learn $M^\star$ where $F \in \mathbb{R}^{n_1 \times k}$ and $G \in \mathbb{R}^{n_2 \times k}$. Same as in the symmetric case, exact-parameterization means $k = r$ and over-parameterization means $k > r$. We still use gradient descent to optimize the $L_2$ loss for training:

$$L_{\mathrm{tr}}(F, G) = \frac{1}{2m} \sum_{i=1}^{m} \left(y_i - \langle A_i, FG^\top \rangle\right)^2, \tag{1.4}$$

and the performance metric is still: $L(F, G) = \frac{1}{2}\|FG^\top - M^\star\|_F^2$. To enable the analysis, we assume throughout the paper that the matrices $\{A_i\}_{i=1}^m$ satisfies the Restricted Isometry Property (RIP) of order $2k + 1$ with parameter $\delta \leq \tilde{\mathcal{O}}(\frac{1}{\sqrt{kr}})$. (See Definition 2.1 for the detailed definition).

Also note that even for the *symmetric matrix sensing problem* where $M^\star$ is positive semi-definite, one can still use *asymmetric parameterization*. Although doing so seems unnecessary at the first glance, we will soon see using asymmetric parameterization enjoys an *exponential gain*.

**Contribution 2: Global Exact Convergence of Gradient Descent for Asymmetric Exact-Parameterization with a Linear Convergence Rate**. Our second major contribution is a global exact convergence result for randomly initilized gradient descent, and we show it enjoys a linear convergence rate.[3]

**Theorem 1.2** (Informal). *In the exact-parameterization setting ($k = r$), suppose we initialize $F$ and $G$ using a Gaussian distribution with small enough variance $\alpha^2$ and use gradient descent with a small enough constant step size to optimize the asymmetric matrix sensing loss (1.4). Let $F_t$ and $G_t$ denote the factor matrices at the $t$-the iteration. Then with high probability over the random initialization, there exists $T^{(1)} > 0$ such that we have*

$$\|F_t G_t^\top - M^\star\|_F^2 = \exp\left(-\Omega\left(t\right)\right), \forall t \geq T^{(1)}. \tag{1.5}$$

Compared to our results, prior results either require initialization to be close to optimal (Ma et al., 2021), or can only guarantee to find a point with an error of similar scale as the initialization (Soltanolkotabi et al., 2023). In contrast, our result only relies on random initialization and guarantees the error goes to $0$ as $t$ goes to infinity. Notably, this convergence rate is independent of $\alpha$. See Figure 2(a). Naturally, such a result is expected and can be derived (with considerable effort) by the works (Ma et al., 2021, Theorem 1) and (Soltanolkotabi et al., 2023). Our proof strategy is very different from (Ma et al., 2021) as we further decompose the factors $F$ and $G$, and we only need (Soltanolkotabi et al., 2023, Theroem 3.3) to deal with the initial phase.

**Contribution 3: Global Exact Convergence of Gradient Descent for Asymetric Over-Parameterization with an Initialization-Dependent Linear Convergence Rate.** Our next contribution is analogue theorem for the over-parameterization case with caveat that the initialization scale $\alpha$ also appears in the convergence rate.

---

[1]For clarity, in our informal theorems in Section 1, we only display the dependency on $\alpha$ and $T$, and ignore parameters such as dimension, condition number, and step size.

[2]$T^{(0)}$ here and $T^{(1)}$, $T^{(2)}$, $T^{(3)}$ in theorems below represent the burn-in time to get to a neighborhood of an optimum, which can depend on initilization scale $\alpha$, condition number, dimension, and step size.

[3]By exact convergence we mean the error goes to $0$ as $t$ goes to infinity in contrast to prior works which only gurantee to converge to a point with the error proportional to the initialization scale $\alpha$.

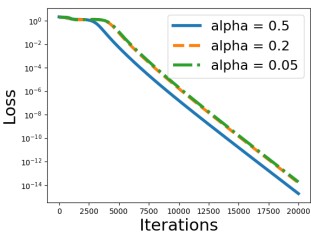 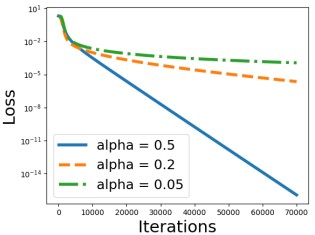 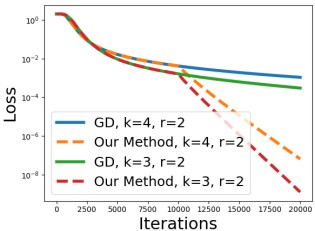

| (a) Exact-parameterized case | (b) Over-parameterized case | (c) Loss curve for our new method |

Figure 1.2: Curve of asymmetric matrix sensing. Figure 2(a) shows that the convergence rate is linear and independent on the initialization scale under the exact-parameterized case. Figure 2(b) shows that the convergence rate is linear and dependent on the initialization scale under the over-parameterized case. When the initialization scale is larger, the convergence speed is faster. Figure 2(c) shows the efficacy of our new method. See §H for experimental details.

**Theorem 1.3** (Informal). *In the over-parameterization setting ($k > r$), suppose we initialize $F$ and $G$ using a Gaussian distribution with small enough variance $\alpha^2$ and use gradient descent with a small enough constant step size to optimize the asymmetric matrix sensing loss (1.4). Let $F_t$ and $G_t$ denote the factor matrices at the $t$-th iteration. Then with high probability over the random initialization there exists $T^{(2)} > 0$ such that we have*

$$\|F_t G_t^\top - M^\star\|_F^2 = \exp\left(-\Omega\left(\alpha^2 t\right)\right), \forall t \geq T^{(2)}. \tag{1.6}$$

This is also the first global exact convergence result of randomly initialized gradient descent in the over-parameterized case. Recall that for the symmetric matrix sensing problem, even if $M^\star$ is positive semi-definite, one can still use an asymmetric parameterization $FG^\top$ to learn $M^\star$, and Theorem 1.3 still holds. Comparing Theorem 1.3 and Theorem 1.1, we obtain a surprising corollary:

*For the **symmetric** matrix sensing problem, using **asymmetric** parameterization is **exponentially faster** than using symmetric parameterization.*

Also notice that different from Theorem 1.2, the convergence rate of Theorem 1.3 also depends on the initialization scale $\alpha$ which we require to be small. Empirically we verify this dependency is necessary. See Figure 2(b). We also study a special case in Section 4.1 to show the dependency on the initialization scale is necessary in the worst case.

**Technical Insight:** Our key technical finding that gives the exponential acceleration is the *imbalance* of $F$ and $G$. Denote the imbalance matrix $\Delta_t = F_t^\top F_t - G_t^\top G_t$. We show that the converge rate is linear when $\Delta_t$ is positive definite, and the rate depends on the minimum eigenvalue of $\Delta_t$. We use imbalance initialization so that the minimum eigenvalue of $\Delta_0$ is proportional to $\alpha$, we can further show that the minimum eigenvalue $\Delta_t$ will not decrease too much, so the final convergence rate is linear. Furthermore, such a connection to $\alpha$ also inspires us to design a faster algorithm below.

**Contribution 4: A Simple Algorithm with Initialization-Independent Linear Convergence Rate for Asymetric Over-Parameterization.** Our key idea is to *increase the degree of imbalance* when $F$ and $G$ are close to the optimum. We develop a new simple algorithm to accelerate GD. The algorithm only involves transforming the factor matrices $F$ and $G$ in one of iteration to intensify the degree of imbalance (cf. Equation (5.1)).

**Theorem 1.4** (Informal). *In the over-parameterization setting ($k > r$), suppose we initialize $F$ and $G$ using a Gaussian distribution with small enough variance $\alpha^2$, gradient descent with a small enough constant step size, and the procedure described in Section 5 to optimize the loss (1.4). Let $F_t$ and $G_t$ denote the factor matrices at the $t$-the iteration. Then with high probability over the random initialization, there exists $T^{(3)} > 0$ such that we have*

$$\|F_t G_t^\top - M^\star\|_F^2 = \exp\left(-\Omega\left(t - T^{(3)}\right)\right), \forall t \geq T^{(3)}. \tag{1.7}$$

### 1.3 RELATED WORK

Matrix sensing is a canonical problem and has been widely studied via the nuclear norm minimization approach (Candes & Recht, 2012; Liu et al., 2012; Recht et al., 2010; Wu & Rebeschini, 2021),

Table 1: Comparison of previous representative work. The second column shows that the results hold for symmetric matrix factorization/sensing or asymmetric matrix factorization/sensing. The third column lists different types of initialization, where "Random" means the algorithm uses random initialization (typically Gaussian), "Local" indicates a requirement for initialization to be close to the optimal point. The fourth column "exact-cnvrg" represents whether the loss will go to zero when round $T$ goes to infinity. The fifth column indicates whether the result applies to over-parameterization case or just the exact-parameterization case. The last row lists the convergence rate of algorithms with exact-convergence results.

| | Is Symmetric | Init. | exact-cnvrg | $k$ Range | Rate |
|---|---|---|---|---|---|
| Stöger & Soltanolkotabi (2021) | Symmetric | Random | ✗ | $k \geq r$ | N/A |
| Zhuo et al. (2021) | Symmetric | Local | ✓ | $k \geq r$ | $\mathcal{O}(1/T^2)$ |
| Stöger & Soltanolkotabi (2021) + Zhuo et al. (2021) | Symmetric | Random | ✓ | $k \geq r$ | $\mathcal{O}(1/T^2)$ |
| Soltanolkotabi et al. (2023) | Asymmetric | Random | ✗ | $k \geq r$ | N/A |
| Tu et al. (2016) | Both | Local | ✓ | $k = r$ | $\exp(-\Omega(T))$ |
| Ma et al. (2021) | Asymmetric | Local | ✓ | $k = r$ | $\exp(-\Omega(T))$ |
| Theorem 4.3 (our paper) | Asymmetric | Random | ✓ | $k = r$ | $\exp(-\Omega(T))$ |
| Theorem 4.2 (our paper) | Asymmetric | Random | ✓ | $k > r$ | $\exp(-\Omega(\alpha^2 T))$ |
| Theorem 3.1 (our paper) | Symmetric | Random | ✓ | $k \geq r$ | $\Omega(1/T^2)$ |

spectral method (Ma et al., 2021; Tu et al., 2016) and landscape analysis (Ge et al., 2017; Zhu et al., 2018) The most relevant line of work considered global convergence of gradient descent (Zhuo et al., 2021; Stöger & Soltanolkotabi, 2021; Soltanolkotabi et al., 2023; Tu et al., 2016). We compare our results with the them in Table 1. The detailed discussions on related work is deferred to Appendix A.

## 2 PRELIMINARIES

**Norm and Big-$\mathcal{O}$ Notations.** Given a vector $v$, we use $\|v\|$ to denote its Euclidean norm. For a matrix $M$, we use $\|M\|$ to denote its spectral norm and $\|M\|_F$ Frobenius norm. The notations $\mathcal{O}(\cdot), \Theta(\cdot)$, and $\Omega(\cdot)$ in the rest of the paper only omit absolute constants.

**Asymmetric Matrix Sensing.** Our primary goal is to recover an unknown fixed rank $r$ matrix $M^\star \in \mathbb{R}^{n_1 \times n_2}$ from data $(y_i, A_i)$, $i = 1, \ldots, m$ satisfying $y_i = \langle A_i, M^\star \rangle = \text{tr}(A_i^\top M^\star), i = 1, \ldots, m,$ or compactly $y = \mathcal{A}(M^\star)$, where $y \in \mathbb{R}^m$ and $\mathcal{A} : \mathbb{R}^{n_1 \times n_2} \to \mathbb{R}^m$ is a linear map with $[\mathcal{A}(M)]_i = \text{tr}(A_i^\top M)$. We further denote the singular values of $M^\star$ as $\sigma_1 \geq \cdots \geq \sigma_r > \sigma_{r+1} = 0 = \cdots = \sigma_n$, the condition number $\kappa = \frac{\sigma_1}{\sigma_r}$, and the diagonal singular value matrix as $\Sigma$ with $(\Sigma)_{ii} = \sigma_i$. To recover $M^\star$, we minimize the following loss function:

$$L_{\text{tr}}(F, G) = \frac{1}{2} \|\mathcal{A}(FG^\top) - y\|^2, \tag{2.1}$$

where $F, G \in \mathbb{R}^{n \times k}$, where $k \geq r$ is the user-specified rank. The gradient descent update rule with a step size $\eta > 0$ with respect to loss (2.1) can be written explicitly as

$$F_{t+1} = F_t - \eta \mathcal{A}^* \mathcal{A}(F_t G_t^\top - \Sigma) G_t, \quad G_{t+1} = G_t - \eta (\mathcal{A}^* \mathcal{A}(F_t G_t^\top - \Sigma))^\top F_t, \tag{2.2}$$

where $\mathcal{A}^* : \mathbb{R}^m \to \mathbb{R}^{n \times n}$ is the adjoint map of $\mathcal{A}$ and admits an explicit form: $\mathcal{A}^*(z) = \sum_{i=1}^m z_i A_i$.

We make the following assumption on $\mathcal{A}$: Restricted Isometry Property (RIP) (Recht et al., 2010).

**Definition 2.1.** *[Restricted Isometry Property] An operator $\mathcal{A} : \mathbb{R}^{n_1 \times n_2} \to \mathbb{R}^m$ satisfies the Restricted Isometry Property of order $r$ with constant $\delta > 0$ if for all matrices $M : \mathbb{R}^{n_1 \times n_2}$ with rank at most $r$, we have $(1 - \delta)\|M\|_F^2 \leq \|\mathcal{A}(M)\|^2 \leq (1 + \delta)\|M\|_F^2$.*

From (Candes & Plan, 2011), if the matrix $A_i$ has i.i.d. $N(0, 1/m)$, the operator $\mathcal{A}$ has RIP of order $2k + 1$ with constant $\delta \in (0, 1)$ when $m = \widetilde{\Omega}(nk/\delta^2)$. Thus, $m = \widetilde{\Omega}(nk^2 r)$ is needed with (4.8).

**Diagonal Matrix Simplification.** Since both the RIP and the loss are invariant to orthogonal transformation, we assume without loss generality that $M^\star = \Sigma$ in the rest of the paper for clarity,

following prior work (Ye & Du, 2021; Jiang et al., 2022). For the same reason, we also assume $n_1 = n_2 = n$ to simplify notations, and our results can be easily extended to $n_1 \neq n_2$.

**Symmetric Matrix Factorization.** In this setting, we further assume $M^\star$ is symmetric and positive semidefinite, and $\mathcal{A}$ is the identity map. Since $M^\star$ admits a factorization $M^\star = F_\star F_\star^\top$ for some $F_\star \in \mathbb{R}^{n \times r}$, we can force the factor $F = G = X$ in (2.1) and the loss becomes $L(X) = \frac{1}{2}\|XX^\top - \Sigma\|_F^2$. Here, the factor $X \in \mathbb{R}^{n \times k}$. We shall focus on the over-parameterization setting, i.e., $k > r$ in the Setion 3 below. The gradient descent with a step size $\eta > 0$ becomes

$$X_{t+1} = X_t - 2\eta(X_t X_t^\top - \Sigma)X_t. \tag{2.3}$$

## 3 LOWER BOUND OF SYMMETRIC MATRIX FACTORIZATION

We present a sublinear lower bound of the convergence rate of the gradient descent (2.3) for symmetric matrix factorization with a small random initialization. Our result supports the empirical observations of the slowdown in Zhuo et al. (2021); Zhang et al. (2021b; 2023) and Figure 1.

**Theorem 3.1.** *Let $X_0 = \alpha \cdot \tilde{X}_0$, where each entry of $\tilde{X}_0$ is independent initialized from Gaussian distribution $\mathcal{N}(0, 1/k)$. For some universal constants $c_i, 1 \leq i \leq 7$, if the gradient descent method (2.3) starting at $X_0$ with the initial scale $\alpha$, the search rank $k$, and the stepsize $\eta$ satisfying that*

$$0 < \alpha \leq \frac{c_1 \sqrt{\sigma_1}}{\sqrt{n}\log(r\sqrt{n})}, \quad k \geq c_2 \left((r\kappa)^2 \log(r\sqrt{\sigma_1}/\alpha)\right)^4, \quad and \quad 0 < \eta \leq \frac{c_3}{n^2 \kappa \sigma_1}, \tag{3.1}$$

*then with probability at least $1 - 2n^2 \exp(-\sqrt{c_4 k}) - 2n \exp(-c_5 k/4)$, for all $t \geq T^{(0)} = \frac{c_6 \log(r\sqrt{\sigma_1})/\alpha}{\eta \sigma_r}$, we have*

$$\|X_t X_t^\top - \Sigma\|_F^2 \geq \left(\frac{c_7 \log(\sqrt{r\sigma_1}/\alpha)\alpha^2}{8\sigma_r \eta \sqrt{n} t}\right)^2, \quad \forall t \geq T^{(0)}. \tag{3.2}$$

The proof of Theorem 1.1 demonstrates that, following a rapid convergence phase, the gradient descent eventually transitions to a *sublinear* convergence rate. Also, the over-parameterization rank $k$ is subject to a lower bound requirement in Eq. (3.1) that depends on $\alpha$. However, since $\alpha$ only appears in the logarithmic term, this requirement is not overly restrictive. It is also consistent with the phenomenon that the gradient descent first converges to a small error that depends on $\alpha$ with a linear convergence rate (Stöger & Soltanolkotabi, 2021), since our lower bound has a term $\alpha^4$.

### 3.1 PROOF SKETCH OF THEOREM 3.1

We provide a proof sketch of Theorem 3.1 in this section, deferring the details to Appendix B. The main intuition of Theorem 3.1 is that the last $n - r$ rows of $X_t$, corresponding to the space of $0$ eigenvalues of $\Sigma$, converge to $0$ at speed no faster than $\frac{1}{T^2}$. To make this intuition precise, for each $t \geq 0$, we let $X_t \in \mathbb{R}^{n \times k} = [x_1^t, \cdots, x_n^t]^\top$ where $x_i^t \in \mathbb{R}^k$. We let the potential function be $A_t = \sum_{i>r} \|x_i^t\|^2$. We aim to show the following two key inequalities,

$$\|x_i^{T^{(0)}}\|^2 \geq \alpha^2/8, \text{ for all } i > r, \tag{3.3a}$$

$$A_{t+1} \geq A_t(1 - \mathcal{O}(\eta A_t)), \text{ for all } t \geq T^{(0)}. \tag{3.3b}$$

Suppose (3.3) is true, then it implies that $A_t \geq \mathcal{O}\left(\frac{\alpha^2}{t}\right)$ for all $t \geq T^{(0)}$. Since $(X_t X_t^\top - \Sigma)_{ii} = \|x_i\|^2$, the lower bound (3.2) is established by noting that $\|X_t X_t^\top - \Sigma\|_F^2 \geq \sum_{i>r} \|x_i^t\|^4 \geq A_t^2/n$, where the last inequality uses the Cauchy's inequality. See more details in Appendix B.

## 4 CONVERGENCE OF ASYMMETRIC MATRIX SENSING

Here we investigate the dynamic of GD in the context of the asymmetric matrix sensing problem. Surprisingly, we demonstrate that the convergence rate of gradient descent for asymmetric matrix sensing problems is linear, so long as the initialization is *imbalanced*. However, this linear rate is contingent upon the chosen initialization scale.

## 4.1 A Toy Example of Asymmetric Matrix Factorization

We first use a toy example of asymmetric matrix factorization to demonstrate the behavior of GD. If we assume $\mathcal{A}$ is the identity map, and the loss and the GD update become

$$L(F, G) = \frac{1}{2}\|FG^\top - \Sigma\|_F^2. \tag{4.1}$$

$$F_{t+1} = F_t - \eta(F_t G_t^\top - \Sigma)G_t, \quad G_{t+1} = G_t - \eta(F_t G_t^\top - \Sigma)^\top F_t \tag{4.2}$$

The following theorem tightly characterizes the convergence rate for a toy example.

**Theorem 4.1.** *Consider the asymmetric matrix factorization* (4.1), *with* $k = r + 1$. *Choose* $\eta \in [0, 1/6]$ *and* $\alpha \in [0, 1]$. *Assume that the diagonal matrix* $\Sigma = diag(\sigma_1, \ldots, \sigma_n)$, *where* $\sigma_i = 1$ *for* $i \leq r$ *and is* 0 *otherwise. Also assume that gradient descent* (4.2) *starts at* $F_0, G_0$, *where* $(F_0)_{ii} = \alpha$ *for* $1 \leq i \leq k$, *and* $(G_0)_{ii} = \alpha$ *for* $1 \leq i \leq r$, $(G_0)_{ii} = \alpha/3$ *for* $i = r + 1$, *and all other entries of* $F_0$ *and* $G_0$ *are zero. Then, the iterate* $(F_t, G_t)$ *of* (4.2) *satisfies that*

$$\frac{\alpha^4}{36}(1 - 4\eta\alpha^2)^{2t} \leq \|F_t G_t^\top - \Sigma\|_F^2 \leq 4n \cdot (1 - \eta\alpha^2/4)^{(t-T_1)}, \quad \forall t \geq T_1.$$

*where* $T_1 = c_1 \log(1/\alpha)/\eta$, *and* $c_1$ *is a universal constant.*

The above initialization implicitly assumes that we know the singular vectors of $\Sigma$. Such an assumption greatly simplifies our presentations below. Note that we have a different initialization scale for $F_t$ and $G_t$. As we shall see, such an *imbalance* is the key to establishing the convergence of $F_t G_t^\top$.

We introduce some notations before our proof. Denote the matrix of the first $r$ row of $F, G$ as $U, V \in \mathbb{R}^{r \times k}$ respectively, and the matrix of the last $n - r$ row of $F, G$ as $J, K \in \mathbb{R}^{(n-r) \times k}$ respectively. Further denote the corresponding iterate of gradient descent as $U_t, V_t, J_t$, and $K_t$. The difference $F_t G_t^\top - \Sigma$ can be written in a block form as $F_t G_t^\top - \Sigma = \begin{pmatrix} U_t V_t^\top - \Sigma_r & J_t V_t^\top \\ U_t K_t^\top & J_t K_t^\top \end{pmatrix}$ where $\Sigma_r \in \mathbb{R}^{r \times r}$ is the identity matrix. Hence, we may bound $\|F_t G_t^\top - \Sigma\|$ by

$$\|J_t K_t^\top\| \leq \|F_t G_t^\top - \Sigma\| \leq \|U_t V_t^\top - \Sigma_r\| + \|J_t V_t^\top\| + \|U_t K_t^\top\| + \|J_t K_t^\top\|. \tag{4.3}$$

From (4.3), we shall upper bound $\|U_t V_t^\top - \Sigma_r\|, \|J_t V_t^\top\|, \|U_t K_t^\top\|$, and $\|J_t K_t^\top\|$, and lower bound $\|J_t K_t\|^\top$. Let us now prove Theorem 4.1.

*Proof.* With our particular initialization (4.2), we have the following equality for all $t$:

$$U_t K_t^\top = 0, \quad J_t V_t^\top = 0, \quad U_t = V_t, \quad J_t = a_t A, \quad K_t = b_t A, \quad U_t = (\alpha_t I_r, 0), \tag{4.4a}$$

$$a_0 = \alpha, \quad b_0 = \alpha/3, \quad \alpha_0 = \alpha \tag{4.4b}$$

$$a_{t+1} = a_t - \eta a_t b_t^2, \tag{4.4c}$$

$$b_{t+1} = b_t - \eta a_t^2 b_t. \tag{4.4d}$$

$$\alpha_{t+1} = \alpha_t(1 + \eta - \eta\alpha_t^2), \tag{4.4e}$$

where $A \in \mathbb{R}^{(n-r) \times k}$ is the matrix that $(A)_{1k} = 1$ and other elements are all zero, and $a_t, b_t, \alpha_t \in \mathbb{R}$. We leave the detailed verification of (4.4) to Appendix C. By considering (4.3) and (4.4), we see that we only need to keep track of three sequences $a_t, b_t, \alpha_t$. In particular, we have the following inequalities for $a_t, b_t, \alpha_t$ for all $t \geq T_1$:

$$a_t \in \left[\frac{1}{2}\alpha, \alpha\right], \quad b_t \in \left[\frac{\alpha}{3}(1 - 4\eta\alpha^2)^t, \frac{\alpha}{3}(1 - \frac{\eta\alpha^2}{4})^{t/2}\right], \quad \text{and } |\alpha_t - 1| \leq (1 - \eta/2)^{t-T_1}. \tag{4.5}$$

It is then easy to derive the upper and lower bounds. We leave the detail in checking (4.5) to Appendix C. Our proof is complete. □

**Technical Insight.** This proof of the toy case tells us why the imbalance initialization in the asymmetric matrix factorization helps us to break the $\Omega(1/T^2)$ convergence rate lower bound of the symmetric case. Since we initialize $F_0$ and $G_0$ with a *different scale*, this difference makes the norm of $K$ converge to zero at a linear rate while keeping $J$ larger than a constant. However, in the symmetric case, we have $a_t = b_t$, so they must both converge to zero when the loss goes to zero (as $\|F_t G_t^\top - \Sigma\| \geq a_t b_t$), leading to a sublinear convergence rate. In short, the imbalance property in the initialization causes faster convergence in the asymmetric case.

## 4.2 THEORETICAL RESULTS FOR ASYMMETRIC MATRIX SENSING

Motivated by the toy case in Section 4.1, the imbalance property is the key ingredient for a linear convergence rate. If we use a slightly imbalanced initialization $F_0 = \alpha \cdot \tilde{F}_0, G_0 = (\alpha/3) \cdot \tilde{G}_0$, where the elements of $\tilde{F}_0$ and $\tilde{G}_0$ are $\mathcal{N}(0, 1/n)$, we have $\|F_0^\top F_0 - G_0^\top G_0\| = \Omega(\alpha^2)$. Then, we can show that the imbalance property keeps true when the step size is small, and thus, the gradient descent (2.2) converges with a linear rate similar to the toy case.

Our result is built upon the recent work (Soltanolkotabi et al., 2023) in dealing with the initial phase. Define the following quantities $\alpha_0, \eta_0$ according to (Soltanolkotabi et al., 2023, Theorem 1):

$$\alpha_0 = \frac{c\sqrt{\sigma_1}}{k^5 \max\{2n, k\}^2} \cdot \left( \frac{\sqrt{k} - \sqrt{r-1}}{\kappa^2 \sqrt{\max\{2n, k\}}} \right)^{C\kappa}, \eta_0 = \frac{c}{k^5 \sigma_1 \log\left( \frac{2\sqrt{2\sigma_1}}{\alpha(\sqrt{k}-\sqrt{r-1})} \right)}, \quad (4.6)$$

where $c$ and $C$ are some numerical constants. Below, we show exact convergence results for both $k = r$ and $k > r$.

**Theorem 4.2.** *Consider the matrix sensing problem (1.4) and the gradient descent (2.2). For some numerical constants $c_i > 0$, $1 \le i \le 7$, if the search rank $k$ satisfies $r < k < \frac{n}{8}$, the initial scale $\alpha$ and $\eta$ satisfy*

$$\alpha \le \min\left\{ c_1 \kappa^{-2} \sqrt{\sigma_r}, \alpha_0 \right\}, \quad \eta \le \min\left\{ c_1 \alpha^4/\sigma_1^3, \eta_0 \right\}, \quad (4.7)$$

*where $\alpha_0, \eta_0$ are defined in (4.6), and the operator $\mathcal{A}$ has the RIP of order $2k + 1$ with constant $\delta$ satisfying*

$$\delta\sqrt{2k+1} \le \min\left\{ c_1 \kappa^{-6} \log^{-1}(\sqrt{\sigma_r}/(n\alpha)), \frac{c_1}{\kappa^3 \sqrt{r}}, 1/128 \right\}, \quad (4.8)$$

*then the gradient descent (2.2) starting with $F_0 = \alpha \cdot \tilde{F}_0, G_0 = (\alpha/3) \cdot \tilde{G}_0$, where $\tilde{F}_0, \tilde{G}_0 \in \mathbb{R}^{n \times k}$ whose entries are i.i.d. $\mathcal{N}(0, 1/n)$, satisfies that*

$$\|F_t G_t^\top - \Sigma\|_F^2 \le \frac{\sigma_r^4 n}{c_7 \alpha^4 \kappa^2} \left( 1 - \frac{\eta \alpha^2}{8} \right)^{t/4}, \quad \forall t \ge T^{(1)}, \quad (4.9)$$

*with probability at least $1 - 2e^{-c_2 n} - c_3 e^{-c_4 k} - (c_5 v)^{(k-r+1)}$, where $T^{(1)} = c_6 \log(\sqrt{\sigma_r}/n\alpha v)/\eta\sigma_r$ and $v \in [0, 1]$ is an arbitrary parameter.*

Next, we state our results on exact parametrization.

**Theorem 4.3.** *Consider the same setting as Theorem 4.2 except assuming $k = r$, then with probability at least $1 - 2e^{-c_2 n} - c_3 e^{-c_4 k} - c_5 v$, the gradient descent (2.2) achieves*

$$\|F_t G_t^\top - \Sigma\|_F^2 \le 2n\sigma_r \cdot \left( 1 - \frac{\eta \sigma_r^2}{64\sigma_1} \right)^t, \quad \forall t \ge T^{(2)}, \quad (4.10)$$

*where $T^{(2)} = c_7 \log(\sqrt{\sigma_r}/n\alpha v)/\eta\sigma_r$ for some numerical constant $c_7$.*

Now we highlight two bullet points of Theorem 4.2 and 4.3.

**Exact Convergence.** The main difference between the above theorems and previous convergence results in (Soltanolkotabi et al., 2023) is that we prove the *exact convergence* property, i.e., the loss finally degenerates to zero when $T$ tends to infinity (cf. Table 1). Moreover, we prove that the convergence rate of the gradient descent depends on the initialization scale $\alpha$, which matches our empirical observations in Figure 1.2.

**Discussions about Parameters.** First, since we utilize the initial phase result in (Soltanolkotabi et al., 2023) to guarantee that the loss degenerates to a small scale, our parameters $\delta, \alpha$, and $\eta$ should satisfy the requirement $\delta_0 = \mathcal{O}(\frac{1}{\kappa^3 \sqrt{r}}), \alpha_0, \eta_0$ in (Soltanolkotabi et al., 2023). We further require $\delta_{2k+1} = \widetilde{\mathcal{O}}(\kappa^{-6})$, $\alpha = \mathcal{O}(\kappa^{-2}\sqrt{\sigma_r})$, which are both polynomials of the conditional number $\kappa$. In addition, the step size $\eta$ has the requirement $\eta = \mathcal{O}(\alpha^4/\sigma_1^3)$, which can be much smaller than the requirements $\eta = \widetilde{\mathcal{O}}(1/\kappa^5 \sigma_1)$ in (Soltanolkotabi et al., 2023). In Section 5, we propose a novel algorithm that allows larger learning rate which is independent of $\alpha$.

**Technical insight** Similar to the asymmetric matrix factorization case in the proof of Theorem 4.1, the main effort is in characterizing the behavior of $J_t K_t^\top$. In particular, the update rule of $K_t$ is

$$K_{t+1} = K_t(1 - \eta F_t^\top F_t) + \eta E, \tag{4.11}$$

where $E$ is some error matrix since $\mathcal{A}$ is not an identity. Because of our initialization, we know the following holds for $t = 0$ and $\Delta_t = F_t^\top F_t - G_t^\top G_t$,

$$c\alpha^2 I \preceq \Delta_t \preceq C\alpha^2 I. \tag{4.12}$$

for some numerical constant $c, C > 0$. Hence, we can show $\|K_t\|$ shrinks towards $0$ so long as (4.11) is true, $E = 0$, and $G_t$ is well-bounded. Indeed, we can prove (4.12) and $G_t, J_t$ upper bounded for all $t \geq 0$ via a proper induction. We may then be tempted to conclude $J_t K_t^\top$ converges to $0$. However, the actual analysis of the gradient descent (2.2) for matrix sensing is much more complicated due to the error $E$. It is now unclear whether $\|K_t\|$ will shrink under (4.12). To deal with it, we further consider the structure of $E$. We leave the details to Appendix D.

## 5  A SIMPLE FAST CONVERGENCE METHOD

As discussed in Section 4, the fundamental reason that the convergence rate depends on the initialization scaling $\alpha$ is that the imlabace between $F$ and $G$ determines the convergence rate, but the imbalance between $F$ and $G$ remains at the initialization scale. This observation motivates us to do a straightforward additional step in one iteration to *intensify the imbalance*. Specifically, suppose at the $T_0$ iteration we have reached a neighborhood of an optimum that satisfies: $\|\mathcal{A}^* \mathcal{A}(\widetilde{F}_{T^{(3)}} \widetilde{G}_{T^{(3)}}^\top - \Sigma)\| \leq \gamma$ where the radius $\sigma_r^{1/4} \cdot \|F_{T^{(3)}} G_{T^{(3)}}^\top\|^{3/4}/8$ is chosen for some technical reasons (cf. Section F). Here we use $\widetilde{F}_t$ and $\widetilde{G}_t$ to denote the iterates before we make the change we describe below and $F_t$ and $G_t$ to denote the iterates after make the change.

Let the singular value decomposition of $\widetilde{F}_{T^{(3)}} = A\Sigma'B$ with the diagonal matrix $\Sigma' \in \mathbb{R}^{k \times k}$ and $\Sigma'_{ii} = \sigma'_i$, then let $\Sigma_{inv} \in \mathbb{R}^{k \times k}$ be a diagonal matrix and $(\Sigma_{inv})_{ii} = \beta/\sigma'_i$ for some small constant $\beta = O(\sigma_r)$, then we transform the matrix $F_{T^{(3)}}, G_{T^{(3)}}$ by

$$F_{T^{(3)}} = \widetilde{F}_{T^{(3)}} B^\top \Sigma_{inv}, \quad G_{T^{(3)}} = \widetilde{G}_{T^{(3)}} B \Sigma_{inv}^{-1} \tag{5.1}$$

We can show that, when $F$ and $G$ have reached a local region of an optimum, their magnitude will have similar scale as $M^\star$. Therefore, the step Equation (5.1) can create an imbalance between $F$ and $G$ as large the magnitude of $M^\star$, which is significantly larger than the initial scaling $\alpha$. The following theorem shows we can obtain a convergence rate independent of the initialization scaling $\alpha$. The proof is deferred to Appendix F.

**Theorem 5.1.** *With the same setting as Theorem 4.2, suppose that at the step $T^{(3)}$ we have $\|\mathcal{A}^* \mathcal{A}(\widetilde{F}_{T^{(3)}} \widetilde{G}_{T^{(3)}}^\top - \Sigma)\| \leq \gamma$ for some $\gamma > 0$, and we do one step as in Equation (5.1). Then with probability at least $1 - 2e^{-c_2 n} - c_3 e^{-c_4 k} - (c_5 \upsilon)^{(k-r+1)}$, we have for all $t > T^{(3)}$,*

$$\|F_t G_t^\top - \Sigma\|_F^2 \leq \frac{n\beta^{12}}{\sigma_1^4} \left(1 - \frac{\eta\beta^2}{2}\right)^{2(t-T^{(3)})},$$

*so long as $0 < c_7 \gamma^{1/6} \sigma_1^{1/3} \leq \beta \leq c_8 \sigma_r$, and the step size satisfies $\eta \leq c_9 \beta^2/\sigma_1^2$ from the iteration $T^{(3)} \leq c_{10} \log(\sqrt{\sigma_r}/n\alpha\upsilon)/\eta\sigma_r$ for some positive numerical constants $c_i, i = 1, \ldots, 10$.*

## 6  CONCLUSION

This paper demonstrated qualitatively different behaviors of GD in the exact-pasteurization and over-pasteurization scenarios in symmetric and asymmetric settings. For the symmetric matrix sensing problem, we provide a $\Omega(1/T^2)$ lower bound. For the asymmetric matrix sensing problem, we show that the gradient descent converges at a linear rate, where the rate is dependent on the initialization scale. Moreover, we introduce a simple procedure to get rid of the initialization scale dependency. We believe our analyses are also useful for other problems, such as deep linear networks.

ACKNOWLEDGMENTS

Simon S. Du is supported by supported by NSF IIS 2110170, NSF DMS 2134106, NSF CCF 2212261, NSF IIS 2143493, NSF CCF 2019844, NSF IIS 2229881.

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

# Supplementary Material

### CONTENTS

# Appendix

## A   RELATED WORK

**Matrix Sensing.**   Matrix sensing aims to recover the low-rank matrix based on measurements. Candes & Recht (2012); Liu et al. (2012) propose convex optimization-based algorithms, which minimize the nuclear norm of a matrix, and Recht et al. (2010) show that projected subgradient methods can recover the nuclear norm minimizer. Wu & Rebeschini (2021) also propose a mirror descent algorithm, which guarantees to converge to a nuclear norm minimizer. See (Davenport & Romberg, 2016) for a comprehensive review.

**Non-Convex Low-Rank Factorization Approach.**   The nuclear norm minimization approach involves optimizing over a $n \times n$ matrix, which can be computationally prohibitive when $n$ is large. The factorization approach tries to use the product of two matrices to recover the underlying matrix, but this formulation makes the optimization problem non-convex and is significantly more challenging for analysis. For the exact-parameterization setting ($k = r$), Tu et al. (2016); Zheng & Lafferty (2015) shows the linear convergence of gradient descent when starting at a local point that is close to the optimal point. This initialization can be implemented by the spectral method. For the over-parameterization scenario ($k > r$), in the symmetric setting, Stöger & Soltanolkotabi (2021) shows that with a small initialization, the gradient descent achieves a small error that *dependents* on the initialization scale, rather than the *exact-convergence*. Zhuo et al. (2021) shows exact convergence with $\mathcal{O}(1/T^2)$ convergence rate in the overparamterization setting. These two results together imply the global convergence of randomly initialized GD with an $O\left(1/T^2\right)$ convergence rate *upper bound*. Jin et al. (2023) also provides a fine-grained analysis of the GD dynamics. More recently, Zhang et al. (2021b; 2023) empirically observe that in practice, in the over-parameterization case, GD converges with a sublinear rate, which is exponentially slower than the rate in the exact-parameterization case, and coincides with the prior theory's upper bound (Zhuo et al., 2021). However, no rigorous proof of the *lower bound* is given whereas we bridge this gap. On the other hand, Zhang et al. (2021b; 2023) propose a preconditioned GD algorithm with a shrinking damping factor to recover the linear convergence rate. Xu et al. (2023) show that the preconditioned GD algorithm with a constant damping factor coupled with small random initialization requires a less stringent assumption on $\mathcal{A}$ and achieves a linear convergence rate up to some prespecified error. Ma & Fattahi (2023) study the performance of the subgradient method with $L_1$ loss under a different set of assumptions on $\mathcal{A}$ and showed a linear convergence rate up to some error related to the initialization scale. We show that by simply using the *asymmetric parameterization*, without changing the GD algorithm, we can still attain the linear rate.

For the asymmetric matrix setting, many previous works (Ye & Du, 2021; Ma et al., 2021; Tong et al., 2021; Ge et al., 2017; Du et al., 2018a; Tu et al., 2016; Zhang et al., 2018a;b; Wang et al., 2017; Zhao et al., 2015) consider the exact-parameterization case ($k = r$). Tu et al. (2016) adds a balancing regularization term $\frac{1}{8}\|F^\top F - G^\top G\|_F^2$ to the loss function, to make sure that $F$ and $G$ are balanced during the optimization procedure and obtain a local convergence result. More recently, some works (Du et al., 2018a; Ma et al., 2021; Ye & Du, 2021) show GD enjoys an *auto-balancing* property where $F$ and $G$ are approximately balanced; therefore, additional balancing regularization is unnecessary. In the asymmetric matrix factorization setting, Du et al. (2018a) proves a global convergence result of GD with a diminishing step size and the GD recovers $M^*$ up to some error. Later, Ye & Du (2021) gives the first global convergence result of GD with a constant step size. Ma et al. (2021) shows linear convergence of GD with a local initialization and a larger stepsize in the asymmetric matrix sensing setting. Although exact-parameterized asymmetric matrix factorization and matrix sensing problems have been explored intensively in the last decade, our understanding of the over-parameterization setting, i.e., $k > r$, remains limited. Jiang et al. (2022) considers the asymmetric matrix factorization setting, and proves that starting with a small initialization, the vanilla gradient descent sequentially recovers the principled component of the ground-truth matrix. Soltanolkotabi et al. (2023) proves the convergence of gradient descent in the asymmetric matrix sensing setting. Unfortunately, both works only prove that GD achieves a small error when stopped early, and the error depends on the initialization scale. Whether the gradient descent can achieve *exact-convergence* remains open, and we resolve this problem by novel analyses. Furthermore, our analyses highlight the importance of the *imbalance between $F$ and $G$*.

Lastly, we want to remark that we focus on gradient descent for $L_2$ loss, there are works on more advanced algorithms and more general losses (Tong et al., 2021; Zhang et al., 2021b; 2023; 2018a;b; Ma & Fattahi, 2021; Wang et al., 2017; Zhao et al., 2015; Bhojanapalli et al., 2016; Xu et al., 2023). We believe our theoretical insights are also applicable to those setups.

**Landscape Analysis of Non-convex Low-rank Problems.** The aforementioned works mainly focus on studying the dynamics of GD. There is also a complementary line of works that studies the landscape of the loss functions, and shows the loss functions enjoy benign landscape properties such as (1) all local minima are global, and (2) all saddle points are strict Ge et al. (2017); Zhu et al. (2018); Li et al. (2019); Zhu et al. (2021); Zhang et al. (2023). Then, one can invoke a generic result on *perturbed gradient descent*, which injects noise to GD Jin et al. (2017), to obtain a convergence result. There are some works establishing the general landscape analysis for the non-convex low-rank problems. Zhang et al. (2021a) obtains less conservative conditions for guaranteeing the non-existence of spurious second-order critical points and the strict saddle property, for both symmetric and asymmetric low-rank minimization problems. The paper Bi et al. (2022) analyzes the gradient descent for the symmetric case and asymmetric case with a regularized loss. They provide the local convergence result using PL inequality, and show the global convergence for the perturbed gradient descent. We remark that injecting noise is required if one solely uses the landscape analysis alone because there exist exponential lower bounds for standard GD (Du et al., 2017).

**Slowdown Due to Over-parameterization.** Similar exponential slowdown phenomena caused by over-parameterization have been observed in other problems beyond matrix recovery, such as teacher-student neural network training (Xu & Du, 2023; Richert et al., 2022) and Expectation-Maximization algorithm on Gaussian mixture model (Wu & Zhou, 2021; Dwivedi et al., 2020).

## B  PROOF OF THEOREM 3.1

In this proof, we denote

$$X \in \mathbb{R}^{n \times k} = \begin{bmatrix} x_1^\top \\ x_2^\top \\ \cdots \\ x_n^\top \end{bmatrix}, \tag{B.1}$$

where $x_i \in \mathbb{R}^{k \times 1}$ is the transpose of the row vector. Since the updating rule can be written as

$$X_{t+1} = X_t - \eta(X_t X_t^\top - \Sigma)X_t,$$

where we choose $\eta$ instead of $2\eta$ for simplicity, which does not influence the subsequent proof. By substituting the equation (B.1), the updating rule can be written as

$$(x_i^{t+1})^\top = (1 - \eta(\|x_i^t\|^2 - \sigma_i))x_i^\top - \sum_{j=1, j \neq i}^{n} \eta((x_i^t)^\top x_j^t (x_j^t)^\top)$$

where $\sigma_i = 0$ for $i > r$. Denote

$$\theta = \max_{j,k} \frac{(x_j^\top x_k)^2}{\|x_j\|^2 \|x_k\|^2}$$

is the maximum angle between different vectors in $x_1, \cdots, x_n$. We start with the outline of the proof.

### B.1  PROOF OUTLINE OF THEOREM 3.1

Recall we want to establish the key inequalities (3.3). The updating rule (2.3) gives the following lower bound of $x_i^{t+1}$ for $i > r$:

$$\|x_i^{t+1}\|^2 \geq \|x_i^t\|^2 \left(1 - 2\eta \theta_t^U \sum_{j \leq r} \|x_j^t\|^2 - 2\eta \sum_{j > r} \|x_j^t\|^2 \right), \tag{B.2}$$

where the quantity $\theta_t^U = \max_{i,j:\min\{i,j\}\leq r} \theta_{ij,t}$ and the square cosine $\theta_{ij,t} = \cos^2 \angle(x_i, x_j)$. Thus, to establish the key inequalities (3.3), we need to control the quantity $\theta_t^U$. Our analysis then consists of three phases. In the last phase, we show (3.3) holds and our proof is complete.

In the first phase, we show that $\|x_i^t\|^2$ for $i \leq r$ becomes large, while $\|x_i^t\|^2$ for $i > r$ still remains small yet bounded away from 0. In addition, the quantity $\theta_{ij,t}$ remains small. Phase 1 terminates when $\|x_i^t\|^2$ is larger than or equal to $\frac{3}{4}\sigma_i$.

After the first phase terminates, in the second and third phases, we show that $\theta_t^U$ converges to 0 linearly and the quantity $\theta_t^U \sigma_1 / \sum_{j>r} \|x_j^t\|^2$ converges to zero at a linear rate as well. We also keep track of the magnitude of $\|x_i^t\|^2$ and show $\|x_i^t\|$ stays close to $\sigma_i$ for $i \leq r$, and $\|x_i^t\|^2 \leq 2\alpha^2$ for $i > r$.

The second phase terminates once $\theta_t^U \leq \mathcal{O}(\sum_{j>r} \|x_j^t\|^2/\sigma_1)$ and we enter the last phase: the convergence behavior of $\sum_{j>r} \|x_j^t\|^2$. Note with $\theta_t^U \leq \mathcal{O}(\sum_{j>r} \|x_j^t\|^2/\sigma_1)$ and $\|x_i^t\|^2 \leq 2\sigma_r$ for $i \leq r$, we can prove (3.3b). The condition (3.3a) can be proven since the first two phases are quite short and the updating formula of $x_i$ for $i > r$ shows $\|x_i\|^2$ cannot decrease too much.

## B.2 PHASE 1

In this phase, we show that $\|x_i^t\|^2$ for $i \leq r$ becomes large, while $\|x_i^t\|^2$ for $i > r$ still remains small. In addition, the maximum angle between different column vectors remains small. Phase 1 terminates when $\|x_i^t\|^2$ is larger than a constant.

To be more specific, we have the following two lemmas. Lemma B.1 states that the initial angle $\theta_0 = \mathcal{O}(\log^2(r\sqrt{\sigma_1}/\alpha)(r\kappa)^2)$ is small because the vectors in the high-dimensional space are nearly orthogonal.

**Lemma B.1.** *For some constant $c_4$ and $c$, if $k \geq \frac{c^2}{16\log^4(r\sqrt{\sigma_1}/\alpha)(r\kappa)^4}$, with probability at least $1 - c_4 n^2 k \exp(-\sqrt{k})$, we have*

$$\theta_0 \leq \frac{c}{\log^2(r\sqrt{\sigma_1}/\alpha)(r\kappa)^2} \tag{B.3}$$

*Proof.* See §G.1 for proof. □

Lemma B.2 states that with the initialization scale $\alpha$, the norm of randomized vector $x_i^0$ is $\Theta(\alpha^2)$.

**Lemma B.2.** *With probability at least $1 - 2n\exp(-c_5 k/4)$, for some constant $c$, we have*

$$\|x_i^0\|^2 \in [\alpha^2/2, 2\alpha^2].$$

*Proof.* See §G.2 for the proof. □

Now we prove the following three conditions by induction.

**Lemma B.3.** *There exists a constant $C_1$, such that $T_1 \leq C_1(\log(\sqrt{\sigma_1}/n\alpha)/\eta\sigma_r)$ and then during the first $T_1$ rounds, with probability at least $1 - 2c_4 n^2 k \exp(-\sqrt{k}) - 2n\exp(-c_5 k/4)$ for some constant $c_4$ and $c_5$, the following four statements always hold*

$$\|x_i^t\|^2 \leq 2\sigma_1 \tag{B.4}$$

$$\alpha^2/4 \leq \|x_i^t\|^2 \leq 2\alpha^2 \quad (i > r) \tag{B.5}$$

$$2\theta_0 \geq \theta_t \tag{B.6}$$

*Also, if $\|x_i^t\|^2 \leq 3\sigma_i/4$, we have*

$$\|x_i^{t+1}\|^2 \geq (1 + \eta\sigma_r/4)\|x_i^t\|^2. \tag{B.7}$$

*Moreover, at $T_1$ rounds, $\|x_i^{T_1}\|^2 \geq 3\sigma_i/4$, and Phase 1 terminates.*

*Proof.* By Lemma B.1 and Lemma B.2, with probability at least $1 - 2c_4 n^2 k \exp(-\sqrt{k}) - 2n \exp(-c_5 k/4)$, we have $\|x_i^0\|^2 \in [\alpha^2/2, 2\alpha^2]$ for $i \in [n]$, and $\theta_0 \leq \frac{c}{\log^2(r\sqrt{\sigma_1}/\alpha)(r\kappa)^2}$. Then assume that the three conditions hold for rounds before $t$, then at the $t+1$ round, we proof the four statements above one by one.

**Proof of Eq.(B.5)** For $i > r$, we have

$$(x_i^{t+1})^\top = (x_i^t)^\top - \eta \sum_{j=1}^n (x_i^t)^\top x_j^t (x_j^t)^\top$$

Then, the updating rule of $\|x_i^t\|^2$ can be written as

$$\|(x_i^{t+1})\|_2^2 = \|x_i^t\|^2 - 2\eta \sum_{j=1}^n ((x_i^t)^\top x_j^t)^2 + \eta^2 (\sum_{j,k=1}^n (x_i^t)^\top x_j^t (x_j^t)^\top x_k^t (x_k^t)^\top x_i^t) \leq \|x_i^t\|^2. \quad \text{(B.8)}$$

The last inequality in (B.8) is because

$$(x_i^t)^\top x_j^t (x_j^t)^\top x_k^t (x_k^t)^\top (x_i^t) \leq (x_j^t)^\top x_k^t (((x_i^t)^\top x_j^t)^2 + ((x_k^t)^\top x_i^t)^2)/2 \quad \text{(B.9)}$$
$$\leq \sigma_1 (((x_i^t)^\top x_j^t)^2 + ((x_k^t)^\top x_i^t)^2), \quad \text{(B.10)}$$

and then

$$\eta^2 \sum_{j,k=1}^n (x_i^t)^\top x_j^t (x_j^t)^\top x_k^t (x_k^t)^\top (x_i^t) \leq \eta^2 \sum_{j,k=1}^n \sigma_1 (((x_i^t)^\top x_j^t)^2 + ((x_k^t)^\top x_i^t)^2)$$

$$= \eta^2 \cdot n\sigma_1 \sum_{j=1}^n ((x_i^t)^\top x_j^t)^2$$

$$\leq \eta \sum_{j=1}^n ((x_i^t)^\top x_j^t)^2. \quad \text{(B.11)}$$

where the last inequality holds because $\eta \leq 1/n\sigma_1$. Thus, the $\ell_2$-norm of $x_i^\top$ does not increase, and the right side of Eq.(B.5) holds.

Also, we have

$$\|x_i^{t+1}\|^2 \geq \|x_i^t\|^2 - 2\eta \sum_{j=1}^n ((x_i^t)^\top x_j^t)^2 + \eta^2 \left\| \sum_{j=1}^n (x_i^t)^\top x_j^t (x_j^t)^\top \right\|^2$$

$$\geq \|x_i^t\|^2 - \|x_i^t\|^2 \cdot 2\eta\theta_t \cdot \sum_{j \neq i}^n \|x_j^t\|^2 - 2\eta\|x_i\|^4 \quad \text{(B.12)}$$

Equation (B.2) is because $\frac{((x_i^t)^\top x_j^t)^2}{\|x_i^t\|^2 \|x_j^t\|^2} = \theta_{ij,t} \leq \theta_t$. Now by (B.4) and (B.5), we can get

$$\sum_{j \neq i}^n \|x_j^t\|^2 \leq r \cdot 2\sigma_1 + (n-r) \cdot 2\alpha^2 \leq 2\sigma_1 + 2n\alpha^2$$

Hence, we can further derive

$$\|x_i^{t+1}\|^2 \geq \|x_i^t\|^2 \cdot \left(1 - 2\eta\theta_t(2r\sigma_1 + 2n\alpha^2) - 2\eta \cdot 2\alpha^2\right)$$
$$\geq \|x_i^t\|^2 \cdot \left(1 - \eta(8\theta_t\sigma_1 + 4\alpha^2)\right),$$

where the last inequality is because $\alpha \leq \sqrt{r\sigma_1}/\sqrt{n}$. Thus, by $(1-a)(1-b) \geq (1-a-b)$ for $a, b > 0$, we can get

$$\|x_i^{T_1}\|^2 \geq \|x_i^0\|^2 \cdot (1 - \eta(8\theta_t\sigma_1 + 4\alpha^2))^{T_1}$$

$$\geq \frac{\alpha^2}{2} \cdot (1 - T_1\eta(8 \cdot (2\theta_0)\sigma_1 + 4\alpha^2)) \quad \text{(B.13)}$$

$$\geq \frac{\alpha^2}{4}. \quad \text{(B.14)}$$

Equation (B.13) holds by induction hypothesis (B.6), and the last inequality is because of our choice on $T_1$, $\alpha$, and $\theta_0 \leq O(\frac{1}{r\kappa \log(\sqrt{\sigma_1}/\alpha)})$ from the induction hypothesis. Hence, we complete the proof of Eq.(B.5).

**Proof of Eq.(B.7)** For $i \leq r$, if $\|x_i^t\|^2 \leq 3\sigma_i/4$, by the updating rule,

$$\|x_i^{t+1}\|_2^2 \geq (1 - \eta(\|x_i^t\|^2 - \sigma_i))^2 \|x_i^t\|^2 - 2\eta \sum_{j \neq i}^n ((x_i^t)^\top x_j^t)^2 + \eta^2(\|x_i^t\|^2 - \sigma_i) \sum_{j \neq i}^n ((x_i^t)^\top x_j^t)^2$$
(B.15)

$$\geq (1 - \eta(\|x_i^t\|^2 - \sigma_i))^2 \|x_i^t\|^2 - 2\eta \sum_{j \neq i}^n ((x_i^t)^\top x_j^t)^2 - \eta^2 |\|x_i^t\|^2 - \sigma_i| \cdot \sum_{j \neq i}^n \|x_i^t\|^2 \|x_j^t\|^2$$

$$\geq (1 - \eta(\|x_i^t\|^2 - \sigma_i))^2 \|x_i^t\|^2 - 2\eta \sum_{j \neq i}^n ((x_i^t)^\top x_j^t)^2 - 4\eta^2 (n\sigma_1^2) \|x_i^t\|^2.$$

THe last inequality uses the fact that $|\|x_i^t\|^2 - \sigma_i| \leq 2\sigma_1$ and $\|x_j^t\|^2 \leq 2\sigma_1$. Then, by $((x_i^t)^\top x_j^t)^2 \leq \|x_i^t\|^2 \|x_j^t\|^2 \cdot \theta$, we can further get

$$\|x_i^{t+1}\|^2 \geq \left( 1 - 2\eta(\|x_i^t\|^2 - \sigma_i) - 2\eta \sum_{j \neq i}^n \|x_j^t\|^2 \theta - 2\eta^2(n\sigma_1^2) \right) \|x_i^t\|^2$$

$$\geq (1 + \eta\sigma_i/2 - 2\eta^2(n\sigma_1^2) - \eta\sigma_r/16) \|x_i^t\|^2$$
(B.16)

$$\geq (1 + \sigma_i(\eta/2 - \eta/16 - \eta/16)) \|x_i^t\|^2$$
(B.17)

$$\geq (1 + \eta\sigma_i/4) \|x_i^t\|^2.$$

The inequality (B.16) uses the fact $\theta \leq 2\theta_0 \leq \frac{1}{128\kappa r}$ and $\sum_{j \neq i}^n \|x_j\|^2 \leq 2\sigma_1 r + 2n\alpha^2 \leq 4\sigma_1 r \leq \frac{\sigma_r}{32\theta}$. The inequality (B.17) uses the fact that $\eta \leq \frac{1}{32n\sigma_1^2}$.

**Proof of Eq.(B.4)** If $\|x_i^t\|^2 \geq 3\sigma_i/4$, by the updating rule, we can get

$$\left| \|x_i^{t+1}\|_2^2 - \sigma_i \right| \leq \left( 1 - 2\eta\|x_i^t\|^2 + \eta^2(\|x_i^t\|^2 - \sigma_i)\|x_i^t\|^2 + \eta^2 \sum_{j \neq i}^n ((x_i^t)^\top x_j^t)^2 \right) |\|x_i^t\|^2 - \sigma_i|$$

$$+ 2\eta \sum_{j \neq i}^n ((x_i^t)^\top x_j^t)^2 + \eta^2 \left( \sum_{j,k \neq i}^n ((x_i^t)^\top x_j^t (x_j^t)^\top x_k^t (x_k^t)^\top x_i^t) \right)$$

$$\leq (1 - \eta\sigma_i) |\|x_i^t\|^2 - \sigma_i| + 3 \underbrace{\eta \sum_{j \neq i}^n ((x_i^t)^\top x_j^t)^2}_{(a)}$$
(B.18)

The last inequality holds by Eq.(B.11) and

$$2\eta\|x_i^t\|^2 - \eta^2(\|x_i^t\|^2 - \sigma_i)\|x_i^t\|^2 - 2\eta^2 \sum_{j \neq i}^n ((x_i^t)^\top x_j^t)^2$$
(B.19)

$$\geq \frac{3\eta}{2}\sigma_i - \eta^2(2\sigma_1) \cdot 2\sigma_1 - 2\eta^2 n\sigma_1^2$$
(B.20)

$$\geq \eta\sigma_i,$$
(B.21)

where (B.20) holds by $\|x_i^t\|^2 \geq \frac{3\sigma_i}{4}$, $\|x_i^t\|^2 \leq 2\sigma_1$ for all $i \in [n]$. The last inequality (B.21) holds by $\eta \leq C(\frac{1}{n\sigma_1\kappa})$ for small constant $C$. The first term of (B.18) represents the main converge part, and (a) represents the perturbation term. Now for the perturbation term (a), since $\alpha \leq \frac{1}{4\kappa n^2}$ and

$\theta \leq 2\theta_0 \leq \frac{1}{20r\kappa^2} = \frac{\sigma_i^2}{20r\sigma_1^2}$, we can get

$$(a) = \sum_{j \neq i, j \leq r} ((x_i^t)^\top x_j^t)^2 + \sum_{j \neq i, j > r} ((x_i^t)^\top x_j^t)^2 \tag{B.22}$$

$$\leq (r\sigma_1 + 2n\alpha^2)\theta_t \cdot 2\sigma_1 \tag{B.23}$$

$$\leq 2r\sigma_1 \cdot \theta_t \cdot 2\sigma_1 \tag{B.24}$$

$$= 4r\sigma_1^2 \cdot \theta_t$$

$$\leq \sigma_i^2/5, \tag{B.25}$$

where (B.23) holds by (B.4) and (B.5). (B.24) holds by $\alpha = \mathcal{O}(\sqrt{r\sigma_1/n})$, and the last inequality (B.25) holds by $\theta$ is small, i.e. $\theta_t \leq 2\theta_0 = \mathcal{O}(1/r\kappa^2)$. Now it is easy to get that $(x_i^{t+1})^\top x_i^{t+1} \leq 2\sigma_i$ by

$$|\|x_i^{t+1}\|^2 - \sigma_i| \leq (1 - \eta\sigma_i)(\|x_i^t\|^2 - \sigma_i) + \frac{3\eta\sigma_i^2}{5} \leq (1 - \eta\sigma_i)\sigma_i + \frac{3\eta\sigma_i^2}{5} \leq \sigma_i. \tag{B.26}$$

Hence, we complete the proof of Eq.(B.4).

**Proof of Eq.(B.6)** Now we consider the change of $\theta$. For $i \neq j$, denote

$$\theta_{ij,t} = \frac{((x_i^t)^\top x_j^t)^2}{\|x_i\|^2 \|x_j\|^2}$$

Now we first calculate the $(x_i^{t+1})^\top x_j^{t+1}$ by the updating rule:

$$(x_i^{t+1})^\top x_j^{t+1}$$
$$= \underbrace{\left(1 - \eta(\|x_i^t\|^2 - \sigma_i)\right)\left(1 - \eta(\|x_j^t\|^2 - \sigma_j)\right)(x_i^t)^\top x_j^t}_{A} \underbrace{-\eta\|x_j^t\|^2(1 - \eta(\|x_j^t\|^2 - \sigma_j))(x_i^t)^\top x_j^t}_{B}$$

$$\underbrace{-\eta\|x_i^t\|^2(1 - \eta(\|x_i^t\|^2 - \sigma_j))(x_i^t)^\top x_j^t}_{C} + \underbrace{\eta^2 \sum_{k,l \neq i,j} (x_i^t)^\top x_k^t (x_k^t)^\top x_l^t (x_l^t)^\top x_j^t}_{D}$$

$$\underbrace{-\eta(2 - \eta(\|x_i^t\|^2 - \sigma_i) - \eta(\|x_j^t\|^2 - \sigma_j)) \sum_{k \neq i,j}^{n} (x_i^t)^\top x_k^t (x_k^t)^\top x_j}_{E}$$

$$+ \underbrace{\eta^2 \sum_{k \neq i,j} x_i^\top x_j^t (x_j^t)^\top x_k^t (x_k^t)^\top x_j^t + \eta^2 \sum_{k \neq i,j} (x_i^t)^\top x_k^t (x_k^t)^\top x_i^t (x_i^t)^\top x_j^t}_{F}.$$

Now we bound A, B, C, D, E and F respectively. First, by $\|x_i^t\|^2 \leq 2\sigma_1$ for any $i \in [m]$, we have

$$A \leq \left(1 - \eta(\|x_i^t\|^2 - \sigma_i) - \eta(\|x_j^t\|^2 - \sigma_j) + \eta^2(\|x_i^t\|^2 - \sigma_i)\left(\|x_j^t\|^2 - \sigma_j\right)\right)(x_i^t)^\top x_j^t$$

$$\leq \left(1 - \eta\left(\|x_i^t\|^2 + \|x_j^t\|^2 - \sigma_i - \sigma_j\right) + \eta^2 \cdot 4\sigma_1^2\right)(x_i^t)^\top x_j^t, \tag{B.27}$$

Now we bound term B. We have

$$B + C = \left(-\eta(\|x_i^t\|^2 + \|x_j^t\|^2) + \eta^2\left((\|x_j^t\|^2 - \sigma_j)\|x_j^t\|^2 + (\|x_i^t\|^2 - \sigma_i)\|x_i^t\|^2\right)\right)(x_i^t)^\top x_j^t$$

$$\leq \left(-\eta(\|x_i^t\|^2 + \|x_j^t\|^2) + \eta^2 \cdot (8\sigma_1^2)\right)(x_i^t)^\top x_j^t. \tag{B.28}$$

Then, for D, by $\theta_t \leq 1$, we have

$$D = \eta^2 \left( \sum_{k,l \neq i,j} \|x_k^t\|^2 \|x_l^t\|^2 \cdot \sqrt{\theta_{ik,t}\theta_{kl,t}\theta_{lj,t}/\theta_{ij,t}} \right)(x_i^t)^\top x_j^t$$

$$\leq \left( \eta^2 \cdot n^2 \cdot 4\sigma_1^2 \cdot \theta_t/\sqrt{\theta_{ij,t}} \right)(x_i^t)^\top x_j^t. \tag{B.29}$$

For E, since we have

$$\mathrm{E} \leq 2\eta \sum_{k\neq i,j} |(x_i^t)^\top x_k^t (x_k^t)^\top x_j^t| + 4\sigma_1 \eta^2 \sum_{k\neq i,j} |(x_i^t)^\top x_k^t (x_k^t)^\top x_j^t|$$

$$\leq \left( 2\eta \sum_{k\neq i,j} \|x_k^t\|^2 \cdot \sqrt{\theta_{ik,t}\theta_{kj,t}/\theta_{ij,t}} + 4\sigma_1 \eta^2 \sum_{k\neq i,j} \|x_k^t\|^2 \cdot \sqrt{\theta_{ik,t}\theta_{kj,t}/\theta_{ij,t}} \right) (x_i^t)^\top x_j^t$$

$$\leq \left( 2\eta \sum_{k\neq i,j} \|x_k^t\|^2 \cdot \sqrt{\theta_{ik,t}\theta_{kj,t}/\theta_{ij,t}} + 4n\sigma_1 \eta^2 \cdot (2\sigma_1) \cdot \theta_t/\sqrt{\theta_{ij,t}} \right) (x_i^t)^\top x_j^t. \tag{B.30}$$

Lastly, for F, since $(x_j^t)^\top x_k^t (x_k^t)^\top x_j^t \leq \|x_j^t\|^2 \|x_k^t\|^2 \leq 4\sigma_1^2$, we have

$$\mathrm{F} \leq \eta^2 8n\sigma_1^2 (x_i^t)^\top x_j^t. \tag{B.31}$$

Now combining (B.27), (B.28), (B.29), (B.30) and (B.31), we can get

$$(x_i^{t+1})^\top x_j^{t+1} \tag{B.32}$$

$$\leq \left( 1 - \eta(2\|x_i\|^2 + 2\|x_j\|^2 - \sigma_i - \sigma_j) + 2\eta \sum_{k\neq i,j} \|x_k\|^2 \cdot \sqrt{\theta_{ik,t}\theta_{kj,t}/\theta_{ij,t}} + 30n^2\sigma_1^2\eta^2\theta_t/\sqrt{\theta_{ij,t}} \right) (x_i^t)^\top x_j^t. \tag{B.33}$$

On the other hand, consider the change of $\|x_i^t\|^2$. By Eq.(B.15),

$$\|x_i^{t+1}\|^2 \geq (1 - \eta(\|x_i^t\|^2 - \sigma_i))^2 \|x_i^t\|^2 - 2\eta \sum_{j\neq i}^n ((x_i^t)^\top x_j^t)^2 + \eta^2(\|x_i^t\|^2 - \sigma_i) \sum_{j\neq i}^n ((x_i^t)^\top x_j^t)^2$$

$$\geq (1 - 2\eta(\|x_i^t\| - \sigma_i) - 2\eta \sum_{j\neq i}^n \|x_j^t\|^2 \theta_{ij,t} - 4\eta^2 n\theta_t\sigma_1^2)\|x_i^t\|^2$$

$$\geq (1 - 2\eta(\|x_i^t\| - \sigma_i) - 2\eta \sum_{k=1}^n \|x_j^t\|^2 \theta_{ij,t} - 4\eta^2 n\theta_t\sigma_1^2)\|x_i^t\|^2$$

Hence, the norm of $x_i^{t+1}$ and $x_j^{t+1}$ can be lower bounded by

$$\|x_i^{t+1}\|^2 \|x_j^{t+1}\|^2$$

$$\geq \left( 1 - 2\eta(\|x_i^t\|^2 - \sigma_i) - 2\eta(\|x_j^t\|^2 - \sigma_j) - 2\eta \sum_{k\neq i,j} \|x_k\|^2 (\theta_{ik,t} + \theta_{jk,t}) - 2\eta(\|x_j\|^2 + \|x_i\|^2)\theta_{ij,t} \right.$$

$$\left. - 4\eta^2\theta_t n^2 \sigma_1^2 + \sum_{l=i,j} 4\eta^2(\|x_l^t\|^2 - \sigma_l) \sum_{k=1}^n \|x_k^t\|^2 \theta_{ik,t} + \sum_{l=i,j} 2\eta(\|x_l^t\|^2 - \sigma_l)\eta^2 n^2 \theta_t \sigma_1^2 \right) \|x_i^t\|^2 \|x_j^t\|^2$$

$$\geq \left( 1 - 2\eta(\|x_i^t\|^2 - \sigma_i) - 2\eta(\|x_j^t\|^2 - \sigma_j) - 2\eta \sum_{k\neq i,j} \|x_k\|^2 (\theta_{ik,t} + \theta_{jk,t}) - 2\eta(\|x_j\|^2 + \|x_i\|^2)\theta_{ij,t} \right.$$

$$\left. - 4\eta^2\theta_t n^2 \sigma_1^2 - 2 \cdot 4\eta^2 \cdot (2\sigma_1)n \cdot (2\sigma_1)\theta_t - 2 \cdot 4\eta\sigma_1 \cdot \eta^2 n^2 \theta_t \sigma_1^2 \right) \|x_i^t\|^2 \|x_j^t\|^2 \tag{B.34}$$

$$\geq \left( 1 - 2\eta(\|x_i^t\|^2 - \sigma_i) - 2\eta(\|x_j^t\|^2 - \sigma_j) - 2\eta \sum_{k\neq i,j} \|x_k\|^2 (\theta_{ik,t} + \theta_{jk,t}) - 2\eta(\|x_j\|^2 + \|x_i\|^2)\theta_{ij,t} \right.$$

$$\left. - 6\eta^2\theta_t n^2 \sigma_1^2 \right) \|x_i^t\|^2 \|x_j^t\|^2, \tag{B.35}$$

where (B.35) holds by $n > 8k \geq 8$ and $2\eta(\|x_i^t\|^2 - \sigma_i) \leq 4\eta\sigma_1 \leq 1$. Then, by (B.33) and (B.35), we have

$$\theta_{ij,t+1} = \theta_{ij,t} \cdot \frac{(x_i^{t+1})^\top x_j^{t+1}}{(x_i^t)^\top x_j^t} \cdot \frac{\|x_i^{t+1}\|^2 \|x_j^{t+1}\|^2}{\|x_i^t\|^2 \|x_j^t\|^2}$$

$$\leq \theta_{ij,t} \cdot \left( \frac{1 - A + B}{1 - A - C} \right) \tag{B.36}$$

where

$$A = 2\eta(\|x_i^t\|^2 - \sigma_i + \|x_j^t\|^2 - \sigma_i)) \leq 4\eta\sigma_1 \tag{B.37}$$

$$B = 2\eta\|x_k\|^2 \cdot \sqrt{\theta_{ik,t}\theta_{kj,t}/\theta_{ij,t}} + 30n^2\sigma_1^2\eta^2\theta_t/\sqrt{\theta_{ij,t}} \tag{B.38}$$

and

$$C = 2\eta \sum_{k \neq i,j} \|x_k\|^2(\theta_{ik,t} + \theta_{jk,t}) + 2\eta(\|x_j\|^2 + \|x_i\|^2)\theta_{ij,t} + 6\eta^2 n^2\theta_t\sigma_1^2 \tag{B.39}$$

$$\leq \left(8\eta\sigma_1 + 2\eta(2n\alpha^2 + 2r\sigma_1) + 6\eta^2 n^2\sigma_1^2\right)\theta_t, \tag{B.40}$$

where the last inequality uses the fact that

$$\sum_{k \neq i,j} \|x_k^t\|^2 \leq \sum_{k \leq r} \|x_k^t\|^2 + \sum_{k > r} \|x_k^t\|^2 \leq 2r\sigma_1 + 2n\alpha^2.$$

Hence, we choose $\eta \leq \frac{1}{1000n\sigma_1}$ to be sufficiently small so that $\max\{A, C\} \leq 1/100$, then by $\frac{1-A+B}{1-A-C} \leq 1 + 2B + 2C$ for $\max\{A, C\} \leq 1/100$,

$$\theta_{ij,t} \cdot \left(\frac{1-A+B}{1-A-C}\right)$$

$$\leq \theta_{ij,t}(1 + 2B + 2C)$$

$$\leq \theta_{ij,t} + 4\eta \sum_{k \neq i,j} \|x_k\|^2 \cdot \sqrt{\theta_{ik,t}\theta_{kj,t}\theta_{ij,t}} + 60n^2\sigma_1^2\eta^2\theta_t\sqrt{\theta_{ij,t}}$$

$$\quad + \theta_t^2\left(8\eta\sigma_1 + 2\eta(2n\alpha^2 + 2r\sigma_1) + 6\eta^2 n^2\sigma_1^2\right)$$

$$\leq \theta_{ij,t} + 4\eta(2r\sigma_1 + 2n\alpha^2)\theta_t^{3/2} + 60n^2\sigma_1^2\eta^2\theta_t^{3/2}$$

$$\quad + \theta_t^2\left(8\eta\sigma_1 + 2\eta(2n\alpha^2 + 2r\sigma_1) + 6\eta^2 n^2\sigma_1^2\right)$$

$$\leq \theta_{ij,t} + 6\eta(2r\sigma_1 + 2n\alpha^2)\theta_t^{3/2} + 60n^2\sigma_1^2\eta^2\theta_t^{3/2} + 8\eta\sigma_1\theta_t^2 + 6n^2\eta^2\sigma_1^2\theta_t^2)$$

$$\leq \theta_{ij,t} + 98\eta \cdot (r\sigma_1\theta_t^{3/2})$$

The last inequality holds by $\alpha \leq \sqrt{\sigma_1}/\sqrt{n}$, and $n^2\sigma_1\eta^2 \leq \eta$ because $\eta \leq \frac{1}{n^2\sigma_1}$.

Hence,

$$\theta_{t+1} \leq \theta_t + 98\eta(r\sigma_1)\theta_t^{3/2} \tag{B.41}$$

The Phase 1 terminates when $\|x_i^{T_1}\|^2 \geq \frac{3\sigma_i}{4}$. Since $\|x_i^0\|^2 \geq \alpha^2/2$ and

$$\|x_i^{t+1}\|^2 \geq (1 + \eta\sigma_i/4)\|x_i^t\|^2, \tag{B.42}$$

there is a constant $C_3$ such that $T_1 \leq C_1(\log(\sqrt{\sigma_1}/\alpha)/\eta\sigma_i)$. Hence, before round $T_1$,

$$\theta_{T_1} \leq \theta_0 + 98\eta T_1 \cdot r\sigma_1 \cdot (2\theta_0)^{3/2} \leq \theta_0 + 98C_1 r\kappa(2\theta_0)^{3/2}\log(\sqrt{\sigma_1}/\alpha) \leq 2\theta_0.$$

This is because

$$\theta_0 = \mathcal{O}((\log^2(r\sqrt{\sigma_1}/\alpha)(r\kappa))^2)$$

by Lemma B.1 and choosing $k \geq c_2((r\kappa)^2\log(r\sqrt{\sigma_1/\alpha}))^4$ for large enough $c_2$ $\qquad\square$

## B.3 PHASE 2

Denote $\theta_t^U = \max_{\min\{i,j\} \leq r} \theta_{ij,t}$. In this phase, we prove that $\theta_t^U$ is linear convergence, and the convergence rate of the loss is at least $\Omega(1/T^2)$. To be more specific, we will show that

$$\theta_{t+1}^U \leq \theta_t^U \cdot (1 - \eta \cdot \sigma_r/4) \leq \theta_t^U \tag{B.43}$$

$$\frac{\theta_{t+1}^U}{\sum_{i>r}\|x_i^{t+1}\|^2} \leq \frac{\theta_t^U}{\sum_{i>r}\|x_i^t\|^2} \cdot \left(1 - \frac{\eta\sigma_r}{8}\right) \tag{B.44}$$

$$|\|x_i^t\|^2 - \sigma_i| \leq \frac{1}{4}\sigma_i \ (i \leq r) \tag{B.45}$$

$$\|x_i^t\|^2 \leq 2\alpha^2 \ (i > r) \tag{B.46}$$

First, the condition (B.45) and (B.46) hold at round $T_1$. Then, if it holds before round $t$, consider round $t + 1$, similar to Phase 1, condition (B.46) also holds. Now we prove Eq.(B.43), (B.44) and (B.45) one by one.

**Proof of Eq.(B.45)** For $i \leq r$, if $\|x_i^t\|^2 \geq 3\sigma_i/4$, by Eq.(B.18)

$$|\|x_i^{t+1}\|_2^2 - \sigma_i| \leq (1 - \eta\sigma_i)|\|x_i^t\|^2 - \sigma_i| + 3\eta \sum_{j \neq i}^n ((x_i^t)^\top x_j^t)^2 \tag{B.47}$$

Hence, by (B.45) and (B.46), we can get

$$\sum_{j \neq i}^n ((x_i^t)^\top x_j^t)^2 \leq \sum_{j \neq i, j \leq r} ((x_i^t)^\top x_j^t)^2 + \sum_{j \neq i, j > r} ((x_i^t)^\top x_j^t)^2$$

$$\leq (r\sigma_1 + 4n\sigma_1\alpha^2)\theta_t^U$$

$$\leq 2r\sigma_1\theta_t^U \tag{B.48}$$

$$\leq 2r\sigma_1\theta_{T_1}^U \tag{B.49}$$

$$\leq 2r\sigma_1 \cdot 2\theta_0 \leq \sigma_i/20. \tag{B.50}$$

The inequality (B.48) is because $\alpha \leq \frac{1}{4n\sigma_1}$, the inequality (B.49) holds by induction hypothesis (B.43), and the last inequality (B.50) is because of (B.6) and $\theta_0 \leq \frac{1}{80r\kappa}$.

Hence, if $|\|x_i^t\|^2 - \sigma_i| \leq \sigma_i/4$, by combining (B.47) and (B.50), we have

$$|\|x_i^{t+1}\|^2 - \sigma_i| \leq (1 - \eta\sigma_i)|\|x_i^t\| - \sigma_i| + 3\eta\sigma_i/20 \leq \sigma_i/4.$$

Now it is easy to get that $|\|x_i^t\|^2 - \sigma_i| \leq 0.25\sigma_i$ for $t \geq T_1$ by induction because of $|\|x_i^{T_1}\|^2 - \sigma_i| \leq 0.25\sigma_i$. Thus, we complete the proof of Eq.(B.45).

**Proof of Eq.(B.43)** First, we consider $i \leq r, j \neq i \in [n]$ and $\theta_{ij,t} > \theta_t^U/2$, since (B.4) and (B.5) still holds with (B.45) and (B.46), similarly, we can still have equation (B.36), i.e.

$$\theta_{ij,t+1} = \theta_{ij,t} \cdot \left( \frac{1 - A - B}{1 - A - C} \right).$$

where

$$A = 2\eta(\|x_i^t\|^2 - \sigma_i) + 2\eta(\|x_j^t\|^2 - \sigma_j) \geq -2\eta(2 \cdot (\sigma_i/4)) \geq -1/100.$$

$$B = 2\eta(\|x_i^t\|^2 + \|x_j^t\|^2) - 2\eta \sum_{k \neq i,j} \|x_k\|^2 \cdot \sqrt{\theta_{ik,t}\theta_{kj,t}/\theta_{ij,t}} - 30n^2\eta^2\sigma_1^2\sqrt{\theta_t^U}/\sqrt{\theta_{ij,t}}$$

$$\geq 2\eta(\|x_i^t\|^2 + \|x_j^t\|^2) - 4\eta \sum_{k \leq r} \|x_k\|^2\sqrt{\theta^U} - 4n\eta\alpha^2 - 40n^2\eta^2\sigma_1^2 \tag{B.51}$$

$$\geq 2\eta \cdot \frac{3\sigma_i}{4} - 8\eta r\sigma_1\sqrt{2\theta_{T_0}} - 4n\eta\alpha^2 - 40n^2\eta^2\sigma_1^2 \tag{B.52}$$

$$\geq \eta \cdot \sigma_r \tag{B.53}$$

The inequality Eq.(B.51) holds by $\theta_{ij,t} > \theta_t^U/2$, the inequality (B.52) holds by (B.43), and (B.53) holds by

$$\theta_{T_0} = \mathcal{O}\left( \frac{1}{r^2\kappa^2} \right), \quad \alpha = \mathcal{O}(\sqrt{\sigma_r/n}), \quad \eta = \mathcal{O}(1/n^2\kappa\sigma_1). \tag{B.54}$$

The term $C$ is defined and can be bounded by

$$C = 2\eta \sum_{k \neq i,j} \|x_k\|^2(\theta_{ik,t} + \theta_{jk,t}) + 2\eta(\|x_i\|^2 + \|x_j\|^2)\theta_{ij,t} + 6\eta^2\theta_t n^2\sigma_1^2$$

$$\leq 4\eta \sum_{k \leq r} \|x_k\|^2\theta_t^U + 4\eta n\alpha^2\theta_t + 6\eta^2\theta_t n^2\sigma_1^2$$

$$\leq 8r\eta\sigma_1\theta_t^U + 4\eta n\alpha^2 + 6\eta^2 n^2\sigma_1^2$$

$$\leq 8r\eta\sigma_1\theta_{T_0} + 4\eta n\alpha^2 + 6\eta^2 n^2\sigma_1^2 \tag{B.55}$$

$$\leq \eta \cdot \sigma_r/2. \tag{B.56}$$

The inequality (B.55) holds by (B.43), and the inequality (B.56) holds by (B.54).

Then, for $i \leq r, j \neq i \in [n]$ and $\theta_{ij,t} > \theta_t^U/2$, we can get

$$
\begin{aligned}
\theta_{ij,t+1} &\leq \theta_{ij,t} \cdot \left( \frac{1 - A - B}{1 - A - C} \right) \\
&\leq \theta_{ij,t} \cdot \left( \frac{2 - \eta \cdot \sigma_r}{2 - \eta \cdot \sigma_r/2} \right) \\
&\leq \theta_{ij,t} \cdot \left( \frac{1 - \eta \cdot \sigma_r/2}{1 - \eta \cdot \sigma_r/4} \right) \leq \theta_{ij,t} \cdot (1 - \eta \cdot \sigma_r/4)
\end{aligned}
\tag{B.57}
$$

For $i \leq r, j \in [n]$ and $\theta_{ij,t} \leq \theta_t^U/2$, we have

$$
B \geq -2\eta \sum_{k \leq r} \|x_k\|^2 \theta_t^U / \sqrt{\theta_{ij,t}} - 2\eta \sum_{k > r} \|x_k\|^2 \sqrt{\theta_t^U}/\sqrt{\theta_{ij,t}} - 30n^2\eta^2\sigma_1^2 \sqrt{\theta_t^U}/\sqrt{\theta_{ij,t}}
\tag{B.58}
$$

$$
\geq -4\eta r \sigma_1 \theta_t^U / \sqrt{\theta_{ij,t}} - (4n\eta\alpha^2 + 30n^2\eta^2\sigma_1^2)\sqrt{\theta_t^U}/\sqrt{\theta_{ij,t}}
\tag{B.59}
$$

$$
\begin{aligned}
\theta_{ij,t+1} &\leq \theta_{ij,t} \cdot \left( \frac{1 - A - B}{1 - A - C} \right) \\
&\leq \theta_{ij,t} \cdot (1 - 2B + 2C) \\
&\leq \theta_{ij,t} + 8\eta r \sigma_1 \theta_t^U \sqrt{\theta_{ij,t}} + (4n\eta\alpha^2 + 30n^2\eta^2\sigma_1^2)\sqrt{\theta_t^U \theta_{ij,t}} + 2C\theta_{ij,t} \\
&\leq \frac{\theta_t^U}{2} + 8\eta r \sigma_1 \theta_t^U + (4n\eta\alpha^2 + 30n^2\eta^2\sigma_1^2)\theta_t^U + \eta\sigma_r \theta_t^U \\
&\leq \frac{3\theta_t^U}{4}.
\end{aligned}
\tag{B.60}
$$

The last inequality is because $8\eta r \sigma_1 + 4n\eta\alpha^2 + 30n^2\eta^2\sigma_1^2 + \eta\sigma_r \leq \frac{1}{4}$ by $\eta \leq \mathcal{O}(1/n\sigma_1)$ and $\eta \leq \mathcal{O}(1/n\alpha^2)$. Hence, by Eq.(B.57) and (B.60) and the fact that $\eta\sigma_r/4 \leq 1/4$,

$$
\theta_{t+1}^U \leq \theta_t^U \cdot \max\left\{ \frac{3}{4}, 1 - \eta \cdot \sigma_r/4 \right\} = (1 - \eta \cdot \sigma_r/4)\theta_t^U.
\tag{B.61}
$$

Thus, we complete the proof of Eq.(B.43)

**Proof of Eq.(B.44)** Also, for $i > r$, denote $\theta_{ii,t} = 1$, then

$$
\begin{aligned}
\|x_i^{t+1}\|^2 &= \|x_i^t\|^2 - 2\eta \sum_{j=1}^n ((x_i^t)^\top x_j^t)^2 + \eta^2 \left( \sum_{j,k=1}^n (x_i^t)^\top x_j^t (x_j^t)^\top \right)^2 \\
&\geq \|x_i^t\|^2 (1 - 2\eta \sum_{j=1}^n \|x_j^t\|^2 \theta_{ij,t}) \\
&\geq \|x_i\|^2 (1 - 2\eta r \sigma_1 \theta_t^U - 2\eta n \alpha^2) \\
&\geq \|x_i\|^2 (1 - \eta \cdot \sigma_r/8)
\end{aligned}
\tag{B.62}
$$

The last inequality holds because

$$
\theta_t^U \leq \theta_0 \leq \mathcal{O}(1/r\kappa)
\tag{B.63}
$$

$$
\alpha \leq \sqrt{\sigma_r/n}
\tag{B.64}
$$

Hence, the term $\theta^U/\|x_i\|^2$ for $i > r$ is also linear convergence by

$$
\frac{\theta_{t+1}^U}{\sum_{i>r} \|x_i^{t+1}\|^2} \leq \frac{\theta_t^U}{\sum_{i>r} \|x_i^t\|^2} \cdot \frac{1 - \eta \cdot \sigma_r/4}{1 - \eta \cdot \sigma_r/8} \leq \frac{\theta_t^U}{\sum_{i>r} \|x_i^t\|^2} \cdot \left( 1 - \frac{\eta\sigma_r}{8} \right).
$$

Hence, we complete the proof of Eq.(B.44).

## B.4 PHASE 3: LOWER BOUND OF CONVERGENCE RATE

Now by (B.44), there are constants $c_6$ and $c_7$ such that, if we denote $T_2 = T_1 + c_7(\log(\sqrt{r\sigma_1}/\alpha)/\eta\sigma_r) = c_6(\log(\sqrt{r\sigma_1}/\alpha)/\eta\sigma_r)$, then we will have

$$\theta_{T_2}^U < \sum_{i>r} \|x_i^{T_2}\|^2/r\sigma_1 \tag{B.65}$$

because of the fact that $\theta_{T_1}^U / \sum_{i>r} \|x_i^{T_1}\|^2 \leq \frac{4}{n\cdot\alpha^2} \leq 4/\alpha^2$. Now after round $T_2$, consider $i > r$, we can have

$$\|x_i^{t+1}\|^2 \geq \|x_i^t\|^2(1 - 2\eta\sum_{j=1}^n \|x_j^t\|^2\theta_{ij,t})$$

$$\geq \|x_i^t\|^2(1 - 2\eta r\sigma_1\theta_t^U - 2\eta\sum_{j>r}\|x_j^t\|^2)$$

Hence, by Eq.(B.62), we have

$$\sum_{j>r}\|x_j^{t+1}\|^2 \geq \left(\sum_{j>r}\|x_j^t\|^2\right)\left(1 - 2\eta r\sigma_1\theta_t^U - 2\eta\sum_{j>r}\|x_j^t\|^2\right) \tag{B.66}$$

$$\geq \left(\sum_{j>r}\|x_j^t\|^2\right)\left(1 - 4\eta\sum_{j>r}\|x_j^t\|^2\right), \tag{B.67}$$

where the second inequality is derived from (B.65).

Hence, we can show that $\sum_{j>r}\|x_j^t\|^2 = \Omega(1/T^2)$. In fact, suppose at round $T_2$, we denote $A_{T_2} = \sum_{j>r}\|x_j^{T_2}\|^2$, then by

$$\|x_i^{t+1}\|^2 \geq \|x_i^t\|^2(1 - 2\eta\sum_{k=1}^n \|x_k^t\|^2\theta_{ik,t}))$$

$$\geq \|x_i^t\|^2(1 - 2\eta r\sigma_1\theta^U - 2\eta n\alpha^2)$$

we can get

$$\begin{aligned}\|x_i^{T_2}\|^2 &\geq \|x_i^{T_1}\|^2(1 - 2\eta r\sigma_1\theta_{T_1}^U - 2\eta n\alpha^2)^{T_2-T_1}\\ &\geq \|x_i^{T_1}\|^2 \cdot (1 - c_5(\log(r\sqrt{\sigma_1}/\alpha)/\eta\sigma_r) \cdot (2\eta r\sigma_1\theta_{T_1} + 2\eta n\alpha^2))\\ &\geq \|x_i^{T_1}\|^2 \cdot (1 - c_5\log(r\sqrt{\sigma_1}/\alpha) \cdot (4r\kappa\theta_0 + 2n\alpha^2/\sigma_r))\\ &\geq \frac{1}{2}\|x_i^{T_1}\|^2 \tag{B.68}\\ &\geq \frac{\alpha^2}{8}\end{aligned}$$

where the inequality (B.68) is because

$$\theta_0 \leq \mathcal{O}\left(\frac{1}{r\kappa\log(r\sqrt{\sigma_1}/\alpha)}\right) \tag{B.69}$$

$$\alpha^2 \leq \mathcal{O}\left(\frac{\sqrt{\sigma_r}}{n\log(r\sqrt{\sigma_1}/\alpha)}\right). \tag{B.70}$$

Hence,

$$T_2 A_{T_2} \geq T_2 \cdot (n-r)\frac{\alpha^2}{8} \geq c_7(\log(\sqrt{r\sigma_1}/\alpha)/\eta\sigma_r) \cdot \frac{\alpha^2}{8}. \tag{B.71}$$

by $n > r$. Define $A_{T_2+i+1} = A_{T_2+i}(1 - 4\eta A_{T_2+i})$, by Eq.(B.67), we have

$$A_{T_2+i} \leq A_{T_2} = \sum_{i>r}\|x_i^{T_2}\|^2 \leq 2n\alpha^2. \tag{B.72}$$

On the other hand, if $\eta(T_2 + i)A_{T_2+i} \leq 1/8$, and then

$$
\begin{aligned}
\eta(T_2 + i + 1)A_{T_2+i+1} &= \eta(T_2 + i + 1)A_{T_2+i}(1 - 4\eta A_{T_2+i}) \\
&= \eta(T_2 + i)A_{T_2+i} - (T_2 + i)4\eta^2 A_{T_2+i}^2 + \eta A_{T_2+i}(1 - 4\eta A_{T_2+i}) \\
&\geq \eta(T_2 + i)A_{T_2+i} - (T_2 + i)4\eta^2 A_{T_2+i}^2 + \eta A_{T_2+i}/2 \qquad \text{(B.73)} \\
&\geq \eta(T_2 + i)A_{T_2+i} - \eta A_{T_2+i}/2 + \eta A_{T_2+i}/2 \\
&\geq \eta(T_2 + i)A_{T_2+i},
\end{aligned}
$$

where (B.73) holds by $\eta A_{T_2+i} \leq 2n\eta\alpha^2 \leq 1/8$.

If $\eta(T_2 + i)A_{T_2+i} > 1/8$, since $\eta A_{T_2+i} \leq 1/8$, we have $\eta A_{T_2} \leq 2n\eta\alpha^2 \leq 1/8$.

$$
\begin{aligned}
\eta(T_2 + i + 1)A_{T_2+i+1} &\geq \eta(T_2 + i)A_{T_2+i}(1 - 4\eta A_{T_2+i}) + \eta A_{T_2+i}(1 - 4\eta A_{T_2+i}) \\
&\geq \frac{1}{8} \cdot \frac{1}{2} + \eta A_{T_2+i} \cdot \frac{1}{2} \\
&\geq \frac{1}{16}.
\end{aligned}
$$

Thus, by the two inequalities above, at round $t \geq T_2$, we can have

$$
\eta t A_t \geq \min\{\eta T_2 A_{T_2}, 1/16\}.
$$

Now by (B.71),

$$
\eta T_2 A_{T_2} \geq \frac{c_7 \log(\sqrt{r\sigma_1}/\alpha)\alpha^2}{8\sigma_r}, \qquad \text{(B.74)}
$$

then for any $t \geq T_2$, we have

$$
\eta t A_t \geq \min\left\{ \frac{c_7 \log(\sqrt{r\sigma_1}/\alpha)\alpha^2}{8\sigma_r}, 1/16 \right\} \qquad \text{(B.75)}
$$

Now by choosing $\alpha = \widetilde{\mathcal{O}}(\sqrt{\sigma_r})$ so that $\frac{c_7 \log(\sqrt{r\sigma_1}/\alpha)\alpha^2}{8\sigma_r} \leq 1/16$, we can derive

$$
A_t \geq \frac{c_7 \log(\sqrt{r\sigma_1}/\alpha)\alpha^2}{8\sigma_r \eta t}. \qquad \text{(B.76)}
$$

Since for $j > r$, $(X_t X_t^\top - \Sigma)_{jj} = \|x_j^t\|^2$, we have $\|X_t X_t^\top - \Sigma\|^2 \geq \sum_{j>r} \|x_j^t\|^4 \geq A_t^2/n$ and

$$
\|X_t X_t^\top - \Sigma\|^2 \geq A_t^2/n \geq \left( \frac{c_7 \log(\sqrt{r\sigma_1}/\alpha)\alpha^2}{8\sigma_r \eta \sqrt{n} t} \right)^2.
$$

## C  PROOF OF THEOREM 4.1

Denote the matrix of the first $r$ row of $F, G$ as $U, V$ respectively, and the matrix of the last $n - r$ row of $F, G$ as $J, K$ respectively. Hence, $U, V \in \mathbb{R}^{r \times k}, J, K \in \mathbb{R}^{(n-r) \times k}$. In this case, the difference $F_t G_t^\top - \Sigma$ can be written in a block form as

$$
F_t G_t^\top - \Sigma = \begin{pmatrix} U_t V_t^\top - \Sigma_r & J_t V_t^\top \\ U_t K_t^\top & J_t K_t^\top \end{pmatrix}, \qquad \text{(C.1)}
$$

where $\Sigma_r = I \in \mathbb{R}^{r \times r}$. Hence, the loss can be bounded by

$$
\|J_t K_t^\top\| \leq \|F_t G_t^\top - \Sigma\| \leq \|U_t V_t^\top - \Sigma_r\| + \|J_t V_t^\top\| + \|U_t K_t^\top\| + \|J_t K_t^\top\|. \qquad \text{(C.2)}
$$

The updating rule for $(U, V, J, K)$ under gradient descent in (4.2) can be rewritten explicitly as

$$
\begin{aligned}
U_{t+1} &= U_t + \eta \Sigma_r V_t - \eta U_t(V_t^\top V_t + K_t^\top K_t) \\
V_{t+1} &= V_t + \eta \Sigma_r U_t - \eta V_t(U_t^\top U_t + J_t^\top J_t) \\
J_{t+1} &= J_t - \eta J_t(V_t^\top V_t + K_t^\top K_t) \\
K_{t+1} &= K_t - \eta K_t(U_t^\top U_t + J_t^\top J_t).
\end{aligned}
$$

Note that with our particular initialization, we have the following equality for all $t$:

$$U_t K_t^\top = 0, J_t V_t^\top = 0, \quad \text{and} \quad U_t = V_t. \tag{C.3}$$

Indeed, the conditions (C.3) are satisfied for $t = 0$. For $t + 1$, we have

$$U_{t+1} = U_t + \eta(\Sigma_r - U_t V_t^\top)V_t = V_t + \eta(\Sigma_r - U_t V_t^\top)U_t = V_{t+1}, \ \ K_{t+1} = K_t - \eta K_t J_t^\top J_t$$

$$U_{t+1} K_{t+1}^\top = U_t K_t^\top + \eta(\Sigma_r - U_t V_t^\top)U_t K_t^\top - \eta V_t J_t^\top J_t K_t^\top - \eta^2(\Sigma_r - U_t V_t^\top)U_t J_t^\top J_t K_t^\top = 0$$

The last equality arises from the fact that $U_t K_t^\top = 0$, $J_t V_t^\top = 0$ and $U_t = V_t$. Similarly, we can get $J_{t+1} V_{t+1}^\top = 0$. Hence, we can rewrite the updating rule of $J_t$ and $K_t$ as

$$J_{t+1} = J_t - \eta J_t K_t^\top K_t \tag{C.4}$$

$$K_{t+1} = K_t - \eta K_t J_t^\top J_t. \tag{C.5}$$

Let us now argue why the convergence rate can not be faster than $\Omega((1 - 6\eta\alpha^2)^t)$. Denote $A \in \mathbb{R}^{(n-r)\times k}$ as the matrix that $(A)_{1k} = 1$ and other elements are all zero. We have that $J_0 = \alpha A$ and $K_0 = (\alpha/3) \cdot A$. Combining this with Eq.(C.4) and Eq.(C.5), we have $J_t = a_t A$, $K_t = b_t A$, where

$$a_0 = \alpha, b_0 = \alpha/3, \tag{C.6a}$$

$$a_{t+1} = a_t - \eta a_t b_t^2, \tag{C.6b}$$

$$b_{t+1} = b_t - \eta a_t^2 b_t. \tag{C.6c}$$

It is immediate that $0 \le a_{t+1} \le a_t, 0 \le b_{t+1} \le b_t$, $\max\{a_t, b_t\} \le \alpha$ because of $\eta b_t^2 \le \eta b_0^2 = \eta\alpha^2 \le 1$ and similarly $\eta a_t^2 \le 1$. Now by $\eta\alpha^2 \le 1/4$,

$$\|J_{t+1} K_{t+1}^\top\| = a_{t+1} b_{t+1} = (1 - \eta a_t^2)(1 - \eta b_t^2)a_t b_t \ge (1 - 2\eta\alpha^2)^2 a_t b_t \ge (1 - 4\eta\alpha^2)a_t b_t. \tag{C.7}$$

By Eq.(C.2) that $\|F_t G_t^\top - \Sigma\| \ge \|J_t K_t^\top\|$, the convergence rate of $\|F_t G_t^\top - \Sigma\|$ can not be faster than $a_0 b_0 (1 - 4\eta\alpha^2)^t \ge \frac{\alpha^2}{3}(1 - 4\eta\alpha^2)^t$.

Next, we show why the convergence rate is exactly $\Theta((1 - \Theta(\eta\alpha^2))^t)$ in this toy case. By Eq.(C.3), the loss $\|F_t G_t^\top - \Sigma\| \le \|U_t U_t^\top - \Sigma_r\| + \|J_t K_t^\top\|$. First, we consider the norm $\|U_t U_t^\top - \Sigma_r\|$. Since in this toy case, $\Sigma_r = I_r$ and $U_t = V_t$ for all $t$, the updating rule of $U_t$ can be written as

$$U_{t+1} = U_t - \eta(U_t U_t^\top - I)U_t \tag{C.8}$$

Note that $U_0 = (\alpha I_r, 0) \in \mathbb{R}^{r\times k}$. By induction, we can show that $U_t = (\alpha_t I_r, 0)$ and $\alpha_{t+1} = \alpha_t - \eta(\alpha_t^2 - 1)\alpha_t$ for all $t \ge 0$. If $\alpha_t \le 1/2$, we have

$$\alpha_{t+1} = \alpha_t(1 + \eta - \eta\alpha_t^2) \ge \alpha_t(1 + \eta/2).$$

Then, there exists a constant $c_1$ and $T_1 = c_1(\log(1/\alpha)/\eta)$ such that after $T_1$ rounds, we can get $\alpha_t \ge 1/2$. By the fact that $\alpha_{t+1} = \alpha_t(1 + \eta(1 - \alpha_t^2)) \le \max\{\alpha_t, 2\}$ when $\eta < 1$, it is easy to show $\alpha_t \le 2$ for all $t \ge 0$. Thus, when $\eta < 1/6$, we can get $1 - \eta(\alpha_t + 1)\alpha_t > 0$ and then

$$|\alpha_{t+1} - 1| = |(\alpha_t - 1) - \eta(\alpha_t - 1)(\alpha_t + 1)\alpha_t|$$
$$= |\alpha_t - 1|(1 - \eta(\alpha_t + 1)\alpha_t)$$
$$\le |\alpha_t - 1|(1 - \eta/2).$$

we know that $\|U_t U_t^\top - \Sigma_r\| = \alpha_t^2 - 1$ converges at a linear rate

$$\|U_t U_t^\top - \Sigma\| \le (1 - \eta/2)^{t-T_1} \overset{(a)}{\le} (1 - \eta\alpha^2/4)^{(t-T_1)/2}, \tag{C.9}$$

where (a) uses the fact that

$$1 - \eta\alpha^2/4 \ge 1 - \eta \ge (1 - \eta/2)^2 \tag{C.10}$$

Hence, we only need to show that $\|J_t K_t^\top\|$ converges at a relatively slower speed $\mathcal{O}((1 - \Theta(\eta\alpha^2))^t)$. To do this, we prove the following statements by induction.

$$\alpha \ge a_t \ge \alpha/2, \ \ b_{t+1}^2 \le b_t^2(1 - \eta\alpha^2/4) \tag{C.11}$$

Using $b_0 = \alpha/3$, we see the above implies that $\|J_t K_t^\top\| = a_t b_t \leq \mathcal{O}((1 - \Theta(\eta\alpha^2))^t)$.

Let us prove (C.11) via induction. It is trivial to show it holds at $t = 0$ and the upper bound of $a_t$ by (C.6). Suppose (C.11) holds for $t' \leq t$, then at round $t + 1$, we have

$$b_{t+1}^2 = b_t^2(1 - \eta a_t^2)^2 \leq b_t^2(1 - \eta\alpha^2/4)^2 \leq b_t^2(1 - \eta\alpha^2/4). \tag{C.12}$$

Using $a_{t+1} = a_t(1 - \eta b_t^2)$, we have

$$a_{t+1} = a_0 \prod_{i=1}^t (1 - \eta b_i^2) \overset{(a)}{\geq} a_0 \left(1 - \eta \sum_{i=1}^t b_i^2\right) \overset{(b)}{\geq} \alpha \cdot \left(1 - \eta \cdot \frac{\alpha^2}{9} \cdot \frac{4}{\eta\alpha^2}\right) \geq \alpha/2. \tag{C.13}$$

where the step $(a)$ holds by recursively using $(1 - a)(1 - b) \geq (1 - (a + b))$ for $a, b \in (0, 1)$, and the step $(b)$ is due to $b_i^2 \leq b_0^2 \cdot (1 - \eta\alpha^2/4)^t \leq \frac{\alpha^2}{9} \cdot (1 - \frac{\eta\alpha^2}{4})^t$ and the sum formula for geometric series. Thus, the induction is complete, and

$$\|J_t K_t^\top\| = a_t b_t \leq (\alpha^2/3) \cdot (1 - \eta\alpha^2/4)^{t/2} \leq (1 - \eta\alpha^2/4)^{t/2} \leq (1 - \eta\alpha^2/4)^{(t-T_1)/2}. \tag{C.14}$$

Combining (C.9) and (C.14), with $\|A\|_2 \leq \|A\|_F \leq \text{rank}(A) \cdot \|A\|_2$, we complete the proof.

## D    PROOF OF THEOREM 4.2

We prove Theorem 4.2 in this section. We start with some preliminaries.

### D.1    PRELIMINARIES

In the following, we denote $\delta_{2k+1} = \sqrt{2k+1}\delta$. Also denote the matrix of the first $r$ row of $F, G$ as $U, V$ respectively, and the matrix of the last $n - r$ row of $F, G$ as $J, K$ respectively. Hence, $U, V \in \mathbb{R}^{r \times k}, J, K \in \mathbb{R}^{(n-r) \times k}$. We denote the corresponding iterates as $U_t, V_t, J_t$, and $K_t$.

Also, define $E(X) = \mathcal{A}^*\mathcal{A}(X) - X$. We also denote $\Gamma(X) = \mathcal{A}^*\mathcal{A}(X)$. By Lemma G.2, we can show that $\|E(X)\| \leq \delta_{2k+1} \cdot \|X\|$ for matrix $X$ with rank less than $2k$ by Lemma G.2. Decompose the error matrix $E(X)$ into four submatrices by

$$E(X) = \begin{pmatrix} E_1(X) & E_2(X) \\ E_3(X) & E_4(X) \end{pmatrix},$$

where $E_1(X) \in \mathbb{R}^{r \times r}, E_2(X) \in \mathbb{R}^{r \times (n-r)}, E_3(X) \in \mathbb{R}^{(n-r) \times r}, E_4(X) \in \mathbb{R}^{(n-r) \times (n-r)}$. Then the updating rule can be rewritten in this form:

$$U_{t+1} = U_t + \eta\Sigma V_t - \eta U_t(V_t^\top V_t + K_t^\top K_t) + \eta E_1(F_t G_t^\top - \Sigma)V_t + \eta E_2(F_t G_t^\top - \Sigma)K_t \tag{D.1}$$

$$V_{t+1} = V_t + \eta\Sigma U_t - \eta V_t(U_t^\top U_t + J_t^\top J_t) + \eta E_1^\top(F_t G_t^\top - \Sigma)U_t + \eta E_3^\top(F_t G_t^\top - \Sigma)J_t \tag{D.2}$$

$$J_{t+1} = J_t - \eta J_t(V_t^\top V_t + K_t^\top K_t) + \eta E_3(F_t G_t^\top - \Sigma)V_t + \eta E_4(F_t G_t^\top - \Sigma)K_t \tag{D.3}$$

$$K_{t+1} = K_t - \eta K_t(U_t^\top U_t + J_t^\top J_t) + \eta E_2^\top(F_t G_t^\top - \Sigma)U_t + \eta E_4^\top(F_t G_t^\top - \Sigma)J_t. \tag{D.4}$$

Since the submatrices' operator norm is less than the operator norm of the whole matrix, the matrices $E_i(F_t G_t^\top - \Sigma), i = 1, \ldots, 4$ satisfy that

$$\|E_i(F_t G_t^\top - \Sigma)\| \leq \|E(F_t G_t^\top - \Sigma)\| \leq \delta_{2k+1}\|F_t G_t^\top - \Sigma\|, \quad i = 1, \ldots, 4.$$

**Imbalance term**    An important property in analyzing the asymmetric matrix sensing problem is that $F^\top F - G^\top G = U^\top U + J^\top J - V^\top V - K^\top K$ remains almost unchanged when step size $\eta$ is sufficiently small, i.e., the balance between two factors $F$ and $G$ are does not change much throughout the process. To be more specific, by

$$F_{t+1} = F_t - \eta(F_t G_t^\top - \Sigma)G_t - E(F_t G_t^\top - \Sigma)G_t$$

$$G_{t+1} = G_t - \eta(F_t G_t^\top - \Sigma)^\top F_t - (E(F_t G_t^\top - \Sigma))^\top F_t$$

we have

$$\left\|\left(F_{t+1}^\top F_{t+1} - G_{t+1}^\top G_{t+1}\right) - \left(F_t^\top F_t - G_t^\top G_t\right)\right\| \leq 2\eta^2 \cdot \|F_t G_t^\top - \Sigma\|^2 \cdot \max\{\|F_t\|, \|G_t\|\}^2. \tag{D.5}$$

In fact, by the updating rule, we have

$$F_{t+1}^\top F_{t+1} - G_{t+1}^\top G_{t+1}$$
$$= F_t^\top F_t - G_t^\top G_t + \eta^2 \Big( G_t^\top (F_t G_t^\top - \Sigma)^\top (F_t G_t^\top - \Sigma) G_t - F_t^\top (F_t G_t^\top - \Sigma)(F_t G_t^\top - \Sigma)^\top F_t \Big),$$

so that

$$\|F_{t+1}^\top F_{t+1} - G_{t+1}^\top G_{t+1} - (F_t^\top F_t - G_t^\top G_t)\| \leq 2\eta^2 \|F_t\|^2 \|G_t\|^2 \|F_t G_t^\top - \Sigma\|^2$$
$$\leq 2\eta^2 \cdot \|F_t G_t^\top - \Sigma\| \cdot \max\{\|F_t\|^2, \|G_t\|^2\}$$

Thus, we will prove that, during the proof process, the following inequality holds with high probability during all $t \geq 0$:

$$2\alpha^2 I \geq U_t^\top U_t + J_t^\top J_t - V_t^\top V_t - K_t^\top K_t \geq \frac{\alpha^2}{8} I. \tag{D.6}$$

Next, we give the outline of our proof.

## D.2 Proof Outline

In this subsection, we give our proof outline.

• Recall $\Delta_t = F_t^\top F_t - G_t^\top G_t = U_t^\top U_t + J_t^\top J_t - V_t^\top V_t - K_t^\top K_t$. In Section D.3, we show that with high probability, $\Delta_0$ has the scale $\alpha$, i.e., $C\alpha^2 I \geq \Delta_0 \geq c\alpha^2 I$, where $C > c$ are two constants. Then, we apply the converge results in Soltanolkotabi et al. (2023) to argue that the algorithm first converges to a local point. By Soltanolkotabi et al. (2023), this converge phase takes at most $T_0 = \mathcal{O}((1/\eta\sigma_r v)\log(\sqrt{\sigma_1}/n\alpha))$ rounds.

• Then, in Section D.4 (Phase 1), we mainly show that $M_t = \max\{\|U_t V_t^\top - \Sigma\|, \|U_t K_t^\top\|, \|J_t V_t^\top\|\}$ converges linearly until it is smaller than

$$M_t \leq \mathcal{O}(\sigma_1 \delta + \alpha^2)\|J_t K_t^\top\|. \tag{D.7}$$

This implies that the difference between estimated matrix $U_t V_t^\top$ and true matrix $\Sigma$, $\|U_t V_t^\top - \Sigma\|$, will be dominated by $\|J_t K_t^\top\|$. Moreover, during Phase 1 we can also show that $\Delta_t$ has the scale $\alpha$. Phase 1 begins at $T_0$ rounds and terminates at $T_1$ rounds, and $T_1$ may tend to infinity, which implies that Phase 1 may not terminate. In this case, since $M_t$ converges linearly and $M_t > \Omega(\sigma_1 \delta + \alpha^2)\|J_t K_t^\top\|$, the loss also converges linearly. Note that, in the exact-parameterized case, i.e., $k = r$, we can prove that Phase 1 will not terminate since the stopping rule (D.7) is never satisfied as shown in Section E.

• The Section D.5 (Phase 2) mainly shows that, after Phase 1, the $\|U_t - V_t\|$ converges linearly until it achieves

$$\|U_t - V_t\| \leq \mathcal{O}(\alpha^2/\sqrt{\sigma_1}) + \mathcal{O}(\delta_{2k+1}\|J_t K_t^\top\|/\sqrt{\sigma_1}).$$

Assume Phase 2 starts at round $T_1$ and terminates at round $T_2$. Then since we can prove that $\|U_t - V_t\|$ decreases from [4] $\mathcal{O}(\sigma_1)$ to $\Omega(\alpha^2)$, Phase 2 only takes a relatively small number of rounds, i.e. at most $T_2 - T_1 = \mathcal{O}(\log(\sqrt{\sigma_r}/\alpha)/\eta\sigma_r)$ rounds. We also show that $M_t$ remains small in this phase.

• The Section D.6 (Phase 3) finally shows that the norm of $K_t$ converges linearly, with a rate dependent on the initialization scale. As in Section 4.2, the error matrix in matrix sensing brings additional challenges for the proof. We overcome this proof by further analyzing the convergence of (a) part of $K_t$ that aligns with $U_t$, and (b) part of $K_t$ that lies in the complement space of $U_t$. We also utilize that $M_t$ and $\|U_t - V_t\|$ are small from the start of the phase and remain small. See Section D.6 for a detailed proof.

---

[4] The upper bound $\mathcal{O}(\sigma_1)$ of $\|U_t - V_t\|$ is proved in the first two phases.

### D.3 INITIAL ITERATIONS

We start our proof by first applying results in Soltanolkotabi et al. (2023) and provide some additional proofs for our future use. From Soltanolkotabi et al. (2023), the converge takes at most $T_0 = \mathcal{O}((1/\eta\sigma_r\upsilon)\log(\sqrt{\sigma_1}/n\alpha))$ rounds.

Let us state a few properties of the initial iterations using Lemma G.3.

**Initialization** By our imbalance initialization $F_0 = \alpha \cdot \widetilde{F}_0, G_0 = (\alpha/3) \cdot \widetilde{G}_0$, and by random matrix theory about the singular value (Vershynin, 2018, Corollary 7.3.3 and 7.3.4), with probability at least $1 - 2\exp(-cn)$ for some constant $c$, if $n > 8k$, we can show that $[\sigma_{\min}(F_0), \sigma_{\max}(F_0))] \subseteq [\frac{\sqrt{3}\alpha}{2}, \frac{\sqrt{3}\alpha}{\sqrt{2}}], [\sigma_{\min}(G_0), \sigma_{\max}(G_0)] \subseteq [\frac{\sqrt{3}\alpha}{6}, \frac{\alpha}{\sqrt{6}}]$ and

$$\frac{3\alpha^2}{2}I \geq F_0^\top F_0 - G_0^\top G_0 = U_0^\top U_0 + J_0^\top J_0 - V_0^\top V_0 - K_0^\top K_0 \geq \frac{\alpha^2}{2}I \tag{D.8}$$

As we will show later, we will prove the (D.6) during all phases by (D.5) and (D.8).

First, we show the following lemma, which is a subsequent corollary of the Lemma G.3.

**Lemma D.1.** *There exist parameters $\zeta_0$, $\delta_0, \alpha_0, \eta_0$ such that, if we choose $\alpha \leq \alpha_0$, $F_0 = \alpha \cdot \tilde{F}_0, G_0 = (\alpha/2) \cdot \tilde{G}_0$, where the elements of $\tilde{F}_0, \tilde{G}_0$ is $\mathcal{N}(0,1)$,[5] and suppose that the operator $\mathcal{A}$ defined in Eq.(1.1) satisfies the restricted isometry property of order $2r + 1$ with constant $\delta \leq \delta_0$, then the gradient descent with step size $\eta \leq \eta_0$ will achieve*

$$\|F_t G_t^\top - \Sigma\| \leq \min\{\sigma_r/2, \alpha^{1/2} \cdot \sigma_1^{3/4}\} \tag{D.9}$$

*within $T_0 = c_2(1/\eta\sigma_r)\log(\sqrt{\sigma_1}/n\alpha)$ rounds with probability at least $1 - \zeta_0$ and constant $c_2 \geq 1$, where $\zeta_0 = c_1 \exp(-c_2 k) + \exp(-(k - r + 1))$ is a small constant. Moreover, during $t \leq T_0$ rounds, we always have*

$$\max\{\|F_t\|, \|G_t\|\} \leq 2\sqrt{\sigma_1} \tag{D.10}$$

$$\|U_t - V_t\| \leq 4\alpha + \frac{40\delta_{2k+1}\sigma_1^{3/2}}{\sigma_r} \tag{D.11}$$

$$\|J_t\| \leq \mathcal{O}\Big(2\alpha + \frac{\delta_{2k+1}\sigma_1^{3/2}\log(\sqrt{\sigma_1}/n\alpha)}{\sigma_r}\Big) \tag{D.12}$$

$$\frac{13\alpha^2}{8}I \geq \Delta_t \geq \frac{3\alpha^2}{8}I \tag{D.13}$$

*Proof.* Since the initialization scale $\alpha \leq \mathcal{O}(\sqrt{\sigma_1})$, Eq.(D.10), Eq.(D.11), Eq.(D.12) and Eq.(D.13) hold for $t' = 0$. Assume that Eq.(D.9), Eq.(D.10), Eq.(D.11), Eq.(D.12) and Eq.(D.13) hold for $t' = t - 1$.
**Proof of Eq.(D.9) and Eq.(D.10)**

First, by using the previous global convergence result Lemma G.3, the Eq.(D.9) holds by $\alpha^{3/5}\sigma_1^{7/10} < \sigma_r/2$ because $\alpha \leq \mathcal{O}(\sigma_r^{5/3}/\sigma_1^{7/6}) = \mathcal{O}(\kappa^{7/6}\sqrt{\sigma_r})$. Also, by Lemma G.3, Eq.(D.10) holds for all $t \in [T_0]$.

**Proof of Eq.(D.13)**

Recall $\Delta_t = U_t^\top U_t + J_t^\top J_t - V_t^\top V_t - K_t^\top K_t$, then for all $t \leq T_0$, we have

$$\|\Delta_t - \Delta_0\| \leq 2\eta^2 \cdot 25\sigma_1^2 \cdot T_0 \cdot 4\sigma_1 \leq 2c_2\log(\sqrt{\sigma_1}/n\alpha)(20\sigma_1^3\eta/\sigma_r) = 200c_2\eta\kappa\sigma_1^2\log(\sqrt{\sigma_1}/n\alpha) \leq \alpha^2/8.$$

The first inequality holds by Eq.(D.5) and $\|F_t G_t - \Sigma\| \leq \|F_t\|\|G_t\| + \|\Sigma\| \leq 5\sigma_1$. The last inequality uses the fact that $\eta = \mathcal{O}(\alpha^2/\kappa\sigma_1^2\log(\sqrt{\sigma_1}/n\alpha))$. Thus, at $t = T_0$, we have $\lambda_{\min}(\Delta_{T_0}) \geq$

---

[5]Note that in Soltanolkotabi et al. (2023), the initialization is $F_0 = \alpha \cdot \tilde{F}_0$ and $G_0 = \alpha \cdot \tilde{G}_0$, while Lemma G.3 uses an imbalance initialization. It is easy to show that their results continue to hold with this imbalance initialization.

$\lambda_{\min}(\Delta_0) - \alpha^2/8 \geq \alpha^2/2 - \alpha^2/8 = 3\alpha^2/8$ and $\|\Delta_{T_0}\| \leq \|\Delta_0\| + 3\alpha^2/2 + \alpha^2/8 = 13\alpha^2/8$.

**Proof of Eq.(D.11)**

Now we can prove that $\|U - V\|$ keeps small during the initialization part. In fact, by Eq.(D.1) and Eq.(D.2), we have

$$
\begin{aligned}
&\|(U_{t+1} - V_{t+1})\| \\
&\leq \|U_t - V_t\|\|I - \eta\Sigma - \eta(V_t^\top V_t + K_t^\top K_t))\| + \eta\|V_t\|\|U_t^\top U_t + J_t^\top J_t - V_t^\top V_t - K_t^\top K_t\| \\
&\quad + 4\eta\delta_{2k+1}\|F_t G_t^\top - \Sigma\| \max\{\|U_t\|, \|V_t\|, \|J_t\|, \|K_t\|\} \\
&\leq (1 - \eta\sigma_r)\|U_t - V_t\| + 2\eta\alpha^2 \cdot 2\sqrt{\sigma_1} + 4\eta\delta_{2k+1} \cdot (\|F_t\|\|G_t\| + \|\Sigma\|) \cdot 2\sqrt{\sigma_1} \\
&\leq (1 - \eta\sigma_r)\|U_t - V_t\| + 2\eta\alpha^2 \cdot 2\sqrt{\sigma_1} + 40\eta\delta_{2k+1} \cdot \sigma_1^{3/2}.
\end{aligned}
$$

The second inequality uses the inequality (D.6), while the third inequality holds by $\max\{\|F_t\|, \|G_t\|\} \leq 2\sqrt{\sigma_1}$. Thus, since $\alpha = \mathcal{O}(\delta_{2k+1}\sigma_1^{3/2}/\sigma_r)$, we can get $\|U_0 - V_0\| \leq 4\alpha \leq 4\alpha + \frac{40}{\sigma_r}\delta_{2k+1}\sigma_1^{3/2}$. If $\|U_t - V_t\| \leq 4\alpha + \frac{40}{\sigma_r}\delta_{2k+1}\sigma_1^{3/2}$, we know that

$$
\begin{aligned}
\|U_{t+1} - V_{t+1}\| &\leq (1 - \eta\sigma_r)\left(4\alpha + \frac{40}{\sigma_r}\delta_{2k+1}\sigma_1^{3/2}\right) + 4\eta\alpha^2\sqrt{\sigma_1} + 40\eta\delta_{2k+1} \cdot \sigma_1^{3/2} \\
&\leq (1 - \eta\sigma_r)\left(4\alpha + \frac{40}{\sigma_r}\delta_{2k+1}\sigma_1^{3/2}\right) + 4\eta\sigma_r\alpha + \frac{40}{\sigma_r}\delta_{2k+1}\sigma_1^{3/2} \\
&\leq 4\alpha + \frac{40}{\sigma_r}\delta_{2k+1}\sigma_1^{3/2}.
\end{aligned}
$$

Hence, $\|U_t - V_t\| \leq 4\alpha + \frac{40}{\sigma_r}\delta_{2k+1}\sigma_1^{3/2}$ for $t \leq T_0$ by induction. The second inequality holds by $\alpha = \mathcal{O}(\sigma_r/\sqrt{\sigma_1})$

**Proof of Eq.(D.12)**

Now we prove that $J_t$ and $K_t$ are bounded for all $t \leq T_0$. By Eq.(D.3) and $\max\{\|F_t\|, \|G_t\|\} \leq 2\sqrt{\sigma_1}$, denote $C_2 = \max\{21c_2, 32\} \geq 32$, we have

$$
\begin{aligned}
\|J_{T_0}\| &\leq \|J_0\| + \eta \sum_{t=0}^{T_0-1} \max\{\|F_t\|, \|G_t\|\} \cdot 2\delta_{2k+1} \cdot (\|F_t\|\|G_t\| + \|\Sigma\|) \\
&\leq \|J_0\| + \eta T_0 \cdot 20\sigma_1^{3/2} \cdot \delta_{2k+1} \\
&\leq \|J_0\| + 20c_2 \log(\sqrt{\sigma_1}/n\alpha)(\delta_{2k+1} \cdot \sigma_1^{3/2}/\sigma_r) \\
&\leq 2\alpha + 20c_2 \log(\sqrt{\sigma_1}/n\alpha)(\delta_{2k+1} \cdot \sigma_1^{3/2}/\sigma_r) \\
&= 2\alpha + C_2 \log(\sqrt{\sigma_1}/n\alpha)(\delta_{2k+1} \cdot \sigma_1^{3/2}/\sigma_r).
\end{aligned}
$$

Similarly, we can prove that $\|K_{T_0}\| \leq 2\alpha + C_2 \log(\sqrt{\sigma_1}/n\alpha)(\delta_{2k+1} \cdot \sigma_1^{3/2}/\sigma_r)$. We complete the proof of Eq.(D.12). $\qquad \square$

### D.4 PHASE 1: LINEAR CONVERGENCE PHASE.

In this subsection, we analyze the first phase: the linear convergence phase. This phase starts at round $T_0$, and we assume that this phase terminates at round $T_1$. In this phase, the loss will converge linearly, with the rate independent of the initialization scale. Note that $T_1$ may tend to infinity, since this phase may not terminate. For example, when $k = r$, we can prove that this phase will not terminate (§E), and thus leading a linear convergence rate that independent on the initialization scale. In this phase, we provide the following lemma, which shows some induction hypotheses during this phase.

**Lemma D.2.** *Denote $M_t = \max\{\|U_t V_t^\top - \Sigma\|, \|U_t K_t^\top\|, \|J_t V_t^\top\|\}$. Suppose Phase 1 starts at $T_0$ and ends at the first time $T_1$ such that*

$$
\eta\sigma_r^2 M_{t-1}/64\sigma_1 < (17\eta\sigma_1\delta_{2k+1} + \eta\alpha^2)\|J_{t-1}K_{t-1}^\top\| \tag{D.14}
$$

*During Phase 1 that $T_0 \leq t \leq T_1$, we have the following three induction hypotheses:*

$$\max\{\|U_t\|, \|V_t\|\} \leq 2\sqrt{\sigma_1} \tag{D.15}$$

$$\|U_t V_t^\top - \Sigma\| \leq \sigma_r/2. \tag{D.16}$$

$$\max\{\|J_t\|, \|K_t\|\} \leq 2\sqrt{\alpha}\sigma_1^{1/4} + 2C_2 \log(\sqrt{\sigma_1}/n\alpha)(\delta_{2k+1} \cdot \kappa^2\sqrt{\sigma_1}) \leq \sqrt{\sigma_1} \tag{D.17}$$

$$\frac{7\alpha^2}{4} I \geq \Delta_t \geq \frac{\alpha^2}{4} I \tag{D.18}$$

The induction hypotheses hold for $t = T_0$ due to Lemma D.1. Let us assume they hold for $t' < t$, and consider the round $t$. Let us first prove that the $r$-th singular value of $U$ and $V$ are lower bounded by $\mathrm{poly}(\sigma_r, 1/\sigma_1)$ at round $t$, if Eq.(D.16) holds at round $t$. In fact,

$$2\sqrt{\sigma_1} \cdot \sigma_r(U) \geq \sigma_r(U)\sigma_1(V) \geq \sigma_r(UV^\top) \geq \sigma_r/2.$$

which means

$$\sigma_r(U) \geq \sigma_r/4\sqrt{\sigma_1}. \tag{D.19}$$

Similarly, $\sigma_r(V) \geq \sigma_r/4\sqrt{\sigma_1}$.

**Proof of Eq.(D.16)** First, since $\|U_{t-1}V_{t-1}^\top - \Sigma\| \leq \sigma_r/2$, by Eq.(D.19), we can get

$$\min\{\sigma_r(U_{t-1}), \sigma_r(V_{t-1})\} \geq \frac{\sigma_r}{4\sqrt{\sigma_1}} \tag{D.20}$$

Define $M_t = \max\{\|U_t V_t^\top - \Sigma\|, \|U_t K_t^\top\|, \|J_t V_t^\top\|\}$. By the induction hypothesis,

$$\max\{\|U_{t-1}\|, \|V_{t-1}\|\} \leq 2\sqrt{\sigma_1},$$

$$\max\{\|J_{t-1}\|, \|K_{t-1}\|\} \leq 2\sqrt{\alpha}\sigma_1^{1/4} + 2C_2 \log(\sqrt{\sigma_1}/n\alpha)(\delta_{2k+1}\sigma_1^{3/2}/\sigma_r).$$

Then, by the updating rule and $C_2 \geq 1$, we can get

$$U_t K_t = (1 - \eta U_{t-1}U_{t-1}^\top)U_{t-1}K_{t-1}(1 - \eta K_{t-1}K_{t-1}^\top) + \eta(\Sigma - U_{t-1}V_{t-1}^\top)VK^\top$$
$$+ \eta U_{t-1}J_{t-1}^\top J_{t-1}K_{t-1}^\top + A_t, \tag{D.21}$$

where $A_t$ is the perturbation term that contains all $\mathcal{O}(E_i(FG^\top - \Sigma))$ terms and $\mathcal{O}(\eta^2)$ terms such that

$$\|A_t\| \leq 4\eta\delta_{2k+1}\|F_t G_t^\top - \Sigma\| \max\{\|F_t\|^2, \|G_t\|^2\} + 8\eta^2\|F_t G_t^\top - \Sigma\|^2 \max\{\|F_t\|^2, \|G_t\|^2\}$$
$$+ \eta^2 \max\{\|F_t\|^2, \|G_t\|^2\}^2 \cdot \|F_t G_t - \Sigma\|$$
$$\leq 4\eta\delta_{2k+1}\|F_t G_t^\top - \Sigma\| \max\{\|F_t\|^2, \|G_t\|^2\} + 8\eta^2\|F_t G_t^\top - \Sigma\| \cdot 5\sigma_1 \cdot 4\sigma_1$$
$$+ \eta^2 \cdot 16\sigma_1^2 \cdot \|F_t G_t - \Sigma\|$$
$$\leq 4\eta\delta_{2k+1}(3M_{t-1} + \|J_{t-1}K_{t-1}^\top\|)4\sigma_1 + \eta\alpha^2(3M_{t-1} + \|J_{t-1}K_{t-1}^\top\|)$$

Using the similar technique for $J_t V_t^\top$ and $U_t V_t^\top - \Sigma$, we can finally get

$$M_t \leq \left(1 - \frac{\eta\sigma_r^2}{16\sigma_1}\right) M_{t-1} + 2\eta M_{t-1} \cdot 2\sqrt{\sigma_1} \cdot \max\{\|J_{t-1}\|, \|K_{t-1}\|\}$$
$$+ 4\eta\delta_{2k+1}(3M_{t-1} + \|J_{t-1}K_{t-1}^\top\|) \cdot 4\sigma_1 + \eta\alpha^2(3M_{t-1} + \|J_{t-1}K_{t-1}^\top\|)$$
$$\leq \left(1 - \frac{\eta\sigma_r^2}{16\sigma_1}\right) M_{t-1} + 2\eta M_{t-1} \cdot 2\sqrt{\sigma_1} \cdot \left(\alpha + C_2 \log(\sqrt{\sigma_1}/n\alpha)\delta_{2k+1}\sigma_1^{3/2}/\sigma_r\right)$$
$$+ 4\eta\delta_{2k+1}(3M_{t-1} + \|J_{t-1}K_{t-1}^\top\|) \cdot 4\sigma_1 + \eta\alpha^2(3M_{t-1} + \|J_{t-1}K_{t-1}^\top\|)$$
$$\leq \left(1 - \frac{\eta\sigma_r^2}{16\sigma_1}\right) M_{t-1} + \mathcal{O}\left(\eta\sqrt{\sigma_1} \cdot \left(\alpha + C_2 \log(\sqrt{\sigma_1}/n\alpha)\delta_{2k+1}\sigma_1^{3/2}/\sigma_r\right)\right) \cdot M_{t-1}$$
$$+ (17\eta\sigma_1\delta_{2k+1} + \eta\alpha^2)\|J_{t-1}K_{t-1}^\top\|$$
$$\leq \left(1 - \frac{\eta\sigma_r^2}{32\sigma_1}\right) M_{t-1} + (17\eta\sigma_1\delta_{2k+1} + \eta\alpha^2)\|J_{t-1}K_{t-1}^\top\|. \tag{D.22}$$

The last inequality holds by $\delta_{2k+1} = \mathcal{O}(\sigma_r^3/\sigma_1^3 \log(\sqrt{\sigma_1}/n\alpha))$ and $\alpha = \mathcal{O}(\sigma_r^2/\sigma_1^{3/2}) = \mathcal{O}(\sqrt{\sigma_r}\kappa^{-3/2})$.

During Phase 1, we have

$$\eta\sigma_r^2 M_{t-1}/64\sigma_1 \geq (17\eta\sigma_1\delta_{2k+1} + \eta\alpha^2)\|J_{t-1}K_{t-1}^\top\|,$$

then

$$M_t \leq \left(1 - \frac{\eta\sigma_r^2}{64\sigma_1}\right)M_{t-1}. \tag{D.23}$$

Hence, $\|U_t V_t^\top - \Sigma\| \leq M_t \leq M_{T_0} \leq \|F_{T_0}G_{T_0}^\top - \Sigma\| \leq \delta_{2k+1}$.

**Proof of Eq.(D.15)**  Now we bound the norm of $U_t$ and $V_t$. First, note that

$$\|(U_t - V_t)\| \leq (1 - \eta\sigma_r)\|U_{t-1} - V_{t-1}\| + \eta \cdot 2\alpha^2 \cdot 2\sqrt{\sigma_1} + 40\eta \cdot \delta_{2k+1} \cdot \sigma_1^{3/2}$$

Hence, $\|U_t - V_t\| \leq 4\alpha + 40\delta_{2k+1}\sigma_1^{3/2}/\sigma_r$ still holds using the same technique in the initialization part.

Thus, by the induction hypothesis Eq.(D.16) and $\sigma_1 \geq \delta_{2k+1}$, we have

$$\begin{aligned}
2\sigma_1 \geq \sigma_1 + \delta_{2k+1} \geq \|\Sigma\| + \|U_t V_t^\top - \Sigma\| &\geq \|U_t V_t^\top\| = \|V_t V_t^\top + (U_t - V_t)V_t^\top\| \\
&\geq \|V_t V_t^\top\| - \|U_t - V_t\|\|V_t\| \\
&\geq \|V_t\|^2 - \|V_t\| \cdot \left(4\alpha + \frac{40\delta_{2k+1}\sigma_1^{3/2}}{\sigma_r}\right) \\
&\geq \|V_t\|^2 - \|V_t\|.
\end{aligned}$$

Then, we can get $\|V_t\| \leq 2\sqrt{\sigma_1}$. Similarly, $\|U_t\| \leq 2\sqrt{\sigma_1}$.

**Proof of Eq.(D.17)**  Since during Phase 1,

$$\|J_t K_t^\top\| \leq M_t \cdot \frac{\sigma_r^2}{64\sigma_1(17\sigma_1\delta_{2k+1} + \alpha^2)} \leq M_t \cdot \frac{1}{1088\kappa^2\delta_{2k+1} + 64\alpha^2\kappa/\sigma_r},$$

by $\delta_{2k+1} < 1/128$ and Eq.(D.23),

$$\|F_t G_t^\top - \Sigma\| \leq 4\max\{\|J_t K_t^\top\|, M_t\} \leq 4M_t \cdot \max\left\{1, \frac{1}{1088\kappa^2\delta_{2k+1} + 64\alpha^2\kappa/\sigma_r}\right\}$$

$$\leq \|F_{T_0}G_{T_0} - \Sigma\|\left(1 - \eta\sigma_r^2/64\sigma_1\right)^{t-T_0}/(1088\kappa^2\delta_{2k+1} + 64\alpha^2\kappa/\sigma_r). \tag{D.24}$$

Thus, the maximum norm of $J_t, K_t$ can be bounded by

$$\begin{aligned}
\|J_t\| &\leq \|J_{T_0}\| + 2\eta \cdot 2\sqrt{\sigma_1}\delta_{2k+1} \cdot \sum_{t'=T_0}^{t-1} \|F_t G_t - \Sigma\| \\
&\leq 2\alpha + C_2\log(\sqrt{\sigma_1}/n\alpha)(\delta_{2k+1} \cdot \sigma_1^{3/2}/\sigma_r) + \frac{4\eta\sqrt{\sigma_1}\delta_{2k+1}}{1088\kappa^2\delta_{2k+1} + 64\alpha^2\kappa/\sigma_r} \cdot \|F_{T_0}G_{T_0} - \Sigma\| \cdot \frac{64\sigma_1}{\eta\sigma_r^2} \\
&= 2\alpha + C_2\log(\sqrt{\sigma_1}/n\alpha)(\delta_{2k+1} \cdot \sigma_1^{3/2}/\sigma_r) + \frac{\sigma_1^{3/2}}{4\kappa^2\sigma_r^2} \cdot \|F_{T_0}G_{T_0} - \Sigma\| \\
&\leq 2\alpha + C_2\log(\sqrt{\sigma_1}/n\alpha)(\delta_{2k+1} \cdot \sigma_1^{3/2}/\sigma_r) + \frac{\alpha^{1/2}\sigma_1^{9/4}}{4\kappa^2\sigma_r^2} \\
&\leq 2\sqrt{\alpha}\sigma_1^{1/4} + C_2\log(\sqrt{\sigma_1}/n\alpha)(\delta_{2k+1} \cdot \kappa^2\sqrt{\sigma_1}) \\
&\leq 2\sqrt{\alpha}\sigma_1^{1/4} + 2C_2\log(\sqrt{\sigma_1}/n\alpha)(\delta_{2k+1} \cdot \kappa^2\sqrt{\sigma_1}).
\end{aligned}$$

The last inequality uses the fact that $2\alpha + \frac{\sqrt{\alpha}\sigma_1^{1/4}}{4} \leq 2\sqrt{\alpha}\sigma_1^{1/4}$ by $\alpha = \mathcal{O}(\sqrt{\sigma_r})$. Similarly, $\|K_t\| \leq 2\sqrt{\alpha}\sigma_1^{1/4} + 2C_2\log(\sqrt{\sigma_1}/n\alpha)(\delta_{2k+1} \cdot \kappa^2 \cdot \sqrt{\sigma_1})$. We complete the proof of Eq.(D.17).

**Proof of Eq.(D.18)**    Last, for $t \in [T_0, T_1)$, we have

$$\|\Delta_t - \Delta_{T_0}\| \leq \sum_{t=T_0}^{T_1-1} 2(\eta^2 \cdot \|F_t G_t^\top - \Sigma\|^2 \cdot \max\{\|F_t\|, \|G_t\|\}^2)$$

$$\leq 2\eta^2 \|F_{T_0} G_{T_0} - \Sigma\|^2 \sum_{t=T_0}^{\infty} \left(1 - \frac{\eta\sigma_r^2}{16\sigma_1}\right)^{2(t-T_0)} \cdot 4\sigma_1$$

$$\leq 2\eta^2 \cdot 25\sigma_1^2 \cdot \frac{16\sigma_1}{\eta\sigma_r^2} \cdot 4\sigma_1$$

$$\leq 3200\eta\kappa^2\sigma_1^2$$

$$\leq \alpha^2/8,$$

where the last inequality arises from the fact that $\eta = \mathcal{O}(\alpha^2/\kappa^2\sigma_1^2)$. By $\frac{3\alpha^2}{8}I \leq \Delta_{T_0} \leq \frac{13\alpha^2}{8}I$, we can have $\|\Delta_t\| \leq 13\alpha^2/8 + \alpha^2/8 \leq 7\alpha^2/4$ and $\lambda_{\min}(\Delta_t) \geq 3\alpha^2/8 - \alpha^2/8 = \alpha^2/4$. Hence, the inequality Eq.(D.18) still holds during Phase 1. Moreover, by Eq.(D.24), during the Phase 1, for a round $t \geq 0$, we will have

$$\|F_{t+T_0} G_{t+T_0}^\top - \Sigma\| \leq \|F_{T_0} G_{T_0} - \Sigma\| \left(1 - \eta\sigma_r^2/64\sigma_1\right)^t / (1088\kappa^2\delta_{2k+1} + 64\alpha^2\kappa/\sigma_r)$$

$$\leq \|F_{T_0} G_{T_0} - \Sigma\| \left(1 - \eta\sigma_r^2/64\sigma_1\right)^t \cdot \frac{\sigma_r}{64\alpha^2\kappa}$$

$$\leq \frac{\sigma_r}{2} \cdot \left(1 - \eta\sigma_r^2/64\sigma_1\right)^t \cdot \frac{\sigma_r}{64\alpha^2\kappa}$$

$$= \frac{\sigma_r^2}{128\alpha^2\kappa} \left(1 - \eta\sigma_r^2/64\sigma_1\right)^t. \tag{D.25}$$

The conclusion (D.25) always holds in Phase 1. Note that Phase 1 may not terminate, and then the loss is linear convergence. We assume that at round $T_1$, Phase 1 terminates, which implies that

$$\sigma_r^2 M_{T_1-1}/64\sigma_1 < (17\sigma_1\delta_{2k+1} + \alpha^2)\|J_{T_1-1} K_{T_1-1}^\top\|, \tag{D.26}$$

and the algorithm goes to Phase 2.

## D.5    PHASE 2: ADJUSTMENT PHASE.

In this phase, we prove $U - V$ will decrease exponentially. This phase terminates at the first time $T_2$ such that

$$\|U_{T_2-1} - V_{T_2-1}\| \leq \frac{8\alpha^2\sqrt{\sigma_1} + 64\delta_{2k+1}\sqrt{\sigma_1}\|J_{T_2-1} K_{T_2-1}^\top\|}{\sigma_r}. \tag{D.27}$$

By stopping rule (D.27), since $\|U_{T_1} - V_{T_1}\| \leq \mathcal{O}(\sigma_1)$, this phase will take at most $\mathcal{O}(\log(\sqrt{\sigma_r}/\alpha)/\eta\sigma_r)$ rounds, i.e.

$$T_2 - T_1 = \mathcal{O}(\log(\sqrt{\sigma_r}/\alpha)/\eta\sigma_r). \tag{D.28}$$

We use the induction to show that all the following hypotheses hold during Phase 2.

$$\max\{\|F_{t-1}\|, \|G_{t-1}\| \leq 2\sqrt{\sigma_1} \tag{D.29}$$

$$M_t \leq (1088\kappa^2\delta_{2k+1} + 64\alpha^2\kappa/\sigma_r)\|J_t K_t^\top\| \leq \|J_t K_t^\top\| \tag{D.30}$$

$$\max\{\|J_{t-1}\|, \|K_{t-1}\|\} \leq 2\sqrt{\alpha}\sigma_1^{1/4} + (2C_2 + 16C_3)\log(\sqrt{\sigma_1}/n\alpha)(\delta_{2k+1} \cdot \kappa^2\sqrt{\sigma_1}) \leq \sigma_r/4\sqrt{\sigma_1} \tag{D.31}$$

$$\|J_t K_t^\top\| \leq \left(1 + \frac{\eta\sigma_r^2}{128\sigma_1}\right) \|J_{t-1} K_{t-1}^\top\| \tag{D.32}$$

$$\|U_t - V_t\| \leq (1 - \eta\sigma_r/2)\|U_{t-1} - V_{t-1}\| \tag{D.33}$$

$$\frac{3\alpha^2}{16} \cdot I \leq \Delta_t \leq \frac{29\alpha^2}{16} \cdot I. \tag{D.34}$$

**Proof of** (D.31)    To prove this, we first assume that this adjustment phase will only take at most $C_3(\log(\alpha)/\eta\sigma_r)$ rounds. By the induction hypothesis for the previous rounds,

$$\|J_t\| \leq J_{T_1} + \sum_{i=T_1}^{t-1} \eta\delta_{2k+1} \cdot \|F_t G_t^\top - \Sigma\|$$

$$\leq 2\sqrt{\alpha}\sigma_1^{1/4} + 2C_2\log(\sqrt{\sigma_1}/n\alpha)(\delta_{2k+1} \cdot \sigma_1^{3/2}/\sigma_r) + \sum_{i=T_1}^{t-1} \eta\delta_{2k+1} \cdot \|F_i G_i^\top - \Sigma\|$$

$$\leq 2\sqrt{\alpha}\sigma_1^{1/4} + 2C_2\log(\sqrt{\sigma_1}/n\alpha)(\delta_{2k+1} \cdot \sigma_1^{3/2}/\sigma_r) + C_3(\log(\sqrt{\sigma_1}/n\alpha)/\eta\sigma_r) \cdot \eta\delta_{2k+1} \cdot 4\|J_{i-1}K_{i-1}^\top\|$$

$$\leq 2\sqrt{\alpha}\sigma_1^{1/4} + 2C_2\log(\sqrt{\sigma_1}/n\alpha)(\delta_{2k+1} \cdot \sigma_1^{3/2}/\sigma_r) + C_3(\log(\sqrt{\sigma_1}/n\alpha)/\eta\sigma_r) \cdot \eta\delta_{2k+1}16\sigma_1$$

$$\leq 2\sqrt{\alpha}\sigma_1^{1/4} + (2C_2 + 16C_3)\log(\sqrt{\sigma_1}/n\alpha)(\delta_{2k+1} \cdot \sigma_1^{3/2}/\sigma_r).$$

Similarly, due to the symmetry property, we can bound the $\|K_t\|$ using the same technique. Thus,

$$\max\{\|J_t\|, \|K_t\|\} \leq 2\sqrt{\alpha}\sigma_1^{1/4} + (2C_2 + 16C_3)\log(\sqrt{\sigma_1}/n\alpha)(\delta_{2k+1} \cdot \sigma_1^{3/2}/\sigma_r).$$

**Proof of** (D.30)    First, we prove that during $t \in [T_1, T_2)$,

$$M_t \leq (1088\kappa^2\delta_{2k+1} + 64\alpha^2\kappa/\sigma_r)\|J_t K_t^\top\| \leq \|J_t K_t^\top\| \leq 4\alpha\kappa^4\sigma_1^{1/2} + \delta_{2k+1}\sigma_1. \qquad \text{(D.35)}$$

in this phase.

Then, by $\delta_{2k+1} \leq \mathcal{O}(1/\log(\sqrt{\sigma_1}/n\alpha)\kappa^2)$ and $\alpha \leq \mathcal{O}(\sigma_r/\sqrt{\sigma_1})$, choosing sufficiently small coefficient, we can have

$$J_t K_t^\top = (I - \eta J_{t-1}J_{t-1}^\top)J_{t-1}K_{t-1}^\top(I - \eta K_{t-1}K_{t-1}^\top) + \eta^2 J_{t-1}J_{t-1}^\top J_{t-1}K_{t-1}^\top K_{t-1}K_{t-1}^\top$$
$$- \eta J_{t-1}V_{t-1}^\top V_{t-1}K_{t-1}^\top - \eta J_t U_t^\top U_t K_t^\top + C_{t-1}, \qquad \text{(D.36)}$$

where $C_t$ represents the relatively small perturbation term, which contains terms of $\mathcal{O}(\delta)$ and $\mathcal{O}(\eta^2)$. By (D.29), we can easily get

$$C_{t-1} \geq - \left(4\eta\delta_{2k+1} \cdot \|F_{t-1}G_{t-1}^\top - \Sigma\| \cdot 4\sigma_1\right) \qquad \text{(D.37)}$$

Thus, combining (D.36) and (D.37), we have

$$\|J_t K_t^\top\|$$
$$\geq \|I - \eta J_{t-1}J_{t-1}^\top\|\|I - \eta K_{t-1}K_{t-1}^\top\|\|J_{t-1}K_{t-1}^\top\| - 4\eta M_{t-1} \cdot 4\sigma_1$$
$$- 4\eta\delta_{2k+1}\|J_{t-1}K_{t-1}\| \cdot 2\sigma_1 - \eta^2 64\sigma_1^3$$
$$\geq \left(1 - 2\eta\max\{\|J_{t-1}\|, \|K_{t-1}\|\}^2 - 16 \cdot 1088\eta\kappa^2\delta_{2k+1}\sigma_1 - 1024\eta\alpha^2\kappa^2 - 8\eta\delta_{2k+1} \cdot \sigma_1\right)\|J_{t-1}K_{t-1}^\top\|$$
$$\geq \left(1 - \frac{\eta\sigma_r^2}{128\sigma_1}\right)\|J_{t-1}K_{t-1}^\top\|.$$

The second inequality is because $M_{t-1} \leq (1088\kappa^2\delta_{2k+1} + 64\alpha^2\kappa/\sigma_r)\|J_{t-1}K_{t-1}^\top\|$, and the last inequality holds by Eq.(D.31) and

$$\delta_{2k+1} = \mathcal{O}(\kappa^{-4}), \alpha = \mathcal{O}(\kappa^{-3/2}\sqrt{\sigma_r}) \qquad \text{(D.38)}$$

Then, note that by Eq.(D.22), we have

$$M_t \leq \left(1 - \frac{\eta\sigma_r^2}{32\sigma_1}\right)M_{t-1} + (17\eta\sigma_1\delta_{2k+1} + \eta\alpha^2)\|J_{t-1}K_{t-1}^\top\|.$$

Then, by $M_{t-1} \leq (1088\kappa^2\delta_{2k+1} + 64\alpha^2\kappa/\sigma_r) \cdot \|J_{t-1}K_{t-1}^\top\|$ and denote $L = 17\sigma_1\delta_{2k+1} + \alpha^2$, we have

$$
\begin{aligned}
M_t &\leq \left(1 - \frac{\eta\sigma_r^2}{32\sigma_1}\right) M_{t-1} + (17\eta\sigma_1\delta_{2k+1} + \eta\alpha^2)\|J_{t-1}K_{t-1}^\top\| \\
&\leq \left(1 - \frac{\eta\sigma_r^2}{32\sigma_1}\right) \cdot (1088\kappa^2\delta_{2k+1} + 64\alpha^2\kappa/\sigma_r)\|J_{t-1}K_{t-1}^\top\| + \eta L\|J_{t-1}K_{t-1}^\top\| \\
&= \left(1 - \frac{\eta\sigma_r^2}{32\sigma_1}\right) \cdot \frac{64L\kappa}{\sigma_r}\|J_{t-1}K_{t-1}^\top\| + \eta L\|J_{t-1}K_{t-1}^\top\| \\
&\leq \left(\frac{64L\kappa}{\sigma_r} - 2\eta L\right) \|J_{t-1}K_{t-1}^\top\| \\
&\leq \left(\frac{64L\kappa}{\sigma_r} - 2\eta L\right) \Big/ \left(1 - \frac{\eta\sigma_r^2}{128\sigma_1}\right) \|J_t K_t^\top\| \\
&\leq \frac{64L\kappa}{\sigma_r}\|J_t K_t^\top\|.
\end{aligned}
$$

Hence,

$$
M_t \leq \frac{64L\kappa}{\sigma_r}\|J_t K_t^\top\| \leq \|J_t K_t^\top\|
$$

for all $t$ in Phase 2. The last inequality is because $\delta_{2k+1} = \mathcal{O}(1/\kappa^2\log(\sqrt{\sigma_1}/n\alpha))$. Moreover, by $\delta_{2k+1} \leq \mathcal{O}(1/\kappa^2\log(\sqrt{\sigma_1}/n\alpha)^2)$ and $(a+b)^2 \leq 2a^2 + 2b^2$ we have

$$
\|J_t K_t^\top\| \leq \|J_t\|\|K_t\| \leq \left(2\sqrt{\alpha}\sigma_1^{1/4} + (2C_2 + 16C_3)\log(\sqrt{\sigma_1}/n\alpha)(\delta_{2k+1} \cdot \kappa^2\sqrt{\sigma_1})\right)^2 \quad \text{(D.39)}
$$

$$
\leq 4\alpha\kappa^4\sigma_1^{1/2} + \delta_{2k+1}\sigma_1. \quad \text{(D.40)}
$$

We complete the proof of Eq.(D.30).

**Proof of Eq.(D.32)**  Moreover, by the updating rule of $J_t$ and $K_t$, (D.36) and (D.37) we have

$\|J_t K_t^\top\|$

$$
\leq \|(I - \eta J_{t-1}J_{t-1}^\top)J_{t-1}K_{t-1}^T(I - \eta K_{t-1}K_{t-1}^\top)\| + \|\eta^2(J_{t-1}J_{t-1}^\top)J_{t-1}K_{t-1}^T(K_{t-1}K_{t-1}^\top)\| \tag{D.41}
$$

$$
+ 4\eta M_{t-1} \cdot 4\sigma_1 + 4\eta\delta_{2k+1}\|J_{t-1}K_{t-1}^\top\| \cdot 2\sigma_1
$$

$$
\leq \|J_{t-1}K_{t-1}^\top\| + \eta^2(\sqrt{\sigma_1}/2)^4\|J_{t-1}K_{t-1}^\top\| + 4\eta\frac{64L\kappa}{\sigma_r}\|J_{t-1}K_{t-1}^\top\| \cdot 4\sigma_1 + 8\eta\sigma_1\delta_{2k+1}\|J_{t-1}K_{t-1}^\top\|
$$

$$
= \|J_{t-1}K_{t-1}^\top\| \cdot \left(1 + \eta^2\sigma_1^2/16 + 1024L\kappa^2 + 8\sigma_1\delta_{2k+1}\right).
$$

The last inequality uses the fact that $\|J_{t-1}\| \leq \sqrt{\sigma_1}/2, \|K_{t-1}\| \leq \sqrt{\sigma_1}/2$ and $M_{t-1} \leq \frac{64L\kappa}{\sigma_r}\|J_{t-1}K_{t-1}^\top\|$. Now by the fact that $L = 17\sigma_1\delta_{2k+1} + \alpha^2 = \mathcal{O}(\frac{\sigma_r^2}{\sigma_1\kappa^2})$, we can choose small constant so that

$$
\eta^2\sigma_1^2/16 \leq \frac{\sigma_r^2}{384\sigma_1}, \quad 1024L\kappa^2 \leq \frac{\sigma_r^2}{384\sigma_1}, \quad 8\sigma_1\delta_{2k+1} \leq \frac{\sigma_r^2}{384\sigma_1}.
$$

Thus, we can have

$$
\|J_t K_t^\top\| \leq \|J_{t-1}K_{t-1}^\top\| \cdot \left(1 + \frac{\eta\sigma_r^2}{128\sigma_1}\right).
$$

We complete the proof of (D.32)

**Proof of (D.33)**  Hence, similar to Phase 1, by $\|U_t V_t^\top - \Sigma\| \leq M_t \leq 4\alpha\kappa^4\sigma_1^{1/2} + \delta_{2k+1}\sigma_1$ and $\|U_t - V_t\| \leq \|U_{T_1} - V_{T_1}\| \leq 4\alpha + \frac{40\delta\sigma_1^{3/2}}{\sigma_r}$, we can show that

$$
\max\{\|U_t\|, \|V_t\|\} \leq 2\sqrt{\sigma_1}
$$

Also, consider

$$
\begin{aligned}
U_t &- V_t \\
&= (I - \eta\Sigma - V_t^\top V_t - K_t^\top K_t)(U_{t-1} - V_{t-1}) - \eta V_t \Delta_t \\
&\quad + \eta \cdot \left( E_1(F_{t-1}G_{t-1}^\top - \Sigma)V_{t-1} + E_2(F_{t-1}G_{t-1}^\top - \Sigma)K_{t-1} \right) \\
&\quad - \eta \cdot \left( E_1^\top(F_{t-1}G_{t-1}^\top - \Sigma)U_{t-1} + E_3^\top(F_{t-1}G_{t-1}^\top - \Sigma)J_{t-1} \right).
\end{aligned}
$$

Hence, by the RIP property and $\Delta_{t-1} \le 2\alpha^2 I$ ((D.34)), we can get

$$
\begin{aligned}
\|(U_t - V_t)\| &\le (1 - \eta\sigma_r)\|U_{t-1} - V_{t-1}\| + 2\eta\alpha^2 \cdot 2\sqrt{\sigma_1} + 4\eta\delta_{2k+1} \cdot 2\sqrt{\sigma_1} \cdot \|F_{t-1}G_{t-1}^\top - \Sigma\| \\
&\le (1 - \eta\sigma_r)\|U_{t-1} - V_{t-1}\| + 2\eta\alpha^2 \cdot 2\sqrt{\sigma_1} + 8\eta\delta_{2k+1} \cdot \sqrt{\sigma_1} \cdot 4\|J_{t-1}K_{t-1}^\top\| \\
&\le (1 - \eta\sigma_r)\|U_{t-1} - V_{t-1}\| + 2\eta\alpha^2 \cdot 2\sqrt{\sigma_1} + 32\eta\delta_{2k+1} \cdot \sqrt{\sigma_1} \cdot \|J_{t-1}K_{t-1}^\top\|
\end{aligned}
$$

Since

$$
\|U_{t-1} - V_{t-1}\| \ge \frac{8\alpha^2\sqrt{\sigma_1} + 64\delta_{2k+1}\sqrt{\sigma_1}\|J_{t-1}K_{t-1}^\top\|}{\sigma_r}.
$$

for all $t$ in Phase 2, we can have

$$
\|U_t - V_t\| \le (1 - \eta\sigma_r/2)\|U_{t-1} - V_{t-1}\|
$$

during Phase 2.

Moreover, since Phase 2 terminates at round $T_2$, such that

$$
\|U_{T_2-1} - V_{T_2-1}\| \le \frac{8\alpha^2\sqrt{\sigma_1} + 64\delta_{2k+1}\sqrt{\sigma_1}\|J_{T_2-1}K_{T_2-1}^\top\|}{\sigma_r},
$$

it takes at most

$$
C_3 \log(\sqrt{\sigma_r}/\alpha)/\eta\sigma_r = t_2^* \tag{D.42}
$$

rounds for some constant $C_3$ because (a) (D.33), (b) and $U_t - V_t$ decreases from $\|U_{T_1} - V_{T_1}\| \le 4\sqrt{\sigma_1}$ to at most $\|U_{T_2} - V_{T_2}\| = \Omega(\alpha^2\sqrt{\sigma_1}/\sigma_r)$. Also, the changement of $\Delta_t$ can be bounded by

$$
\begin{aligned}
\|\Delta_t - \Delta_{T_1}\| &\le \sum_{t=T_1}^{T_2-1} 2(\eta^2 \cdot \|F_t G_t^\top - \Sigma\|^2 \cdot 4\sigma_1) \\
&\le 2(\eta^2) \cdot 100\sigma_1^3 \cdot (T_2 - T_1) \\
&\le 2(\eta^2) \cdot 100\sigma_1^3 \cdot C_3 \log(\sqrt{\sigma_1}/n\alpha)(1/\eta\sigma_r) \\
&\le 10C_3 \log(\sqrt{\sigma_1}/n\alpha)(\eta\kappa\sigma_1^2) \\
&\le \alpha^2/16.
\end{aligned}
$$

The last inequality holds by choosing $\eta \le \alpha^2/160C_3\kappa\sigma_1^2$. Then, $\lambda_{\min}(\Delta_t) \ge \lambda_{\min}\Delta_{T_1} - \alpha^2/16 \ge \alpha^2/4 - \alpha^2/16 = 3\alpha^2/16$ and $\|\Delta_t\| \le \|\Delta_{T_1}\| + \alpha^2/16 \le 7\alpha^2/4 + \alpha^2/16 \le 29\alpha^2/16$. Hence, inequality (D.6) still holds during Phase 2.

### D.6 PHASE 3: LOCAL CONVERGENCE

In this phase, we show that the norm of $K_t$ will decrease at a linear rate. Denote the SVD of $U_t$ as $U_t = A_t\Sigma_t W_t$, where $\Sigma_t \in \mathbb{R}^{r\times r}$, $W_t \in \mathbb{R}^{r\times k}$, and define $W_{t,\perp} \in \mathbb{R}^{(k-r)\times k}$ is the complement of $W_t$.

We use the induction to show that all the following hypotheses hold during Phase 3.

$$\max\{\|J_t\|, \|K_t\|\} \leq \mathcal{O}(2\sqrt{\alpha}\sigma_1^{1/4} + \delta_{2k+1}\log(\sqrt{\sigma_1}/n\alpha) \cdot \kappa^2\sqrt{\sigma_1}) \leq \sqrt{\sigma_1}/2 \tag{D.43}$$

$$M_t \leq \frac{64L\kappa}{\sigma_r}\|J_tK_t^\top\| \leq \|J_tK_t^\top\| \tag{D.44}$$

$$\|J_tK_t^\top\| \leq \left(1 + \frac{\eta\sigma_r^2}{128\sigma_1}\right)\|J_{t-1}K_{t-1}^\top\| \tag{D.45}$$

$$\|U_t - V_t\| \leq \frac{8\alpha^2\sqrt{\sigma_1} + 64\delta_{2k+1}\sqrt{\sigma_1}\|J_tK_t^\top\|}{\sigma_r} \tag{D.46}$$

$$\frac{\alpha^2}{8} \cdot I \leq \Delta_t \leq 2\alpha^2 I \tag{D.47}$$

$$\|K_t\| \leq 2\|K_tW_{t,\perp}^\top\| \tag{D.48}$$

$$\|K_{t+1}W_{t+1,\perp}^\top\| \leq \|K_tW_{t,\perp}^\top\| \cdot \left(1 - \frac{\eta\alpha^2}{8}\right). \tag{D.49}$$

Assume the hypotheses above hold before round $t$, then at round $t$, by the same argument in Phase 1 and 2, the inequalities (D.44) and (D.46) still holds, then $\max\{\|U_t\|, \|V_t\|\} \leq 2\sqrt{\sigma_1}$ and $\min\{\sigma_r(U), \sigma_r(V)\} \geq \sigma_r/4\sqrt{\sigma_1}$.

Last, we should prove the induction hypotheses (D.43) , (D.47), (D.48) and (D.49).

**Proof of Eq.(D.45)**  Similar to the proof of (D.32) in Phase 2, we can derive (D.45) again.

**Proof of Eq.(D.48)**  First, to prove (D.48), note that we can get

$$M_t \geq \|U_tK_t\| = \|A_t\Sigma_tW_tK_t^\top\| = \|\Sigma_tW_tK_t^\top\|$$
$$\geq \sigma_r(U) \cdot \|K_tW_t^\top\| \geq \frac{\|K_tW_t^\top\|\sigma_r}{4\sqrt{\sigma_1}} \geq \frac{\|K_tW_t^\top\|\sqrt{\sigma_r}}{4\sqrt{\kappa}}.$$

Hence,

$$\|K_tW_t^\top\| \leq 4\sqrt{\kappa}M/\sqrt{\sigma_r} \leq \frac{64\sigma_1 L\sqrt{\kappa}}{\sigma_r^{5/2}}\|J_tK_t^\top\| \leq \frac{32L\kappa^{3/2}}{\sigma_r^{3/2}}\|K_t\| \cdot \sqrt{\sigma_1} \leq \frac{32L\kappa^2}{\sigma_r}\|K_t\|. \tag{D.50}$$

Thus,

$$\|K_t\| \leq \|K_tW_{t,\perp}^\top\| + \|K_tW_t^\top\|$$
$$\leq \|K_tW_{t,\perp}^\top\| + \frac{64L\kappa}{\sigma_r}\|K_t\|$$
$$\leq \|K_tW_{t,\perp}^\top\| + \frac{1}{2}\|K_t\|.$$

The last inequality uses the fact that $\delta_{2k+1} = \mathcal{O}(\sigma_r^3/\sigma_1^3)$ Hence, $\|K_tW_{t,\perp}^\top\| \geq \|K_t\|/2$, and (D.48) holds during Phase 3.

**Proof of Eq.(D.47)** To prove the (D.47), by the induction hypothesis of Eq.(D.49), note that

$$
\begin{aligned}
\|\Delta_t - \Delta_{T_2}\| &\leq 2\eta^2 \cdot \sum_{t'=T_2}^{t-1} \|F_{t'} G_{t'}^\top - \Sigma\|^2 4\sigma_1 \\
&\leq 2\eta^2 \sum_{t'=T_2}^{t-1} 16\sigma_1 \|J_{t'} K_{t'}^\top\|^2 \\
&\leq 64\sigma_1 \eta^2 \cdot \sum_{t'=T_2}^{\infty} \|J_{t'}\|^2 \|K_{t'} W_{t',\perp}^\top\|^2 \\
&\leq 64\sigma_1 \cdot \eta^2 \left( \sigma_1 \cdot \|K_{T_2} W_{T_2,\perp}^\top\|^2 \cdot \frac{8}{\eta\alpha^2} \right) \qquad \text{(D.51)} \\
&\leq \frac{512\eta\sigma_1^2}{\alpha^2} \cdot \|K_{T_2}\|^2 \\
&\leq \frac{128\eta\sigma_1^2}{\alpha^2} \cdot \sigma_1 \\
&\leq \alpha^2/16.
\end{aligned}
$$

The Eq.(D.51) holds by the sum of geometric series. The last inequality holds by $\eta \leq \mathcal{O}(\alpha^4/\sigma_1^3)$ Then, we have

$$
\|\Delta_t\| \leq \|\Delta_{T_2}\| + \|\Delta_t - \Delta_{T_2}\| \leq \frac{29\alpha^2}{16} + \frac{\alpha^2}{16} \leq 2\alpha^2.
$$

$$
\lambda_{\min}(\Delta_t) \geq \lambda_{\min}(\Delta_{T_2}) - \|\Delta_t - \Delta_{T_2}\| \geq \frac{3\alpha^2}{16} - \frac{\alpha^2}{16} = \frac{\alpha^2}{8}.
$$

Hence, (D.47) holds during Phase 3.

**Proof of Eq.(D.43)** To prove the (D.43), note that

$$
\|K_t\| \leq 2\|K_t W_{t,\perp}^\top\| \leq 2\|K_{T_2} W_{T_2,\perp}^\top\| \leq 2\|K_{T_2}\| \leq \mathcal{O}(\delta_{2k+1} \log(\sqrt{\sigma_1}/n\alpha) \cdot \sigma_1^{3/2}/\sigma_r). \quad \text{(D.52)}
$$

On the other hand, by $\Delta_t \leq 2\alpha^2 I$, we have

$$
W_{t,\perp} J_t^\top J_t W_{t,\perp}^\top - W_{t,\perp} K_t^\top K_t W_{t,\perp}^\top - W_{t,\perp} V_t^\top V_t W_{t,\perp}^\top \leq 2\alpha^2 \cdot I.
$$

Hence, denote $L_t = \|J_t K_t^\top\| \leq \sigma_1/4$,

$$
\begin{aligned}
W_{t,\perp} J_t^\top J_t W_{t,\perp}^\top &\leq 2\alpha^2 I + W_{t,\perp} K_t^\top K_t W_{t,\perp}^\top + W_{t,\perp} V_t^\top V_t W_{t,\perp}^\top \\
&= 2\alpha^2 I + W_{t,\perp} K_t^\top K_t W_{t,\perp}^\top + W_{t,\perp}(V_t - U_t)^\top (V_t - U_t) W_{t,\perp}^\top \\
&\leq 2\alpha^2 I + W_{t,\perp} K_t^\top K_t W_{t,\perp}^\top + \left( \frac{8\alpha^2 \sqrt{\sigma_1} + 64\delta_{2k+1}\sqrt{\sigma_1}L_t}{\sigma_r} \right)^2 \cdot I \\
&= W_{t,\perp} K_t^\top K_t W_{t,\perp}^\top + \left( 2\alpha + \frac{8\alpha^2\sqrt{\sigma_1} + 64\delta_{2k+1}\sqrt{\sigma_1}L_t}{\sigma_r} \right)^2 I. \qquad \text{(D.53)}
\end{aligned}
$$

Also, by inequality (D.53), we have

$$
\begin{aligned}
\|J_t W_{t,\perp}^\top\| - \|K_t W_{t,\perp}^\top\| &\leq \frac{\|J_t W_{t,\perp}^\top\|^2 - \|K_t W_{t,\perp}^\top\|^2}{\|J_t W_{t,\perp}^\top\| + \|K_t W_{t,\perp}^\top\|} \\
&\leq \frac{\left( 2\alpha + \frac{8\alpha^2\sqrt{\sigma_1} + 64\delta_{2k+1}\sqrt{\sigma_1}L_t}{\sigma_r} \right)^2}{2\|K_t W_{t,\perp}^\top\| + \|J_t W_{t,\perp}^\top\| - \|K_t W_{t,\perp}^\top\|} \\
&\leq \frac{\left( 2\alpha + \frac{8\alpha^2\sqrt{\sigma_1} + 64\delta_{2k+1}\sqrt{\sigma_1}L_t}{\sigma_r} \right)^2}{\|J_t W_{t,\perp}^\top\| - \|K_t W_{t,\perp}^\top\|}
\end{aligned}
$$

Thus, by $L_t \leq \sigma_1/4$, we can get

$$\|J_t W_{t,\perp}^\top\| \leq \|K_t W_{t,\perp}^\top\| + 2\alpha + \frac{8\alpha^2\sqrt{\sigma_1} + 64\delta_{2k+1}\sqrt{\sigma_1}L_t}{\sigma_r}$$

$$\leq \|K_{T_2}\| + 2\alpha + \frac{8\alpha^2\sqrt{\sigma_1} + 64\delta_{2k+1}\sqrt{\sigma_1}L_t}{\sigma_r}$$

$$\leq \mathcal{O}(2\sqrt{\alpha}\sigma_1^{1/4} + \delta_{2k+1}\log(\sqrt{\sigma_1}/n\alpha)\kappa^2\sqrt{\sigma_1}).$$

The second inequality holds by $\|K_t W_{t,\perp}^\top\| \leq \|K_{T_2}W_{T_2,\perp}^\top\| \leq \|K_{T_2}\|$. On the other hand, note that

$$\|J_t\| \leq \|J_t W_t^\top\| + \|J_t W_{t,\perp}^\top\|$$

$$\leq \|J_t U_t^\top\|/\sigma_r(U) + \|J_t W_{t,\perp}^\top\|$$

$$\leq \|J_t V_t\|/\sigma_r(U) + \|J_t(U_t - V_t)\|/\sigma_r(U) + \|J_t W_{t,\perp}^\top\|$$

$$\leq M_t/\sigma_r(U) + \|J_t\|\|(U_t - V_t)\|/\sigma_r(U) + \|J_t W_{t,\perp}^\top\|$$

$$\leq \frac{64L\kappa}{\sigma_r}\|J_t\|\|K_t\| \cdot \frac{4\sqrt{\sigma_1}}{\sigma_r} + \|J_t\|\frac{8\alpha^2\sqrt{\sigma_1} + 64\delta_{2k+1}\sqrt{\sigma_1}\|J_t K_t^\top\|}{\sigma_r} \cdot \frac{4\sqrt{\sigma_1}}{\sigma_r} + \|J_t W_{t,\perp}^\top\|$$

$$\leq \left(\frac{64\sigma_1^{3/2}L}{\sigma_r^3} \cdot \sqrt{\sigma_1} + \frac{32\alpha^2\sigma_1 + 256\delta_{2k+1}\sigma_1 \cdot \sigma_1}{\sigma_r^2}\right)\|J_t\| + \|J_t W_{t,\perp}^\top\|$$

$$\leq \frac{1}{2}\|J_t\| + \|J_t W_{t,\perp}^\top\|. \tag{D.54}$$

The last inequality holds because

$$\delta_{2k+1} = \mathcal{O}(\kappa^{-4}\log^{-1}(\sqrt{\sigma_1}/n\alpha)), \quad \alpha \leq \mathcal{O}(\sigma_r/\sqrt{\sigma_1})$$

Hence, by the inequality (D.54), we can get

$$\|J_t\| \leq 2\|J_t W_{t,\perp}^\top\| = \mathcal{O}(2\sqrt{\alpha}\sigma_1^{1/4} + \delta_{2k+1}\log(\sqrt{\sigma_1}/n\alpha) \cdot \kappa^2\sqrt{\sigma_1}). \tag{D.55}$$

Thus, (D.43) holds during Phase 3.

**Proof of Eq.(D.49)** Now we prove the inequality (D.49). We consider the changement of $K_t$. We have

$$K_{t+1} = K_t(I - U_t^\top U_t - J_t^\top J_t) + E_3(F_t G_t^\top - \Sigma)U_t + E_4(F_t G_t^\top - \Sigma)J_t$$

Now consider $K_{t+1}W_{t,\perp}^\top$, we can get

$$K_{t+1}W_{t,\perp}^\top = K_t(I - \eta W_t^\top \Sigma^2 W_t - J_t^\top J_t)W_{t,\perp}^\top + \eta E_3(F_t G_t^\top - \Sigma)U_t W_{t,\perp}^\top + \eta E_4(F_t G_t^\top - \Sigma)J_t W_{t,\perp}^\top$$

$$= K_t W_{t,\perp}^\top - \eta K_t J_t^\top J_t W_{t,\perp}^\top + \eta E_4(F_t G_t^\top - \Sigma)J_t W_{t,\perp}^\top$$

$$= K_t W_{t,\perp}^\top - \eta K_t W_{t,\perp}^\top W_{t,\perp}^\top J_t^\top J_t W_{t,\perp}^\top - \eta K_t W_t^\top W_t J_t^\top J_t W_{t,\perp}^\top + \eta E_4(F_t G_t^\top - \Sigma)J_t W_{t,\perp}^\top$$

Hence, by the Eq.(D.50),

$$\|K_{t+1}W_{t,\perp}^\top\| \leq \|K_t W_{t,\perp}^\top(I - \eta W_{t,\perp}J_t^\top J_t W_{t,\perp}^\top)\| + \frac{64\eta L\kappa^{3/2}}{\sigma_r^{3/2}}\|J_t K_t^\top\| \cdot \|J_t W_{t,\perp}^\top\|\|J_t\| + 4\eta\delta_{2k+1}M_t\|J_t W_{t,\perp}^\top\|$$

$$\leq \|K_t W_{t,\perp}^\top(I - \eta W_{t,\perp}J_t^\top J_t W_{t,\perp}^\top)\| + \frac{64\eta L\kappa^{3/2}}{\sigma_r^{3/2}}\|J_t K_t^\top\| \cdot \|J_t W_{t,\perp}^\top\|\|J_t\|$$

$$+ \frac{16\sigma_1\eta L}{\sigma_r^2}\|J_t K_t^\top\|\|J_t W_{t,\perp}^\top\|$$

$$\leq \|K_t W_{t,\perp}^\top(I - \eta W_{t,\perp}J_t^\top J_t W_{t,\perp}^\top)\| + \frac{80\eta L\kappa^2}{\sigma_r}\|J_t K_t^\top\| \cdot \|J_t W_{t,\perp}^\top\|$$

The second inequality uses the fact that $\delta_{2k+1} \leq 1/16$ and (D.50). The last inequality uses the fact that $\|J_t\| \leq \sqrt{\sigma_1}$. Note that $\lambda_{\min}(\Delta_t) \geq \alpha^2/8 \cdot I$, then multiply the $W_{t,\perp}^\top$, we can get

$$W_{t,\perp}J_t^\top J_t W_{t,\perp}^\top - W_{t,\perp}V_t^\top V_t W_{t,\perp}^\top - W_{t,\perp}K_t^\top K_t W_{t,\perp}^\top \geq \frac{\alpha^2}{8} \cdot I.$$

Hence,

$$W_{t,\perp} J_t^\top J_t W_{t,\perp}^\top - W_{t,\perp} K_t^\top K_t W_{t,\perp}^\top \geq \frac{\alpha^2}{8} \cdot I.$$

Thus, define $\phi_t = W_{t,\perp} J_t^\top J_t W_{t,\perp}^\top - W_{t,\perp} K_t^\top K_t W_{t,\perp}^\top$, then we can get

$$\|K_{t+1} W_{t,\perp}^\top\| \leq \|K_t W_{t,\perp}^\top (I - W_{t,\perp} J_t^\top J_t W_{t,\perp}^\top)\| + \frac{80 L \kappa^2}{\sigma_r} \|J_t K_t^\top\| \cdot \|J_t W_{t,\perp}^\top\|$$

$$\leq \|K_t W_{t,\perp}^\top (I - W_{t,\perp} K_t^\top K_t W_{t,\perp}^\top - \eta \phi_t)\| + \frac{80 L \kappa^2}{\sigma_r} \|J_t K_t^\top\| \cdot \|J_t W_{t,\perp}^\top\|$$

Define loss $L_t = \|J_t K_t^\top\|$. Note that

$$\begin{aligned}
L_t &= \|J_t K_t^\top\| \\
&= \|J_t W_{t,\perp}^\top W_{t,\perp} K_t^\top + J_t W_t^\top W_t K_t^\top\| \\
&\leq \|J_t W_{t,\perp}^\top W_{t,\perp} K_t^\top\| + \|J_t W_t^\top W_t K_t^\top\| \\
&\leq \|J_t W_{t,\perp}^\top W_{t,\perp} K_t^\top\| + \sqrt{\sigma_1} \cdot \frac{64 L \kappa^{3/2}}{\sigma_r^{3/2}} \|J_t K_t^\top\| \\
&\leq \|J_t W_{t,\perp}^\top W_{t,\perp} K_t^\top\| + \frac{L_t}{2}.
\end{aligned}$$

(D.56)

The Eq.(D.56) holds by Eq.(D.50) and $\|W_t^\top\| = 1$, and the last inequality holds by $\delta_{2k+1} = \mathcal{O}(\kappa^4)$.

Hence,

$$\|J_t W_{t,\perp}^\top W_{t,\perp} K_t^\top\| \geq L_t/2. \tag{D.57}$$

Similarly,

$$\|J_t W_{t,\perp}^\top W_{t,\perp} K_t^\top\| \leq 2L_t \tag{D.58}$$

Then,

$$\|K_{t+1} W_{t,\perp}^\top\| \leq \|K_t W_{t,\perp}^\top (I - \eta W_{t,\perp} K_t^\top K_t W_{t,\perp}^\top - \eta \phi_t)\| + \frac{160 \eta L \kappa^2}{\sigma_r} \|J_t W_{t,\perp}^\top W_{t,\perp} K_t^\top\| \cdot \|J_t W_{t,\perp}^\top\|.$$

If $\|J_t W_{t,\perp}^\top\| \leq 10 \kappa \alpha$, we can get

$$\begin{aligned}
\|K_{t+1} W_{t,\perp}^\top\| &\leq \|K_t W_{t,\perp}^\top (I - \eta W_{t,\perp} K_t^\top K_t W_{t,\perp}^\top - \eta \phi_t)\| + \frac{160 \eta L \kappa^2}{\sigma_r} \|J_t W_{t,\perp}^\top W_{t,\perp} K_t^\top\| \cdot \|J_t W_{t,\perp}^\top\| \\
&\leq \|K_t W_{t,\perp}^\top\| \|(I - \eta W_{t,\perp} K_t^\top K_t W_{t,\perp}^\top - \eta \phi_t)\| + \frac{160 \eta L \kappa^2}{\sigma_r} \|J_t W_{t,\perp}^\top W_{t,\perp} K_t^\top\| \cdot \|J_t W_{t,\perp}^\top\| \\
&\leq \|K_t W_{t,\perp}^\top\| \left(1 - \frac{\eta \alpha^2}{8}\right) + \frac{160 \eta L \kappa^2}{\sigma_r} \|J_t W_{t,\perp}^\top\| \|W_{t,\perp} K_t^\top\| \cdot \|J_t W_{t,\perp}^\top\| \\
&\leq \|K_t W_{t,\perp}^\top\| \cdot \left(1 - \frac{\eta \alpha^2}{8}\right) + \frac{160 \eta L \kappa^2}{\sigma_r} 100 \kappa^2 \alpha^2 \|K_t W_{t,\perp}^\top\| \\
&\leq \|K_t W_{t,\perp}^\top\| \cdot \left(1 - \frac{\eta \alpha^2}{16}\right) \\
&\leq \|K_t W_{t,\perp}^\top\| \cdot \left(1 - \frac{\eta \|J_t W_{t,\perp}^\top\|}{1600 \kappa^2}\right)
\end{aligned}$$

(D.59)

(D.60)

by choosing $\delta_{2k+1} \leq \mathcal{O}(\kappa^{-5})$. Now if $\|J_t W_{t,\perp}^\top\| \geq 10 \kappa \alpha$,

$$W_{t,\perp} J_t^\top J_t W_{t,\perp}^\top - W_{t,\perp} K_t^\top K_t W_{t,\perp}^\top - W_{t,\perp} V_t^\top V_t W_{t,\perp}^\top \leq 2\alpha^2 \cdot I$$

$$W_{t,\perp} J_t^\top J_t W_{t,\perp}^\top - W_{t,\perp} K_t^\top K_t W_{t,\perp}^\top \leq 2\alpha^2 \cdot I + W_{t,\perp} (U_t - V_t)^\top (U_t - V_t) W_{t,\perp}^\top$$

Hence,

If $\|J_t W_{t,\perp}^\top\| \geq 10\kappa\alpha$, then

$$
\begin{aligned}
\|J_t W_{t,\perp}\|^2 &= \|W_{t,\perp} J_t^\top J_t W_{t,\perp}^\top\| \\
&\leq \|W_{t,\perp} K_t^\top K_t W_{t,\perp}^\top\| + \left(2\alpha + \frac{8\alpha^2\sqrt{\sigma_1} + 64\delta_{2k+1}\sqrt{\sigma_1}L_t}{\sigma_r}\right)^2 \\
&\leq \|W_{t,\perp} K_t^\top K_t W_{t,\perp}^\top\| + \left(2\alpha + \frac{8\alpha^2\sqrt{\sigma_1} + 64\delta_{2k+1}\sqrt{\sigma_1}\|J_t W_{t,\perp}^\top\| \cdot \sqrt{\sigma_1}}{\sigma_r}\right)^2 \\
&\leq \|W_{t,\perp} K_t^\top K_t W_{t,\perp}^\top\| + (10\alpha + 64\delta_{2k+1}\kappa\|J_t W_{t,\perp}^\top\|)^2 \\
&\leq \|W_{t,\perp} K_t^\top K_t W_{t,\perp}^\top\| + (1/10\kappa + 64\delta_{2k+1}\kappa) \cdot \|J_t W_{t,\perp}^\top\|^2 \\
&\leq \|W_{t,\perp} K_t^\top K_t W_{t,\perp}^\top\| + (1/2) \cdot \|J_t W_{t,\perp}^\top\|^2.
\end{aligned}
$$

Thus, $\|K_t W_{t,\perp}^\top\| \geq \|J_t W_{t,\perp}^\top\|/\sqrt{2} \geq \|J_t W_{t,\perp}^\top\|/2$.

$$
\begin{aligned}
\|K_{t+1} W_{t,\perp}^\top\| &\leq \|K_t W_{t,\perp}^\top(I - \eta W_{t,\perp} K_t^\top K_t W_{t,\perp}^\top - \eta\phi_t)\| + \frac{160\eta L\kappa^2}{\sigma_r}\|J_t W_{t,\perp}^\top W_{t,\perp} K_t^\top\| \cdot \|J_t W_{t,\perp}^\top\| \\
&\leq \|K_t W_{t,\perp}^\top\|\|(I - \eta W_{t,\perp} K_t^\top K_t W_{t,\perp}^\top - \eta\phi_t)\| + \frac{160\eta L\kappa^2}{\sigma_r}\|J_t W_{t,\perp}^\top W_{t,\perp} K_t^\top\| \cdot \|J_t W_{t,\perp}^\top\|
\end{aligned}
$$

Then, if we denote $K' = K_t W_{t,\perp}^\top$, then we know $\|K'(1 - \eta(K')^\top K')\| \leq (1 - \eta\frac{\sigma_1^2(K')}{2})\|K'\|$. Let $K' = A'\Sigma'W'$

$$
\begin{aligned}
\|K'(1 - \eta(K')^\top K')\| &= \|A'\Sigma'W'(I - \eta(W')^\top(\Sigma')^2 W')\| \\
&= \|\Sigma'(I - \eta(\Sigma')^2)\|
\end{aligned}
$$

Let $\Sigma'_{ii} = \zeta_i$ for $i \leq r$, then $\Sigma'(I - \eta(\Sigma')^2)_{ii} = \zeta_i - \eta\zeta_i^3$, then by the fact that $\zeta_1 = \sigma_1(K_t W_{t,\perp}^\top) \leq 1$, we can have $\zeta_1 - \eta\zeta_1^3 = \max_{1 \leq i \leq r} \zeta_i - \eta\zeta_i^3$ and then

$$
\|\Sigma(I - \eta\Sigma^2)\| = (1 - \eta\|K'\|^2)\|K'\|.
$$

Hence,

$$
\begin{aligned}
\|K_{t+1} W_{t,\perp}^\top\| &\leq \|K_t W_{t,\perp}^\top(I - \eta W_{t,\perp} K_t^\top K_t W_{t,\perp}^\top - \eta\phi_t)\| + \frac{160\eta L\kappa^2}{\sigma_r}\|J_t W_{t,\perp}^\top W_{t,\perp} K_t^\top\| \cdot \|J_t W_{t,\perp}^\top\| \\
&\leq \|K_t W_{t,\perp}^\top(I - \eta W_{t,\perp} K_t^\top K_t W_{t,\perp}^\top)\| + \frac{160\eta L\kappa^2}{\sigma_r}\|J_t W_{t,\perp}^\top W_{t,\perp} K_t^\top\| \cdot \|J_t W_{t,\perp}^\top\| \\
&\leq \|K_t W_{t,\perp}^\top\|\left(1 - \eta\frac{\|K_t W_{t,\perp}^\top\|^2}{2}\right) + \frac{160\eta L\kappa^2}{\sigma_r}\|J_t W_{t,\perp}^\top\|\|W_{t,\perp} K_t^\top\| \cdot \|J_t W_{t,\perp}^\top\| \\
&\leq \|K_t W_{t,\perp}^\top\|\left(1 - \eta\frac{\|J_t W_{t,\perp}^\top\|^2}{8}\right) + \frac{160\eta L\kappa^2}{\sigma_r}\|J_t W_{t,\perp}^\top\|\|W_{t,\perp} K_t^\top\| \cdot \|J_t W_{t,\perp}^\top\| \\
&\leq \|K_t W_{t,\perp}^\top\|\left(1 - \eta\frac{\|J_t W_{t,\perp}^\top\|^2}{16}\right) \qquad\qquad\qquad (D.61) \\
&\leq \|K_t W_{t,\perp}^\top\|\left(1 - 4\eta\kappa^2\alpha^2\right). \qquad\qquad\qquad (D.62)
\end{aligned}
$$

The fifth inequality is because $\delta_{2k+1} = O(\kappa^{-4})$. Thus, for all cases, by Eq.(D.59), (D.60), (D.62) and (D.61), we have

$$
\begin{aligned}
\|K_{t+1} W_{t,\perp}^\top\| &\leq \|K_t W_{t,\perp}^\top\| \cdot \min\left\{\left(1 - \frac{\eta\alpha^2}{4}\right), \left(1 - \frac{\eta\|J_t W_{t,\perp}^\top\|^2}{1600\kappa^2}\right)\right\} \\
&\leq \|K_t W_{t,\perp}^\top\| \cdot \left(1 - \frac{\eta\alpha^2}{8}\right) \cdot \left(1 - \frac{\eta\|J_t W_{t,\perp}^\top\|^2}{3200\kappa^2}\right), \qquad (D.63)
\end{aligned}
$$

where we use the inequality $\max\{a, b\} \leq \sqrt{ab}$. Now we prove the following claim:

$$\|K_{t+1}W_{t+1,\perp}^\top\| \leq \|K_{t+1}W_{t,\perp}^\top\| \cdot \left(1 + \mathcal{O}(\eta\delta_{2k+1}\|J_tW_{t,\perp}^\top\|^2/\sigma_r^{3/2})\right). \tag{D.64}$$

First consider the situation that $\|J_tW_{t,\perp}^\top\| \leq 10\kappa\alpha$. We start at these two equalities:

$$K_{t+1} = K_{t+1}W_{t,\perp}^\top W_{t,\perp} + K_{t+1}W_t^\top W_t$$
$$K_{t+1} = K_{t+1}W_{t+1,\perp}^\top W_{t+1,\perp} + K_{t+1}W_{t+1}^\top W_{t+1}.$$

Thus, we have

$$K_{t+1}W_{t,\perp}^\top W_{t,\perp}W_{t+1,\perp}^\top + K_{t+1}W_t^\top W_t W_{t+1,\perp}^\top = K_{t+1}W_{t+1,\perp}^\top$$

Consider

$$
\begin{aligned}
\|W_t W_{t+1,\perp}^\top\| &= \|W_{t+1,\perp}W_t^\top\| \\
&= \|W_{t+1,\perp}U_t^\top(U_tU_t^\top)^{-1/2}\| \\
&= \|W_{t+1,\perp}U_t^\top\|\|(U_tU_t^\top)^{-1/2}\| \\
&\leq \|W_{t+1,\perp}\|\|U_{t+1} - U_t\| \cdot \sigma_r(U)^{-1} \\
&\leq \frac{4\sqrt{\sigma_1}}{\sigma_r} \cdot \eta \cdot (2\sqrt{\sigma_1} \cdot M_t + 2\delta_{2k+1} \cdot (L_t + 3M_t) \cdot 2\sqrt{\sigma_1}) \\
&\leq \frac{4\sqrt{\sigma_1}}{\sigma_r} \cdot \eta \cdot (3\sqrt{\sigma_1} \cdot M_t + 2\delta_{2k+1} \cdot L_t) \\
&\leq \frac{4\sqrt{\sigma_1}}{\sigma_r} \cdot \eta\left(\frac{48L\kappa\sqrt{\sigma_1}}{\sigma_r}\|J_tK_t^\top\| + 2\sqrt{\sigma_1}\delta_{2k+1} \cdot L_t\right) \\
&\leq C\eta(\delta_{2k+1}\kappa^4 + \alpha^2\kappa^2/\sigma_r)\|J_tK_t^\top\|.
\end{aligned}
$$

for some constant $C$. Also, note that $\|F_tG_t^\top - \Sigma\| \leq L_t + 3M_t \leq 4L_t$,

$$
\begin{aligned}
\|K_{t+1}W_t^\top\| &= \|(K_{t+1} - K_t)W_t^\top\| + \|K_tW_t^\top\| \\
&\leq \|\eta K_t(U_t^\top U_t + J_t^\top J_t)W_t^\top\| + \eta\delta_{2k+1} \cdot (4L_t) \cdot 2\sqrt{\sigma_1} + \|K_tW_t^\top\| \\
&\leq \|\eta K_tJ_t^\top J_tW_t^\top\| + 8\sqrt{\sigma_1}\eta\delta_{2k+1} \cdot L_t + \frac{64L\kappa^{3/2}}{\sigma_r^{3/2}}L_t \\
&\leq \eta L_t\|J_tW_t^\top\| + 8\sqrt{\sigma_1}\eta\delta_{2k+1} \cdot L_t + \frac{64L\kappa^{3/2}}{\sigma_r^{3/2}}L_t \\
&\leq L_t \cdot (\eta \cdot \sqrt{\sigma_1} + 8\sqrt{\sigma_1}\eta\delta_{2k+1} + \frac{64L\kappa^{3/2}}{\sigma_r^{3/2}}) \\
&\leq \frac{1}{4\sqrt{\sigma_1}}L_t \\
&\leq \frac{1}{4}\|K_t\|
\end{aligned}
$$

and

$$
\begin{aligned}
\|K_{t+1}W_{t,\perp}^\top\| &\geq \|K_tW_{t,\perp}^\top\| - \|(K_{t+1} - K_t)W_{t,\perp}^\top\| \\
&\geq \frac{1}{2}\|K_t\| - \eta\|K_t(U_t^\top U_t + J_t^\top J_t)W_t^\top\| - 8\sqrt{\sigma_1}\eta\delta_{2k+1} \cdot L_t \\
&\geq \frac{1}{2}\|K_t\| - \eta L_t\|J_tW_t^\top\| - 8\sqrt{\sigma_1}\eta\delta_{2k+1} \cdot L_t \\
&\geq \|K_t\|(\frac{1}{2} - \eta\|J_t\| \cdot \|J_tW_t^\top\| - 8\sqrt{\sigma_1}\eta\delta_{2k+1} \cdot \|J_t\|) \\
&\geq \|K_t\|(\frac{1}{2} - \eta\sigma_1 - 8\eta\delta_{2k+1}\sigma_1) \\
&\geq \frac{1}{4}\|K_t\| \\
&\geq \|K_{t+1}W_t^\top\|
\end{aligned}
$$

Here, we use the fact that $\eta \leq 1/\sigma_1$, $\delta_{2k+1} \leq 1/32$ and $\|J_t\| \leq \sqrt{\sigma_1}$. Hence, we have

$$
\begin{aligned}
\|K_{t+1}W_{t+1,\perp}^\top\| &\leq \|K_{t+1}W_{t,\perp}^\top\|\|W_{t,\perp}W_{t+1,\perp}^\top\| + \|K_{t+1}W_t^\top\|\|W_t W_{t+1,\perp}^\top\| \\
&\leq \|K_{t+1}W_{t,\perp}^\top\| + \|K_{t+1}W_{t,\perp}^\top\| \cdot C\eta(\delta_{2k+1}\kappa^4 + \alpha^2\kappa^2/\sigma_r)L_t \\
&\leq \left(1 + C\eta(\delta_{2k+1}\kappa^4 + \alpha^2\kappa^2/\sigma_r)L_t\right)\|K_{t+1}W_{t,\perp}^\top\| \\
&\leq \left(1 + 2C\eta(\delta_{2k+1}\kappa^4 + \alpha^2\kappa^2/\sigma_r)\|J_t W_{t,\perp}^\top W_{t,\perp}K_t^\top\|\right)\|K_{t+1}W_{t,\perp}^\top\| \\
&\leq \left(1 + 2C\eta(\delta_{2k+1}\kappa^4 + \alpha^2\kappa^2/\sigma_r)\|J_t W_{t,\perp}^\top\|\|W_{t,\perp}K_t^\top\|\right)\|K_{t+1}W_{t,\perp}^\top\|
\end{aligned}
$$

The inequality on the fourth line is because Eq.(D.57).

Note that

$$
W_{t,\perp}J_t^\top J_t W_{t,\perp}^\top - W_{t,\perp}K_t^\top K_t W_{t,\perp}^\top \geq \frac{\alpha^2}{8} \cdot I.
$$

Thus, $\|K_t W_{t,\perp}^\top\| \leq \|J_t W_{t,\perp}^\top\|$ and

$$
\begin{aligned}
\|K_{t+1}W_{t+1,\perp}^\top\| &\leq \left(1 + 2C\eta(\delta_{2k+1}\kappa^4 + \alpha^2\kappa^2/\sigma_r)\|J_t W_{t,\perp}^\top\|\|W_{t,\perp}K_t^\top\|\right)\|K_{t+1}W_{t,\perp}^\top\| \\
&\leq \left(1 + 2C\eta(\delta_{2k+1}\kappa^4 + \alpha^2\kappa^2/\sigma_r)\|J_t W_{t,\perp}^\top\|^2\right)\|K_{t+1}W_{t,\perp}^\top\| \quad \text{(D.65)}
\end{aligned}
$$

By inequalities (D.63) and (D.65), we can get

$$
\begin{aligned}
&\|K_{t+1}W_{t+1,\perp}^\top\| \\
&\leq \left(1 + 2C\eta(\delta_{2k+1}\kappa^4 + \alpha^2\kappa^2/\sigma_r)\|J_t W_{t,\perp}^\top\|^2\right)\|K_{t+1}W_{t,\perp}^\top\| \\
&\leq \left(1 + 2C\eta(\delta_{2k+1}\kappa^4 + \alpha^2\kappa^2/\sigma_r)\|J_t W_{t,\perp}^\top\|^2\right) \cdot \left(1 - \frac{\eta\alpha^2}{8}\right) \cdot \left(1 - \frac{\eta\|J_t W_{t,\perp}^\top\|^2}{3200\kappa^2}\right)\|K_t W_{t,\perp}^\top\| \\
&\leq \left(1 - \frac{\eta\alpha^2}{8}\right)\|K_t W_{t,\perp}^\top\|.
\end{aligned}
$$

The last inequality is because

$$
2C\eta(\delta_{2k+1}\kappa^4 + \alpha^2\kappa^2/\sigma_r)\|J_t W_{t,\perp}^\top\|^2 \leq \frac{\eta\|J_t W_{t,\perp}^\top\|^2}{3200\kappa^2}
$$

by choosing

$$
\delta_{2k+1} = \mathcal{O}(\kappa^{-6}) \tag{D.66}
$$

and

$$
\alpha = \mathcal{O}(\kappa^{-2} \cdot \sqrt{\sigma_r}). \tag{D.67}
$$

Thus, we can prove $\|K_t W_{t,\perp}^\top\|$ decreases at a linear rate.

Now we have completed all the proofs of the induction hypotheses. Hence,

$$
\begin{aligned}
\|F_t G_t^\top - \Sigma\| &\leq 2\|J_t K_t^\top\| \\
&\leq 4\|K_t^\top\| \cdot \sqrt{\sigma_1} \\
&\leq 4\|K_t W_{t,\perp}^\top\|\sqrt{\sigma_1} \\
&\leq 4\|K_t W_{T_2,\perp}^\top\| \cdot \sqrt{\sigma_1}\left(1 - \frac{\eta\alpha^2}{8}\right)^{t-T_2} \\
&\leq 4\|K_{T_2}\| \cdot \sqrt{\sigma_1}\left(1 - \frac{\eta\alpha^2}{8}\right)^{t-T_2} \\
&\leq 2\sigma_1\left(1 - \frac{\eta\alpha^2}{8}\right)^{t-T_2} \quad \text{(D.68)}
\end{aligned}
$$

Now combining three phases (D.25), (D.42) and (D.68), if we denote $t_2^* + T_0 = T' = \widetilde{\mathcal{O}}(1/\eta\sigma_r)$, then for any round $T \geq 4T'$, Phase 1 and Phase 3 will take totally at least $T - T'$ rounds. Now we consider two situations.

**Situation 1:** Phase 1 takes at least $\frac{3(T-T')}{4}$ rounds. Then, by (D.25), suppose Phase 1 starts at $T_0$ rounds and terminates at $T_1$ rounds, we will have

$$
\begin{aligned}
\|F_{T_1} G_{T_1}^\top - \Sigma\| &\leq \frac{\sigma_r^2}{128\alpha^2\kappa} \left(1 - \frac{\eta\sigma_r^2}{64\sigma_1}\right)^{T_1 - T_0} \\
&\leq \frac{\sigma_r^2}{128\alpha^2\kappa} \left(1 - \frac{\eta\sigma_r^2}{64\sigma_1}\right)^{T/2}.
\end{aligned}
\tag{D.69}
$$

The last inequality uses the fact that $T \geq 4T'$ and

$$
T_1 - T_0 \geq \frac{3(T-T')}{4} \geq T/2
$$

Then, by (D.32), (D.30), (D.44) and (D.45), we know that

$$
\begin{aligned}
\|F_T G_T^\top - \Sigma\| &\leq 4\|J_T K_T^\top\| \\
&\leq 4\|J_{T_1} K_{T_1}^\top - \Sigma\| \cdot \left(1 + \frac{\eta\sigma_r^2}{128\sigma_1}\right)^{T - T_1} \\
&\leq 4\|F_{T_1} G_{T_1}^\top - \Sigma\| \cdot \left(1 + \frac{\eta\sigma_r^2}{128\sigma_1}\right)^{T - T_1} \\
&\leq 4\|F_{T_1} G_{T_1}^\top - \Sigma\| \cdot \left(1 + \frac{\eta\sigma_r^2}{128\sigma_1}\right)^{T/2}
\end{aligned}
\tag{D.70}
$$

The last inequality uses the fact that $T_1 - T_0 \geq \frac{3(T-T')}{4} \geq \frac{T}{2}$, which implies that $\frac{T}{2} \geq T - T_1$. Then, combining with (D.69), we can get

$$
\begin{aligned}
\|F_T G_T^\top - \Sigma\| &\leq \frac{\sigma_r^2}{128\alpha^2\kappa} \left(1 - \frac{\eta\sigma_r^2}{64\sigma_1}\right)^{T/2} \cdot \left(1 + \frac{\eta\sigma_r^2}{128\sigma_1}\right)^{T/2} \\
&\leq \frac{\sigma_r^2}{128\alpha^2\kappa} \left(1 - \frac{\eta\sigma_r^2}{128\sigma_1}\right)^{T/2} \\
&\leq \frac{\sigma_r^2}{128\alpha^2\kappa} \left(1 - \frac{\eta\alpha^2}{8}\right)^{T/2}.
\end{aligned}
\tag{D.71}
\tag{D.72}
$$

(D.71) uses the basic inequality $(1 - 2x)(1 + x) \leq (1 - x)$, and (D.72) uses the fact that $\alpha = \mathcal{O}(\kappa^{-2}\sqrt{\sigma_r}) = \mathcal{O}(\sqrt{\kappa\sigma_r})$.

**Situation 2:** Phase 3 takes at least $\frac{T-T'}{4}$ rounds. Then, by (D.68), suppose Phase 3 starts at round $T_2$, we have

$$
\begin{aligned}
\|F_T G_T^\top - \Sigma\| &\leq 2\sigma_1 \left(1 - \frac{\eta\alpha^2}{8}\right)^{t - T_2} \\
&\leq 2\sigma_1 \left(1 - \frac{\eta\alpha^2}{8}\right)^{(T-T')/4} \\
&\leq \frac{\sigma_r^2}{128\alpha^2\kappa} \left(1 - \frac{\eta\alpha^2}{8}\right)^{T/8}.
\end{aligned}
\tag{D.73}
$$

The last inequality uses the fact that $\alpha = \mathcal{O}(\kappa^{-2}\sqrt{\sigma_r}) = \mathcal{O}(\kappa^{-1}\sqrt{\sigma_r})$ and $\frac{T-T'}{4} \geq \frac{T-T/4}{4} \geq T/8$. Thus, by $\|F_T G_T^\top - \Sigma\|^2 \leq n \cdot \|F_T G_T^\top - \Sigma\|^2$, we complete the proof by choosing $4T' = T^{(1)}$ and $c_7 = 1/128^2$.

# E  PROOF OF THEOREM 4.3

By the convergence result in (Soltanolkotabi et al., 2023), the following three conditions hold for $t = T_0$.

$$\max\{\|J_t\|, \|K_t\|\} \leq \mathcal{O}\left(2\alpha + \frac{\delta_{2k+1}\sigma_1^{3/2}\log(\sqrt{\sigma_1}/n\alpha)}{\sigma_r}\right) \tag{E.1}$$

$$\max\{\|U_t\|, \|V_t\|\} \leq 2\sqrt{\sigma_1} \tag{E.2}$$

and

$$\|F_tG_t^T - \Sigma\| \leq \alpha^{1/2}\sigma_1^{3/4} \leq \sigma_r/2. \tag{E.3}$$

Then, we define $M_t = \max\{\|U_tV_t^\top - \Sigma\|, \|U_tK_t^\top\|, \|J_tV_t^\top\|\}$, by the same techniques in Section D.4, if we have

$$\sigma_r^2 M_{t-1}/64\sigma_1 \geq (17\sigma_1\delta_{2k+1} + \alpha^2)\|J_{t-1}K_{t-1}^\top\|, \tag{E.4}$$

we can prove that

$$M_t \leq \left(1 - \frac{\eta\sigma_r^2}{64\sigma_1}\right)M_{t-1}. \tag{E.5}$$

and

$$\max\{\|J_t\|, \|K_t\|\} \leq 2\sqrt{\alpha}\sigma_1^{1/4} + 2C_2\log(\sqrt{\sigma_1}/n\alpha)(\delta_{2k+1}\cdot\kappa^2\sqrt{\sigma_1}) \leq \sqrt{\sigma_1}$$
$$\|F_tG_t^\top - \Sigma\| \leq \sigma_r/2$$
$$\max\{\|U_t\|, \|V_t\|\} \leq 2\sqrt{\sigma_1}.$$

Now note that

$$\|U_{t-1}K_{t-1}^\top\| \geq \lambda_{\min}(U_{t-1})\cdot\|K_{t-1}^\top\| = \sigma_r(U_{t-1})\cdot\|K_{t-1}^\top\| \geq \frac{\sigma_r}{4\sqrt{\sigma_1}}\cdot\|K_{t-1}\|, \tag{E.6}$$

Now since $\delta_{2k+1} = \mathcal{O}(\kappa^{-3})$ and $\alpha = \mathcal{O}(\kappa^{-1}\sqrt{\sigma_r})$ are small parameters, we can derive the $M_t$'s lower bound by

$$M_{t-1} \geq \|U_{t-1}K_{t-1}^\top\|$$
$$\geq \frac{\sigma_r}{4\sqrt{\sigma_1}}\cdot\|K_{t-1}\|$$
$$\geq \frac{\sigma_r}{4\sqrt{\sigma_1}}\|K_{t-1}\|\cdot\frac{\|J_{t-1}\|}{\sqrt{\sigma_1}} \tag{E.7}$$
$$\geq 64\sigma_1\cdot\frac{17\sigma_1\delta_{2k+1} + \alpha^2}{\sigma_r^2}\|J_{t-1}K_{t-1}^\top\|. \tag{E.8}$$

Hence, (E.4) always holds for $t \geq T_0$, and then by (E.5), we will have

$$M_t \leq \left(1 - \frac{\eta\sigma_r^2}{16\sigma_1}\right)^{t-T_0}M_{T_0}$$
$$\leq \left(1 - \frac{\eta\sigma_r^2}{16\sigma_1}\right)^{t-T_0}\|F_{T_0}G_{T_0}^T\|$$
$$\leq \frac{\sigma_r}{2}\cdot\left(1 - \frac{\eta\sigma_r^2}{16\sigma_1}\right)^{t-T_0}.$$

Thus, we can bound the loss by

$$\|F_tG_t^\top - \Sigma\| \leq \|U_tV_t^\top - \Sigma\| + \|J_tV_t^\top\| + \|U_tK_t^\top\| + \|J_tK_t^\top\|$$
$$\leq 3M_t + \|J_tK_t^\top\|$$
$$\leq 3M_t + \mathcal{O}(2\alpha + \delta_{2k+1}\kappa\sqrt{\sigma_1}\log(\sqrt{\sigma_1}/n\alpha)\cdot\frac{4\sqrt{\sigma_1}}{\sigma_r}M_t$$
$$\leq 4M_t \tag{E.9}$$
$$\leq 2\sigma_r\cdot\left(1 - \frac{\eta\sigma_r^2}{64\sigma_1}\right)^{t-T_0}.$$

where Eq.(E.9) uses the fact that $\delta_{2k+1} \leq \mathcal{O}(\kappa^{-2}\log^{-1}(\sqrt{\sigma_1}/n\alpha))$ and $\alpha \leq \mathcal{O}(\sigma_r/\sqrt{\sigma_1})$. Now we can choose $T^{(2)} = 2T_0$, and then by $t - T_0 \geq t/2$ for all $t \geq T^{(2)}$, we have

$$\|F_t G_t^T - \Sigma\|_F^2 \leq n\|F_t G_t^T - \Sigma\|^2 \leq 2n\sigma_r \cdot \left(1 - \frac{\eta\sigma_r^2}{64\sigma_1}\right)^{t-T_0} \leq 2n\sigma_r \cdot \left(1 - \frac{\eta\sigma_r^2}{64\sigma_1}\right)^{t/2}. \tag{E.10}$$

We complete the proof.

## F  PROOF OF THEOREM 5.1

During the proof of Theorem 5.1, we assume $\beta$ satisfy that

$$\max\{c_7\gamma^{1/6}\sigma_1^{1/3}, c\delta_{2k+1}^{1/6}\kappa^{1/6}\sigma_1^{5/12}\} \leq \beta \leq c_8\sqrt{\sigma_r} \tag{F.1}$$

for some large constants $c_7, c$ and small constant $c_8$. In particular, this requirement means that $\gamma \leq \sigma_r/4$. Then, since $\|\mathcal{A}^*\mathcal{A}(\tilde{F}_{T^{(3)}}\tilde{G}_{T^{(3)}}^\top - \Sigma)\| \geq \frac{1}{2}\|\tilde{F}_{T^{(3)}}\tilde{G}_{T^{(3)}}^\top - \Sigma\|$ by RIP property and $\delta_{2k+1} \leq 1/2$, we can further derive $\|F_{T^{(3)}}G_{T^{(3)}}^\top - \Sigma\| = \|\tilde{F}_{T^{(3)}}\tilde{G}_{T^{(3)}}^\top - \Sigma\| \leq \sigma_r/2$.

To guarantee (F.1), we can use choose $\gamma$ to be small enough, i.e., $\gamma \ll \sigma_1\kappa^{-2}$, so that (F.1) holds easily. In the following, we denote $\delta_{2k+1} = \sqrt{2k+1}\delta$.

### F.1  PROOF SKETCH OF THEOREM 5.1

First, suppose we modify the matrix $\widetilde{F}_{T^{(3)}}, \widetilde{G}_{T^{(3)}}$ to $F_{T^{(3)}}$ and $G_{T^{(3)}}$ at $t = T^{(3)}$, then $\|F_{T^{(3)}}\|^2 = \lambda_{\max}((F_{T^{(3)}})^\top F_{T^{(3)}}) = \beta^2$ and $\|U_{T^{(3)}}\|^2 \leq \beta^2$. Also, by $\|\widetilde{F}_{T^{(3)}}\| \leq 2\sqrt{\sigma_1}$, we can get that $\|G_{T^{(3)}}\| \leq \|\widetilde{G}_{T^{(3)}}\| \cdot \frac{\|\widetilde{F}_{T^{(3)}}\|}{\beta} \leq \|\widetilde{G}_{T^{(3)}}\| \cdot \frac{2\sqrt{\sigma_1}}{\beta}$ is still bounded. Similarly, $\|V_{T^{(3)}}\| \leq \|\widetilde{V}_{T^{(3)}}\| \cdot \frac{2\sqrt{\sigma_1}}{\beta}$ and $\|K_{T^{(3)}}\| \leq \|\widetilde{K}_{T^{(3)}}\| \cdot \frac{2\sqrt{\sigma_1}}{\beta}$ is still bounded. With these conditions, define $S_t = \max\{\|U_t K_t^\top\|, \|J_t K_t^\top\|\}$ and $P_t = \max\{\|J_t V_t^\top\|, \|U_t V_t^\top - \Sigma\|\}$. For $\|K_{t+1}\|$, since we can prove $\lambda_{\min}(F_t^\top F_t) \geq \beta^2/2$ for all $t \geq T^{(3)}$ using induction, with the updating rule, we can bound $\|K_{t+1}|$ as the following

$$\|K_{t+1}\| \leq \|K_t\|\|1 - \eta F_t^\top F_t\| + 2\eta\delta_{2k+1} \cdot \|F_t G_t^\top - \Sigma\|\max\{\|U_t\|, \|J_t\|\} \tag{F.2}$$

$$\leq \|K_t\| \cdot \left(1 - \frac{\eta\beta^2}{2}\right) + \left(4\eta\delta_{2k+1}\beta \cdot P_t + 4\beta^2\eta\delta_{2k+1}\|K_t\|\right). \tag{F.3}$$

The first term of (F.3) ensures the linear convergence, and the second term represents the perturbation term. To control the perturbation term, for $P_t$, with more calculation (see details in the rest of the section), we have

$$P_{t+1} \leq \left(1 - \eta\sigma_r^2/8\beta^2\right)P_t + \eta\|K_t\| \cdot \widetilde{\mathcal{O}}\left(\left(\delta_{2k+1}\sigma_1 + \sqrt{\alpha\sigma_1^{7/4}}\right)/\beta\right). \tag{F.4}$$

The last inequality uses the fact that $S_t \leq \|K_t\| \cdot \max\{\|U_t\|, \|J_t\|\} \leq \|K_t\| \cdot \|F_t\| \leq \sqrt{2}\beta \cdot \|K_t\|$.

Combining (F.4) and (F.3), we can show that $P_t + \sqrt{\sigma_1}\|K_t\|$ converges at a linear rate $(1 - \mathcal{O}(\eta\beta^2))$, since the second term of Eq. (F.4) and Eq.(F.3) contain $\delta_{2k+1}$ or $\alpha$, which is relatively small and can be canceled by the first term. Hence, $\|F_t G_t^\top - \Sigma\| \leq 2P_t + 2S_t \leq 2P_t + \sqrt{2}\beta\|K_t\|$ converges at a linear rate.

### F.2  PROOF OF THEOREM 5.1

At time $t \geq T^{(3)}$, we have $\sigma_{\min}(U_{T^{(3)}}V_{T^{(3)}}) \geq \sigma_{\min}(\Sigma) - \|U_{T^{(3)}}V_{T^{(3)}}^\top - \Sigma\| \geq \sigma_r - \alpha^{1/2} \cdot \sigma_1^{3/4} \geq \sigma_r/2$. The last inequality holds because $\alpha = O(\kappa^{-3/2} \cdot \sqrt{\sigma_r})$. Then, given that $\|F_{T^{(3)}}\|^2 = \lambda_{\max}((F_{T^{(3)}})^\top F_{T^{(3)}}) = \beta^2$, we have $\|U_{T^{(3)}}\|^2 \leq \beta^2$. Hence, by $\sigma_1(U) \cdot \sigma_r(V) \geq \sigma_r(UV^\top)$, we have

$$\sigma_r(V_{T^{(3)}}) \geq \frac{\sigma_r(U_{T^{(3)}}V_{T^{(3)}})}{\sigma_1(U_{T^{(3)}})} \geq \frac{\sigma_r}{2\beta}.$$

Also, by $\sigma_1' = \|\widetilde{F}_{T^{(3)}}\| \le 2\sqrt{\sigma_1}$, we can get

$$\|G_{T^{(3)}}\| \le \|\widetilde{G}_{T^{(3)}}\| \|B\Sigma_{inv}^{-1}\| \le \|\widetilde{G}_{T^{(3)}}\| \cdot \frac{\sigma_1'}{\beta} \le \|\widetilde{G}_{T^{(3)}}\| \cdot \frac{2\sqrt{\sigma_1}}{\beta}.$$

Similarly, $\|V_{T^{(3)}}\| \le \|\widetilde{V}_{T^{(3)}}\| \cdot \frac{2\sqrt{\sigma_1}}{\beta}$ and $\|K_{T^{(3)}}\| \le \|\widetilde{K}_{T^{(3)}}\| \cdot \frac{2\sqrt{\sigma_1}}{\beta}$.

Denote $S_t = \max\{\|U_t K_t^\top\|, \|J_t K_t^\top\|\}, P_t = \max\{\|J_t V_t^\top\|, \|U_t V_t^\top - \Sigma\|\}$. Now we prove the following statements by induction:

$$P_{t+1} \le \left(1 - \frac{\eta\sigma_r^2}{8\beta^2}\right) P_t + \eta S_t \cdot \mathcal{O}\left(\frac{\log(\sqrt{\sigma_1}/n\alpha)\delta_{2k+1}\kappa^2\sigma_1^2 + \sqrt{\alpha}\sigma_1^{7/4}}{\beta^2}\right) \tag{F.5}$$

$$\|F_{t+1}G_{t+1}^\top - \Sigma\| \le \frac{\beta^6}{\sigma_1^2}\left(1 - \frac{\eta\beta^2}{2}\right)^{t+1-T^{(3)}} \le \sigma_r/2 \tag{F.6}$$

$$\max\{\|F_{t+1}\|, \|G_{t+1}\|\} \le 4\sigma_1/\beta \tag{F.7}$$

$$\frac{\beta^2}{2}I \le F_{t+1}^\top F_{t+1} \le 2\beta^2 I \tag{F.8}$$

$$\|K_t\| \le \mathcal{O}(2\sqrt{\alpha}\sigma_1^{1/4} + \delta_{2k+1}\log(\sqrt{\sigma_1}/n\alpha) \cdot \kappa^2\sqrt{\sigma_1}) \cdot \frac{2\sqrt{\sigma_1}}{\beta} \tag{F.9}$$

**Proof of Eq.(F.5)**  First, since $\|F_t\|^2 = \lambda_{\max}((F_t)^\top F_t) \le 2\beta^2$, we have $\|U_t\|^2 \le 2\beta^2$. Then, because $\sigma_{\min}(U_t V_t) \ge \sigma_{\min}(\Sigma) - \|U_t V_t^\top - \Sigma\| \ge \sigma_r/2$, by $\sigma_1(U) \cdot \sigma_r(V) \ge \sigma_r(UV^\top)$, we have

$$\sigma_r(V_t) \ge \frac{\sigma_r(U_t V_t)}{\sigma_1(U_t)} \ge \frac{\sigma_r}{2\beta}.$$

we write down the updating rule as

$$
\begin{aligned}
&U_{t+1}V_{t+1}^\top - \Sigma \\
&= (1 - \eta U_t U_t^\top)(U_t V_t^\top - \Sigma)(1 - \eta V_t V_t^\top) - \eta U_t K_t^\top K_t V_t^\top - \eta U_t J_t^\top J_t V_t^\top + B_t
\end{aligned}
$$

where $B_t$ contains the $\mathcal{O}(\eta^2)$ terms and $\mathcal{O}(E_i(F_t G_t^\top - \Sigma))$ terms

$$\|B_t\| \le 4\eta\delta_{2k+1}(F_t G_t^\top - \Sigma)\max\{\|F_t\|^2, \|G_t\|^2\} + \mathcal{O}(\eta^2\|F_t G_t^\top - \Sigma\|^2\max\{\|F_t\|^2, \|G_t\|^2\})$$

Hence, we have

$$
\begin{aligned}
&\|U_{t+1}V_{t+1}^\top - \Sigma\| \\
&\le (1 - \frac{\eta\sigma_r^2}{4\beta^2})\|U_t V_t^\top - \Sigma\| + \eta\|U_t K_t^\top\|\|K_t V_t^\top\| + \eta\|J_t V_t^\top\|\|J_t^\top U_t^\top\| + \|B_t\| \\
&\le (1 - \frac{\eta\sigma_r^2}{4\beta^2})P_t + \eta S_t\|K_t\|\|V_t\| + \eta P_t\|J_t\|\|U_t\| + \|B_t\| \\
&\le (1 - \frac{\eta\sigma_r^2}{4\beta^2})P_t + \eta S_t \cdot \frac{4\sigma_1}{\beta^2} \cdot \mathcal{O}\left(2\sqrt{\alpha}\sigma_1^{1/4} + \delta_{2k+1}\log(\sqrt{\sigma_1}/n\alpha) \cdot \kappa^2\sqrt{\sigma_1}\right) \cdot 2\sqrt{\sigma_1} + \eta P_t\beta \cdot \beta \\
&\quad + 4\eta\delta_{2k+1} \cdot 2(P_t + S_t) \cdot 4\sigma_1 \cdot \frac{4\sigma_1}{\beta^2} + \mathcal{O}(\eta^2(P_t + S_t)^2 \cdot 4\sigma_1 \cdot \frac{4\sigma_1}{\beta^2}) \\
&\le \left(1 - \frac{\eta\sigma_r^2}{8\beta^2}\right) P_t + \eta S_t \cdot \mathcal{O}\left(\frac{\log(\sqrt{\sigma_1}/n\alpha)\delta_{2k+1}\kappa^2\sigma_1^2 + \sqrt{\alpha}\sigma_1^{7/4}}{\beta^2}\right) \tag{F.10}
\end{aligned}
$$

The last inequality uses the fact that

$$\beta^2 = \mathcal{O}(\sigma_r^{1/2})$$
$$\delta_{2k+1} = \mathcal{O}(\kappa^{-2})$$
$$P_t + S_t \le 2\|F_t G_t^\top - \Sigma\| \le \mathcal{O}(\sigma_1^2/\beta^2) \le 1/\eta.$$

Similarly, we have

$$\|J_{t+1}V_{t+1}^\top\|$$
$$\leq \left(1 - \eta J^\top J\right) JV^\top (1 - \eta V^\top V) - \eta JK^\top KV^\top - \eta JU^\top (UV^\top - \Sigma) + C_t$$

where $C_t$ satisfies that

$$\|C_t\| \leq 4\eta\delta_{2k+1}(F_tG_t^\top - \Sigma)\max\{\|F_t\|^2, \|G_t\|^2\} + \mathcal{O}(\eta^2\|F_tG_t^\top - \Sigma\|\max\{\|F_t\|^2, \|G_t\|^2\})$$
$$\leq 4\eta\delta_{2k+1} \cdot 2(P_t + S_t) \cdot \frac{16\sigma_1^2}{\beta^2} + \mathcal{O}(\eta^2(P_t + S_t) \cdot \sigma_1 \cdot \frac{\sigma_1}{\beta^2}).$$

Thus, similar to Eq.(F.10), we have

$$\|J_{t+1}V_{t+1}^\top\| \leq \left(1 - \frac{\eta\sigma_r^2}{8\beta^2}\right) P_t + \eta S_t \cdot \mathcal{O}\left(\frac{\log(\sqrt{\sigma_1}/n\alpha)\delta_{2k+1}\kappa^2\sigma_1^2 + \sqrt{\alpha}\sigma_1^{7/4}}{\beta^2}\right).$$

Hence, we have

$$P_{t+1} \leq \left(1 - \frac{\eta\sigma_r^2}{8\beta^2}\right) P_t + \eta S_t \cdot \mathcal{O}\left(\frac{\log(\sqrt{\sigma_1}/n\alpha)\delta_{2k+1}\kappa^2\sigma_1^2 + \sqrt{\alpha}\sigma_1^{7/4}}{\beta^2}\right).$$

**Proof of Eq.(F.6)**   We have $S_t \leq \|K_t\| \cdot \max\{\|U_t\|, \|J_t\|\} \leq \|K_t\| \cdot \|F_t\| \leq \sqrt{2}\beta \cdot \|K_t\|$. So the inequality above can be rewritten as

$$P_{t+1} \leq \left(1 - \frac{\eta\sigma_r^2}{8\beta^2}\right) P_t + \eta\sqrt{2}\beta \cdot \|K_t\| \cdot \mathcal{O}\left(\frac{\log(\sqrt{\sigma_1}/n\alpha)\delta_{2k+1}\kappa^2\sigma_1^2 + \sqrt{\alpha}\sigma_1^{7/4}}{\beta^2}\right)$$
$$= \left(1 - \frac{\eta\sigma_r^2}{8\beta^2}\right) P_t + \eta\|K_t\| \cdot \mathcal{O}\left(\frac{\log(\sqrt{\sigma_1}/n\alpha)\delta_{2k+1}\kappa^2\sigma_1^2 + \sqrt{\alpha}\sigma_1^{7/4}}{\beta}\right)$$

Also, for $K_{t+1}$, we have

$$\|K_{t+1}\| = \|K_t\|\|(1 - \eta F_t^\top F_t)\| + 2\delta_{2k+1} \cdot \|F_tG_t^\top - \Sigma\|\max\{\|U_t\|, \|J_t\|\}$$
$$\leq \|K_t\|(1 - \frac{\eta\beta^2}{2}) + 2\eta\delta_{2k+1} \cdot (P_t + S_t) \cdot \sqrt{2}\beta$$
$$\leq \|K_t\|(1 - \frac{\eta\beta^2}{2}) + 2\eta\delta_{2k+1} \cdot P_t \cdot \sqrt{2}\beta + 2\eta\delta_{2k+1} \cdot \sqrt{2}\beta\|K_t\| \cdot \sqrt{2}\beta$$
$$= \|K_t\|(1 - \frac{\eta\beta^2}{2}) + 4\eta\delta_{2k+1} \cdot \beta P_t + 4\beta^2\eta\delta_{2k+1} \cdot \|K_t\|$$

Thus, we can get

$$P_{t+1} + \sqrt{\sigma_1}\|K_{t+1}\|$$
$$\leq \max\{1 - \frac{\eta\sigma_r^2}{8\beta^2}, 1 - \frac{\eta\beta^2}{2}\}(P_t + \|K_t\|)$$
$$+ \eta\max\left\{\mathcal{O}\left(\frac{\log(\sqrt{\sigma_1}/n\alpha)\delta_{2k+1}\kappa^2\sigma_1^{3/2} + \sqrt{\alpha}\sigma_1^{5/4}}{c}\right) + 4\beta^2\delta_{2k+1}, 4\beta\sqrt{\sigma_1}\delta_{2k+1}\right\}$$
$$\cdot (P_t + \sqrt{\sigma_1}\|K_t\|)$$
$$\leq (1 - \frac{\eta\beta^2}{4})(P_t + \sqrt{\sigma_1}\|K_t\|).$$

The last inequality uses the fact that $\beta \leq \mathcal{O}(\sigma_r^{1/2})$ and

$$\delta_{2k+1} \leq \mathcal{O}(\beta/\sqrt{\sigma_1}\log(\sqrt{\sigma_1}/n\alpha)). \tag{F.11}$$

Hence,

$$
\begin{aligned}
\|K_t\| &\le (P_{T^{(3)}}/\sqrt{\sigma_1} + \|K_{T^{(3)}}\|) \cdot \left(1 - \frac{\eta\beta^2}{2}\right)^{t-T^{(3)}} \\
&\le \|K_{T^{(3)}}\| + \|F_t G_t^\top - \Sigma\|/\sqrt{\sigma_1} \\
&\le \mathcal{O}(\sqrt{\alpha}\sigma_1^{1/4} + \delta_{2k+1}\log(\sqrt{\sigma_1}/n\alpha) \cdot \kappa^2\sqrt{\sigma_1}) + \alpha^{1/2} \cdot \sigma_1^{1/4} \\
&= \mathcal{O}(\sqrt{\alpha}\sigma_1^{1/4} + \delta_{2k+1}\log(\sqrt{\sigma_1}/n\alpha) \cdot \kappa^2\sqrt{\sigma_1})
\end{aligned}
$$

Hence, $P_t + \sqrt{\sigma_1}\|K_t\|$ is linear convergence. Hence, by $\beta \le \sqrt{\sigma_1}$,

$$
\begin{aligned}
\|F_{t+1}G_{t+1}^\top - \Sigma\| &\le 2P_{t+1} + 2S_{t+1} \\
&\le 2P_{t+1} + \sqrt{2}\beta\|K_{t+1}\| \\
&\le (2 + \sqrt{2}\beta/\sqrt{\sigma_1})(P_{t+1} + \sqrt{\sigma_1}\|K_{t+1}\|) \\
&\le 4(P_{T^{(3)}} + \sqrt{\sigma_1}\|K_{T^{(3)}}\|) \cdot \left(1 - \frac{\eta\beta^2}{2}\right)^{t+1-T^{(3)}}
\end{aligned}
$$

Last, note that by $\beta \ge c_7(\gamma^{1/6}\sigma_1^{1/3})$ and $\beta \ge c\delta_{2k+1}^{1/6}\kappa^{1/6}\sigma_1^{5/12}\log(\sqrt{\sigma_1}/n\alpha)^{1/6}$, by choosing for some constants $c_7$ and $c$, by choosing large $c'$ and $c_7 = 2^6$, we can get

$$
\gamma \le \frac{\beta^6}{2\sigma_1^2}, \quad \sqrt{\sigma_1} \cdot \mathcal{O}(\log(\sqrt{\sigma_1}/n\sqrt{\alpha})\delta_{2k+1} \cdot \sigma_1^{3/2}/\sigma_r) \cdot (2\sqrt{\sigma_1}/\beta) \le \frac{\beta^6}{2\sigma_1^2}
$$

and

$$
P_{T^{(3)}} + \sqrt{\sigma_1}\|K_{T^{(3)}}\| \le \gamma + \sqrt{\sigma_1} \cdot \mathcal{O}(\log(\sqrt{\sigma_1}/n\sqrt{\alpha})\delta_{2k+1} \cdot \sigma_1^{3/2}/\sigma_r) \cdot (2\sqrt{\sigma_1}/\beta) \le \beta^6/\sigma_1^2
$$

we have

$$
\|F_{t+1}G_{t+1}^\top - \Sigma\| \le \left(\frac{\beta^6}{\sigma_1^2}\right)\left(1 - \frac{\eta\beta^2}{2}\right)^{t+1-T^{(3)}} \tag{F.12}
$$

**Proof of Eq.(F.7)**  Note that we have $\max\{\|F_{T^{(3)}}\|, \|G_{T^{(3)}}\|\} \le 4\sqrt{\sigma_1} \cdot \sqrt{\sigma_1}/\beta = 4\sigma_1/\beta$. Now suppose $\max\{\|F_{t'}\|, \|G_{t'}\|\} \le 4\sqrt{\sigma_1} \cdot \sqrt{\sigma_1}/\beta = 4\sigma_1/\beta$ for all $t' \in [T^{(3)}, t]$, then the changement of $F_{t+1}$ and $G_{t+1}$ can be bounded by

$$
\|F_{t+1} - F_{T^{(3)}}\| \le \eta \sum_{t'=T^{(3)}}^{t} 2\|F_{t'}G_{t'} - \Sigma\|\|G_{t'}\| \le \eta \cdot 2 \cdot \left(\frac{\beta^6}{\sigma_1^2} + \frac{\sigma_r}{2}\right) \cdot \frac{2}{\eta\beta^2}\frac{4\sigma_1}{\beta} \le \frac{16\beta^3}{\sigma_1} + \frac{8\sigma_1^2}{\beta^3}
$$

$$
\|G_t - G_{T^{(3)}}\| \le \eta \sum_{t'=T^{(3)}}^{t-1} 2\|F_{t'}G_{t'} - \Sigma\|\|F_{t'}\| \le \frac{16\beta^3}{\sigma_1} + \frac{8\sigma_1^2}{\beta^3}
$$

Then, by the fact that $\beta \le \mathcal{O}(\sigma_1^{-1/2})$, we can show that

$$
\|F_{t+1}\| \le \|F_{T^{(3)}}\| + \|F_{t+1} - F_{T^{(3)}}\| \le \frac{2\sigma_1}{\beta} + \frac{16\beta^3}{\sigma_1} + \frac{8\sigma_1^2}{\beta^3} \le \frac{4\sigma_1}{\beta},
$$

$$
\|G_{t+1}\| \le \|G_{T^{(3)}}\| + \|G_{t+1} - G_{T^{(3)}}\| \le \frac{2\sigma_1}{\beta} + \frac{16c^3}{\sigma_1} + \frac{8\sigma_1^2}{\beta^3} \le \frac{4\sigma_1}{\beta}.
$$

**Proof of Eq.(F.8)**  Moreover, we have

$$
\begin{aligned}
\sigma_k(F_{t+1}) &\ge \sigma_k(F_{T^{(3)}}) - \sigma_{\max}(F_{t+1} - F_{T^{(3)}}) \\
&= \sigma_k(F_{T^{(3)}}) - \|F_{t+1} - F_{T^{(3)}}\| \\
&\ge \beta - \frac{16\beta^3}{\sigma_1} \\
&\ge \beta/\sqrt{2},
\end{aligned}
$$

and

$$\|F_t\| \le \|F_{T^{(3)}}\| + \|F_t - F_{T^{(3)}}\| \le \beta + \frac{16\beta^3}{\sigma_1} \le \sqrt{2}\beta.$$

The last inequality is because $\beta \le \mathcal{O}(\sigma_1^{-1/2})$. Hence, since $F_{t+1} \in \mathbb{R}^{n \times k}$, we have

$$\frac{\beta^2}{2}I \le F_{t+1}^\top F_{t+1} \le 2\beta^2 I \tag{F.13}$$

Thus, we complete the proof.

## G   TECHNICAL LEMMA

### G.1   PROOF OF LEMMA B.1

*Proof.* We only need to prove with high probability,

$$\max_{i,j \in [n]} \cos^2 \theta_{x_j, x_k} \le \frac{c}{\log^2(r\sqrt{\sigma_1}/\alpha)(r\kappa)^2}. \tag{G.1}$$

In fact, since $\cos^2 \theta_{x_j, x_k} = \sin^2(\frac{\pi}{2} - \theta_{x_j, x_k}) \le (\pi/2 - \theta_{x_j, x_k})^2$, we have

$$\mathbb{P}\left[|\pi/2 - \theta_{x_j, x_k}| > \mathcal{O}\left(\frac{\sqrt{c}}{\log(r\sqrt{\sigma_1}/\alpha)r\kappa}\right)\right] \ge \mathbb{P}\left[\cos^2 \theta_{x_j, x_k} > \mathcal{O}\left(\frac{c}{\log^2(r\sqrt{\sigma_1}/\alpha)(r\kappa)^2}\right)\right]. \tag{G.2}$$

Moreover, for any $m > 0$, by Lemma G.1,

$$\mathbb{P}\left[|\pi/2 - \theta_{x_j, x_k}| > m\right] \le \mathcal{O}\left(\frac{(\sin(\frac{\pi}{2} - m))^{k-2}}{1/\sqrt{k-2}}\right) = \mathcal{O}\left(\sqrt{k-2}(\cos m)^{k-2}\right) \tag{G.3}$$

$$\le \mathcal{O}\left(\sqrt{k}(1 - m^2/4)^{k-2}\right) \tag{G.4}$$

$$\le \mathcal{O}\left(\sqrt{k}\exp\left(-\frac{4k}{m^2}\right)\right). \tag{G.5}$$

The second inequality uses the fact that $\cos x \le 1 - x^2/4$. Then, if we choose

$$m = \frac{\sqrt{c}}{\log(r\sqrt{\sigma_1}/\alpha)r\kappa}$$

and let $k \ge 16/m^4 = \frac{16\log^4(r\sqrt{\sigma_1}/\alpha)(r\kappa)^4}{c^2}$, we can have

$$\mathbb{P}\left[\cos^2 \theta_{x_j, x_k} > m^2\right] \le \mathbb{P}\left[|\pi/2 - \theta_{x_j, x_k}| > m\right] \tag{G.6}$$

$$\le \mathcal{O}\left(k\exp\left(-\frac{m^2 k}{4}\right)\right) \tag{G.7}$$

$$\le \mathcal{O}\left(k\exp\left(-\sqrt{k}\right)\right) \tag{G.8}$$

Thus, by taking the union bound over $j, k \in [n]$, there is a constant $c_2$ such that, with probability at least $1 - c_4 n^2 k \exp(-\sqrt{k})$, we have

$$\theta_0 \le \frac{c}{\log^2(r\sqrt{\sigma_1}/\alpha)(r\kappa)^2}. \tag{G.9}$$

$\square$

## G.2  Proof of Lemma B.2

*Proof.* Since $x_i = \alpha/\sqrt{k} \cdot \tilde{x}_i$, where each element in $\tilde{x}_i$ is sampled from $\mathcal{N}(0, 1)$. By Theorem 3.1 in Vershynin (2018), there is a constant $c$ such that

$$\mathbb{P}\left[|\|\tilde{x}_i^0\|_2^2 - k| \geq t\right] \leq 2\exp(-ct) \tag{G.10}$$

Hence, choosing $t = (1 - \frac{1}{\sqrt{2}})k$, we have

$$\mathbb{P}[\|\tilde{x}_i^0\|_2^2 \in [k/\sqrt{2}, \sqrt{2}k]] \leq \mathbb{P}[|\|\tilde{x}_i^0\|_2^2 - k| \geq t] \leq 2\exp(-ct) \leq 2\exp(-ck/4)$$

Hence,

$$\mathbb{P}\left[\|x_i^0\|^2 \in [\alpha^2/2, 2\alpha^2]\right] = \mathbb{P}\left[\|\tilde{x}_i^0\|^2 \in [k/\sqrt{2}, \sqrt{2}k]\right] \leq 2\exp(-ck/4). \tag{G.11}$$

By taking the union bound over $i \in [n]$, we complete the proof. □

**Lemma G.1.** *Assume $x, y \in \mathbb{R}^n$ are two random vectors such that each element is independent and sampled from $\mathcal{N}(0, 1)$, then define $\theta$ as the angle between $x, y$, we have*

$$\mathbb{P}\left(\left|\theta - \frac{\pi}{2}\right| \leq m\right) \leq \frac{3\pi\sqrt{n-2}(\sin(\pi/2 - m))^{n-2}}{4\sqrt{2}}. \tag{G.12}$$

*Proof.* First, it is known that $\frac{x}{\|x\|}$ and $\frac{y}{\|y\|}$ are independent and uniformly distributed over the sphere $\mathbb{S}^{n-1}$. Thus, without loss of generality, we can assume $x$ and $y$ are independent and uniformly distributed over the sphere.

Note that $\theta \in [0, \pi]$, and the CDF of $\theta$ is

$$f(\theta) = \frac{\Gamma(n/2)\sin^{n-2}(\theta)}{\sqrt{\pi}\Gamma(\frac{n-1}{2})} \tag{G.13}$$

Then, we have

$$\mathbb{P}\left(\left|\theta - \frac{\pi}{2}\right| > m\right) = 1 - \frac{\int_{\pi/2-m}^{\pi/2+m} \sin^{n-2}\theta d\theta}{\int_0^\pi \sin^{n-2}\theta d\theta} = \frac{\int_0^{\pi/2-m} \sin^{n-2}\theta d\theta}{\int_0^{\pi/2} \sin^{n-2}\theta d\theta} \tag{G.14}$$

$$\leq \frac{(\pi/2) \cdot \sin^{n-2}(\pi/2 - m)}{\int_0^{\pi/2} \cos^{n-2}\theta d\theta} \tag{G.15}$$

$$\leq \frac{(\pi/2 \cdot (\pi/2 - m)^{n-2})}{\int_0^{\sqrt{2}}(1 - t^2/2)^{n-2}dt} \tag{G.16}$$

$$\leq \frac{(\pi/2) \cdot (\pi/2 - m)^{n-2}}{\frac{2\sqrt{2}}{3\sqrt{n-2}}} \tag{G.17}$$

$$= \frac{3\pi\sqrt{n-2}(\sin(\pi/2 - m))^{n-2}}{4\sqrt{2}}. \tag{G.18}$$

□

**Lemma G.2** (Lemma 7.3 (1) in Stöger & Soltanolkotabi (2021)). *Let $\mathcal{A}$ be a linear measurement operator that satisfies the RIP property of order $2k+1$ with constant $\delta$, then we have for all matrices with rank no more than $2k$*

$$\|(I - \mathcal{A}^*\mathcal{A})(X)\| \leq \sqrt{2k} \cdot \delta\|X\|. \tag{G.19}$$

**Lemma G.3** (Soltanolkotabi et al. (2023)). *There exist parameters $\zeta_0$, $\delta_0, \alpha_0, \eta_0$ such that, if we choose $\alpha \leq \alpha_0$, $F_0 = \alpha \cdot \tilde{F}_0, G_0 = (\alpha/3) \cdot \tilde{G}_0$, where the elements of $\tilde{F}_0, \tilde{G}_0$ is $\mathcal{N}(0, 1/n)$,[6] and*

---

[6]Note that in Soltanolkotabi et al. (2023), the initialization is $F_0 = \alpha \cdot \tilde{F}_0$ and $G_0 = \alpha \cdot \tilde{G}_0$, while Lemma G.3 uses a slightly imbalance initialization. It is easy to show that their techniques also hold with this imbalance initialization.

*suppose that the operator $\mathcal{A}$ defined in Eq.(1.1) satisfies the restricted isometry property of order $2r + 1$ with constant $\delta \leq \delta_0$, then the gradient descent with step size $\eta \leq \eta_0$ will achieve*

$$\|F_t G_t^\top - \Sigma\| \leq \alpha^{3/5} \cdot \sigma_1^{7/10} \tag{G.20}$$

*within $T = \widetilde{\mathcal{O}}(1/\eta\sigma_r)$ rounds with probability at least $1 - \zeta_0$, where $\zeta_0 = c_1 \exp(-c_2 k) + (c_3 \upsilon)^{k-r+1}$ is a small constant. Moreover, during $T$ rounds, we always have*

$$\max\{\|F_t\|, \|G_t\|\} \leq 2\sqrt{\sigma_1}. \tag{G.21}$$

*The parameters $\alpha_0, \delta_0$ and $\eta_0$ are selected by*

$$\alpha_0 = \mathcal{O}\left(\frac{\sqrt{\sigma_1}}{k^5 \max\{2n, k\}^2}\right) \cdot \left(\frac{\sqrt{k} - \sqrt{r-1}}{\kappa^2 \sqrt{\max\{2n, k\}}}\right)^{C\kappa} \tag{G.22}$$

$$\delta_0 \leq \mathcal{O}\left(\frac{1}{\kappa^3 \sqrt{r}}\right) \tag{G.23}$$

$$\eta \leq \mathcal{O}\left(\frac{1}{k^5 \sigma_1} \cdot \frac{1}{\log\left(\frac{2\sqrt{2\sigma_1}}{\upsilon\alpha(\sqrt{k}-\sqrt{r-1})}\right)}\right) \tag{G.24}$$

## H   EXPERIMENT DETAILS

In this section, we provide experimental results to corroborate our theoretical observations.

**Symmetric Lower Bound** In the first experiment, we choose $n = 50, r = 2$, three different $k = 5, 3, 2$ and learning rate $\eta = 0.01$ for the symmetric matrix factorization problem. The results are shown in Figure 1, which matches our $\Omega(1/T^2)$ lower bound result in Theorem 3.1 for the over-parameterized setting, and previous linear convergence results for exact-parameterized setting.

**Asymmetric Matrix Sensing** In the second experiment, we choose configuration $n = 50, k = 4, r = 2$, sample number $m = 700 \approx nk^2$ and learning rate $\eta = 0.2$ for the asymmetric matrix sensing problem. To demonstrate the direct relationship between convergence speed and initialization scale, we conducted multiple trials employing distinct initialization scales $\alpha = 0.5, 0.2, 0.05$. The experimental results in Figure 1.2 offer compelling evidence supporting three key findings:

• The loss exhibits a linear convergence pattern.

• A larger value of $\alpha$ results in faster convergence under the over-parameterization setting

• The convergence rate is not dependent on the initialization scale under the exact-parameterization setting.

These observations highlight the influence of the initialization scale on the algorithm's performance.

In the last experiment, we run our new method with the same $n$ and $r$ but two different $k = 3, 4$. Unlike the vanilla gradient descent, at the midway point of the episode, we applied a transformation to the matrices $F_t$ and $G_t$ as specified by Eq. (5.1). As illustrated in Figure 2(c), it is evident that the rate of loss reduction accelerates after the halfway mark. This compelling observation serves as empirical evidence attesting to the efficacy of our algorithm.

## I   ADDITIONAL EXPERIMENTS

In this section, we provide some additional experiments to further corroborate our theoretical findings.

### I.1   COMPARISONS BETWEEN ASYMMETRIC AND SYMMETRIC MATRIX SENSING

We run both asymmetric and symmetric matrix sensing with $n = 50, n = 4, r = 2$ with sample $m = 1200$ and learning rate $\eta = 0.2$. We run the experiment for three different initialization

scales $\alpha = 0.5, 0.2, 0.05$. The experiment results in Figure I.1 show that asymmetric matrix sensing converges faster than symmetric matrix sensing under different initialization scales.

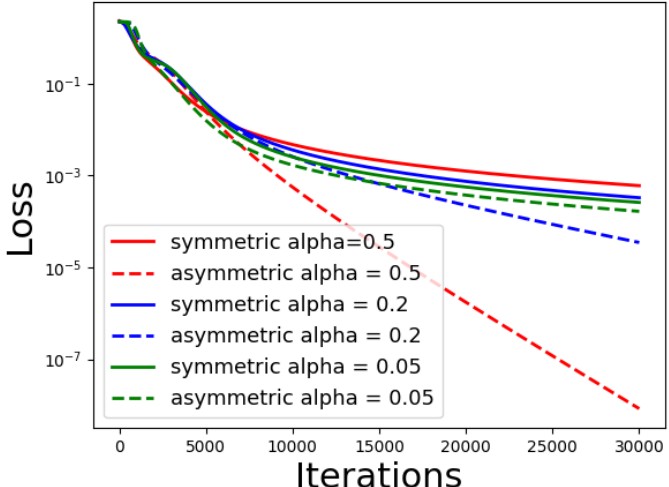

Figure I.1: Comparisons between asymmetric and symmetric matrix sensing with different initialization scales. The dashed line represents the asymmetric matrix sensing, and the solid line represents the symmetric matrix sensing. Different color represents the different initialization scales.

## I.2 Well-Conditioned Case and Ill-Conditioned Case

We run experiments with different conditional numbers of the ground-truth matrix. The conditional number $\kappa$ is selected as $\kappa = 1.5, 3$ and $10$. The minimum eigenvalue is selected by $0.66, 0.33$ and $0.1$ respectively. The experiment results are shown in Figure I.2

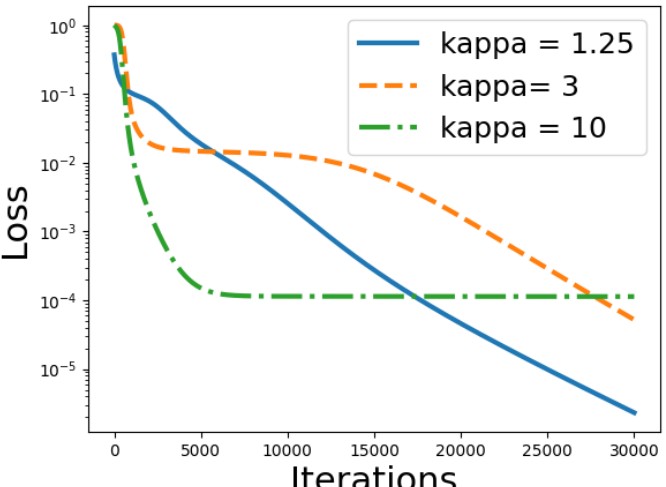

Figure I.2: Comparisons between different conditional numbers

From the experiment results, we can see two phenomena:

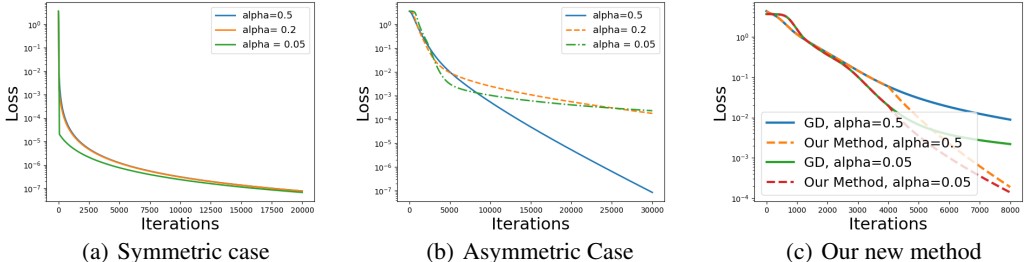

| (a) Symmetric case | (b) Asymmetric Case | (c) Our new method |

Figure I.4: Experiment Results of larger true rank $r = 5$ and over-parameterized rank $k = 10$.

● When the minimum eigenvalue is smaller, the gradient descent will converge to a smaller error at a linear rate. We call this phase the local convergence phase.

● After the local convergence phase, the curve first remains flat and then starts to converge at a linear rate again. We can see that the curve remains flat for a longer time when the matrix is ill-conditioned, i.e. $\kappa$ is larger.

This phenomenon has been theoretically identified by the previous work for the incremental learning (Jiang et al., 2022; Jin et al., 2023), in which GD is shown to sequentially recover singular components of the ground truth from the largest singular value to the smallest singular value.

## I.3 LARGER INITIALIZATION SCALE

We also run experiments with a larger initialization scale $\alpha$. The experiment results are shown in Figure I.3. We find that if $\alpha$ is overly large, i.e. $\alpha = 3$ and $5$, the algorithm actually converges slower and even fails to converge. This is reasonable since there is an upper bound requirement Eq. (4.7) for $\alpha$ in Theorem 4.2.

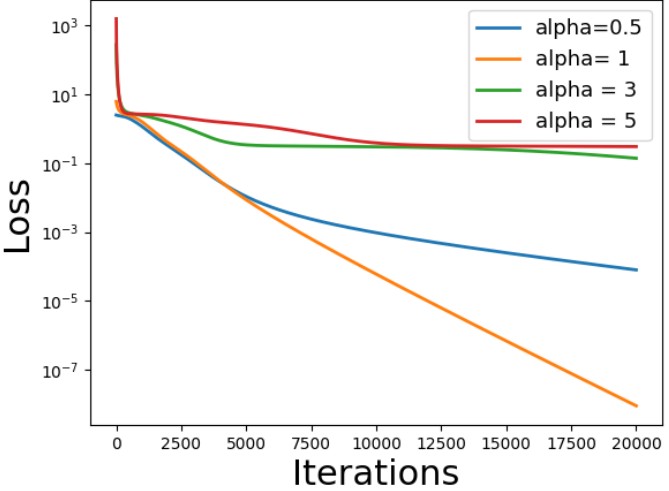

Figure I.3: Comparisons between different large initialization scales

## I.4 LARGER TRUE RANK AND OVER-PARAMETERIZED RANK

We run experiments with larger configurations $n = 50, k = 10$ and $r = 5$. We use $m = 2000$ samples. The experiment results are shown in Figure I.4. We show that similar phenomena of symmetric and asymmetric cases also hold for a larger rank of the true matrix and a larger over-parameterized rank. Moreover, our new method also performs well in this setting.

## I.5 INITIALIZATION PHASE

If we use GD with small initialization, GD always goes through an initialization phase where the loss is relatively flat, and then converges rapidly to a small error. In this subsection, we plot the first 5000 episodes of Figure 2(b). After zooming into the first 5000 iterations, we find the existence of the initialization phase. That is, the loss is rather flat during this phase. We can also see that the initialization phase is longer when $\alpha$ is smaller. The experiment results are shown in Figure I.5.

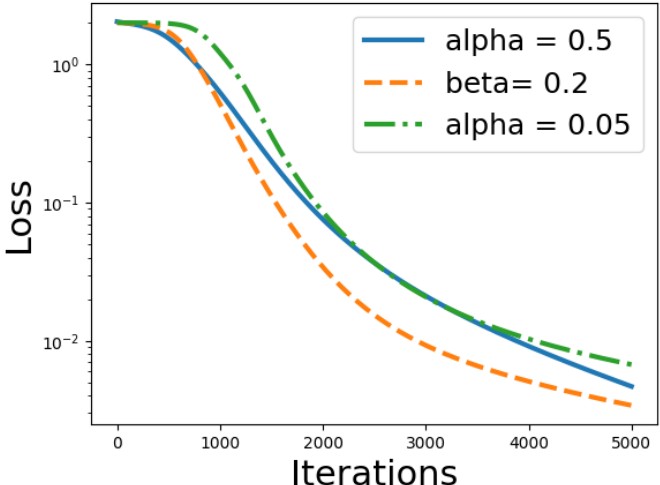

Figure I.5: First 5000 episodes of Figure 2(b)

