\} \le 1/100$, then by $\frac{1-A+B}{1-A-C} \le 1 + 2B + 2C$ for $\max\{A, C\} \le 1/100$,

$$\theta_{ij,t} \cdot \left(\frac{1 - A + B}{1 - A - C}\right)$$

$$\le \theta_{ij,t}(1 + 2B + 2C)$$

$$\le \theta_{ij,t} + 4\eta \sum_{k \neq i,j} \|x_k\|^2 \cdot \sqrt{\theta_{ik,t}\theta_{kj,t}\theta_{ij,t}} + 60n^2\sigma_1^2\eta^2\theta_t\sqrt{\theta_{ij,t}}$$

$$\quad + \theta_t^2\left(8\eta\sigma_1 + 2\eta(2n\alpha^2 + 2r\sigma_1) + 6\eta^2 n^2\sigma_1^2\right)$$

$$\le \theta_{ij,t} + 4\eta(2r\sigma_1 + 2n\alpha^2)\theta_t^{3/2} + 60n^2\sigma_1^2\eta^2\theta_t^{3/2}$$

$$\quad + \theta_t^2\left(8\eta\sigma_1 + 2\eta(2n\alpha^2 + 2r\sigma_1) + 6\eta^2 n^2\sigma_1^2\right)$$

$$\le \theta_{ij,t} + 6\eta(2r\sigma_1 + 2n\alpha^2)\theta_t^{3/2} + 60n^2\sigma_1^2\eta^2\theta_t^{3/2} + 8\eta\sigma_1\theta_t^2 + 6n^2\eta^2\sigma_1^2\theta_t^2)$$

$$\le \theta_{ij,t} + 98\eta \cdot (r\sigma_1\theta_t^{3/2})$$

The last inequality holds by $\alpha \le \sqrt{\sigma_1}/\sqrt{n}$, and $n^2\sigma_1\eta^2 \le \eta$ because $\eta \le \frac{1}{n^2\sigma_1}$.

Hence,

$$\theta_{t+1} \le \theta_t + 98\eta(r\sigma_1)\theta_t^{3/2} \tag{B.41}$$

The Phase 1 terminates when $\|x_i^{T_1}\|^2 \ge \frac{3\sigma_i}{4}$. Since $\|x_i^0\|^2 \ge \alpha^2/2$ and

$$\|x_i^{t+1}\|^2 \ge (1 + \eta\sigma_i/4)\|x_i^t\|^2, \tag{B.42}$$

there is a constant $C_3$ such that $T_1 \le C_1(\log(\sqrt{\sigma_1}/\alpha)/\eta\sigma_i)$. Hence, before round $T_1$,

$$\theta_{T_1} \le \theta_0 + 98\eta T_1 \cdot r\sigma_1 \cdot (2\theta_0)^{3/2} \le \theta_0 + 98C_1 r\kappa(2\theta_0)^{3/2}\log(\sqrt{\sigma_1}/\alpha) \le 2\theta_0.$$

This is because

$$\theta_0 = \mathcal{O}((\log^2(r\sqrt{\sigma_1}/\alpha)(r\kappa))^2)$$

by Lemma B.1 and choosing $k \ge c_2((r\kappa)^2\log(r\sqrt{\sigma_1/\alpha}))^4$ for large enough $c_2$ □

## B.3 PHASE 2

Denote $\theta_t^U = \max_{\min\{i,j\} \le r} \theta_{ij,t}$. In this phase, we prove that $\theta_t^U$ is linear convergence, and the convergence rate of the loss is at least $\Omega(1/T^2)$. To be more specific, we will show that

$$\theta_{t+1}^U \le \theta_t^U \cdot (1 - \eta \cdot \sigma_r/4) \le \theta_t^U \tag{B.43}$$

$$\frac{\theta_{t+1}^U}{\sum_{i>r} \|x_i^{t+1}\|^2} \le \frac{\theta_t^U}{\sum_{i>r} \|x_i^t\|^2} \cdot \left(1 - \frac{\eta\sigma_r}{8}\right) \tag{B.44}$$

$$|\|x_i^t\|^2 - \sigma_i| \le \frac{1}{4}\sigma_i \quad (i \le r) \tag{B.45}$$

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

$$\geq \sigma_r(U) \cdot \|K_t W_t^\top\| \geq \frac{\|K_t W_t^\top\|\sigma_r}{4\sqrt{\sigma_1}} \geq \frac{\|K_t W_t^\top\|\sqrt{\sigma_r}}{4\sqrt{\kappa}}.$$

Hence,

$$\|K_t W_t^\top\| \leq 4\sqrt{\kappa}M/\sqrt{\sigma_r} \leq \frac{64\sigma_1 L\sqrt{\kappa}}{\sigma_r^{5/2}}\|J_t K_t^\top\| \leq \frac{32L\kappa^{3/2}}{\sigma_r^{3/2}}\|K_t\| \cdot \sqrt{\sigma_1} \leq \frac{32L\kappa^2}{\sigma_r}\|K_t\|. \tag{D.50}$$

Thus,

$$\|K_t\| \leq \|K_t W_{t,\perp}^\top\| + \|K_t W_t^\top\|$$
$$\leq \|K_t W_{t,\perp}^\top\| + \frac{64L\kappa}{\sigma_r}\|K_t\|$$
$$\leq \|K_t W_{t,\perp}^\top\| + \frac{1}{2}\|K_t\|.$$

The last inequality uses the fact that $\delta_{2k+1} = \mathcal{O}(\sigma_r^3/\sigma_1^3)$ Hence, $\|K_t W_{t,\perp}^\top\| \geq \|K_t\|/2$, and (D.48) holds during Phase 3.

**Proof of Eq.(D.47)** To prove the (D.47), by the induction hypothesis of Eq.(D.49), note that

$$\|\Delta_t - \Delta_{T_2}\| \le 2\eta^2 \cdot \sum_{t'=T_2}^{t-1} \|F_{t'}G_{t'}^\top - \Sigma\|^2 4\sigma_1$$

$$\le 2\eta^2 \sum_{t'=T_2}^{t-1} 16\sigma_1 \|J_{t'}K_{t'}^\top\|^2$$

$$\le 64\sigma_1\eta^2 \cdot \sum_{t'=T_2}^{\infty} \|J_{t'}\|^2 \|K_{t'}W_{t',\perp}^\top\|^2$$

$$\le 64\sigma_1 \cdot \eta^2 \left( \sigma_1 \cdot \|K_{T_2}W_{T_2,\perp}^\top\|^2 \cdot \frac{8}{\eta\alpha^2} \right) \quad \text{(D.51)}$$

$$\le \frac{512\eta\sigma_1^2}{\alpha^2} \cdot \|K_{T_2}\|^2$$

$$\le \frac{128\eta\sigma_1^2}{\alpha^2} \cdot \sigma_1$$

$$\le \alpha^2/16.$$

The Eq.(D.51) holds by the sum of geometric series. The last inequality holds by $\eta \le \mathcal{O}(\alpha^4/\sigma_1^3)$. Then, we have

$$\|\Delta_t\| \le \|\Delta_{T_2}\| + \|\Delta_t - \Delta_{T_2}\| \le \frac{29\alpha^2}{16} + \frac{\alpha^2}{16} \le 2\alpha^2.$$

$$\lambda_{\min}(\Delta_t) \ge \lambda_{\min}(\Delta_{T_2}) - \|\Delta_t - \Delta_{T_2}\| \ge \frac{3\alpha^2}{16} - \frac{\alpha^2}{16} = \frac{\alpha^2}{8}.$$

Hence, (D.47) holds during Phase 3.

**Proof of Eq.(D.43)** To prove the (D.43), note that

$$\|K_t\| \le 2\|K_tW_{t,\perp}^\top\| \le 2\|K_{T_2}W_{T_2,\perp}^\top\| \le 2\|K_{T_2}\| \le \mathcal{O}(\delta_{2k+1}\log(\sqrt{\sigma_1}/n\alpha) \cdot \sigma_1^{3/2}/\sigma_r). \quad \text{(D.52)}$$

On the other hand, by $\Delta_t \le 2\alpha^2 I$, we have

$$W_{t,\perp}J_t^\top J_t W_{t,\perp}^\top - W_{t,\perp}K_t^\top K_t W_{t,\perp}^\top - W_{t,\perp}V_t^\top V_t W_{t,\perp}^\top \le 2\alpha^2 \cdot I.$$

Hence, denote $L_t = \|J_tK_t^\top\| \le \sigma_1/4$,

$$W_{t,\perp}J_t^\top J_t W_{t,\perp}^\top \le 2\alpha^2 I + W_{t,\perp}K_t^\top K_t W_{t,\perp}^\top + W_{t,\perp}V_t^\top V_t W_{t,\perp}^\top$$

$$= 2\alpha^2 I + W_{t,\perp}K_t^\top K_t W_{t,\perp}^\top + W_{t,\perp}(V_t - U_t)^\top (V_t - U_t)W_{t,\perp}^\top$$

$$\le 2\alpha^2 I + W_{t,\perp}K_t^\top K_t W_{t,\perp}^\top + \left( \frac{8\alpha^2\sqrt{\sigma_1} + 64\delta_{2k+1}\sqrt{\sigma_1}L_t}{\sigma_r} \right)^2 \cdot I$$

$$= W_{t,\perp}K_t^\top K_t W_{t,\perp}^\top + \left( 2\alpha + \frac{8\alpha^2\sqrt{\sigma_1} + 64\delta_{2k+1}\sqrt{\sigma_1}L_t}{\sigma_r} \right)^2 I. \quad \text{(D.53)}$$

Also, by inequality (D.53), we have

$$\|J_tW_{t,\perp}^\top\| - \|K_tW_{t,\perp}^\top\| \le \frac{\|J_tW_{t,\perp}^\top\|^2 - \|K_tW_{t,\perp}^\top\|^2}{\|J_tW_{t,\perp}^\top\| + \|K_tW_{t,\perp}^\top\|}$$