# OpenReview forum: "How Over-Parameterization Slows Down Gradient Descent in Matrix Sensing: The Curses of Symmetry and Initialization"
_ICLR.cc/2024/Conference — ICLR 2024 spotlight_

### Official Review · Reviewer_XP68 · 2023-10-29

**Soundness:** 3 good
**Presentation:** 2 fair
**Contribution:** 3 good
**Rating:** 8
**Confidence:** 4

**Summary:**

This paper studies the matrix sensing problem where one observes y_i = <A_i, M^*> and A_i, and aims to estimate M^*.
This problem makes sense for either symmetric or asymmetric matrices.
The most significant contribution is that this paper unveils a surprising phenomenon that even for the symmetric version of the problem, introducing asymmetry in the initialization and the parametrization produces qualitatively faster convergence rate.

**Strengths:**

The matrix sensing problem requires no more motivation (at least to me) and the results in this paper bring further insights to this classical problem.
The paper is reasonably well-written and the results are definitely sufficiently interesting for ICLR.
The punchlines (the fact that asymmetry helps and why/how this is the case) are clearly addressed within the first 4 pages.
Sec 4.1 is pedagogically helpful.
Several versions of the problems are treated to a reasonably systematic extent.

**Weaknesses:**

I don't see major weaknesses.
Please see technical comments below.

**Questions:**

1. All results only assume RIP for A_i. If A_i are i.i.d. Gaussian matrices, is it possible to derive sharper or even asymptotically exact (in the sense of e.g. https://arxiv.org/abs/2207.09660) results?

2. Could the authors comment on how crucially the results rely on the "linearity" of the problem? Does it make sense to consider a "generalized" matrix sensing problem in which y_i = phi(<A_i, M^*>) for some non-linearity phi? This is somewhat motivated by other models with similar structures such as generalized linear models or single-index models. I guess the information exponent of phi or something like that will play a role in the convergence rate.

3. In Sec 5, an accelerated method is proposed. In particular, step (5.1) should be executed once the iterates are sufficiently close to the optimum. But in practice, how can one verify this neighborhood condition? Note that Sigma is unknown. Please let me know if I missed something simple here.

4. It seems that both the model and the algorithms are deterministic. What happens if the observations are noisy?

5. It's claimed on top of page 6 that the results easily extend to the rectangular case. Could the authors state such results formally (even without formal proofs)? I'm curious to see how the results depend on the aspect ratio n_2 / n_1. In fact, if the matrices are extremely rectangular (e.g. n_2 / n_1 is growing or decaying), I actually doubt if such extensions are so straightforward. Thanks in advance for the clarification.

6. Lemma G.1 assumes x, y are "random vectors". Are they actually independent and uniform over the sphere? For generic joint distribution, not much can be said about their angle. Please make the statement more precise.

Minor notational/grammatical issues.
1. The ground truth is interchangeably denoted by M^* or M^\star. I suggest stick to M^\star to avoid conflict with adjoint operator.

2. In the title of Sec 1.2, where is the word "symmetric" repeated twice?

3. Statement of Theorem 1.3: t-the iteration --> t-th iteration.

3. Page 4: which we require it to be small --> which we require to be small.

4. Description of Table 1: by "row" I think the authors meant "column".

5. Right after equation (2.2): definition of A should be A^*.

---

> ### Author Response · Authors · 2023-11-19
> **Response to the Reviewer XP68**
>
> > 1. All results only assume RIP for A_i. If A_i are i.i.d. Gaussian matrices, is it possible to derive sharper or even asymptotically exact results?
>
>
>
> Since we only use one derived property of RIP, (G.19) in Lemma G.2, the sample complexity might be improved if we use particular properties of Gaussian matrices to prove (G.19). It might be possible to further derive a sharper or even asympototically exact results, but it requires a different set of techniques rather than standard RIP technique in matrix sensing.
>
> > 2. Could the authors comment on how crucially the results rely on the "linearity" of the problem? Does it make sense to consider a "generalized" matrix sensing problem in which $y_i = \phi(<A_i, M^*>)$ for some non-linearity $\phi$?
>
> This is a good question. In this paper, our analysis heavily relies on the linearity of the problem: We utilize a key property that the imbalanced term $F_t^TF_t-G_t^TG_t$ keeps large during the convergence of the gradient descent. This property is derived by the updating rule that the change is $O(\eta^2)$.  However, when the problem is nonlinear, this property does not hold and it seems hard to apply a similar technique. We leave it as a future work.
>
> > 3. In Sec 5, an accelerated method is proposed. In particular, step (5.1) should be executed once the iterates are sufficiently close to the optimum. But in practice, how can one verify this neighborhood condition? Note that Sigma is unknown.
>
>
> In practice, since we can get an estimate of the loss $F_tG_t^T-\Sigma$ by calculating $\mathcal{A}^*(y) = \mathcal{A}(F_tG_t^T-\Sigma)$, we can set a threshold and apply the modification step when the loss first becomes less than the threshold. We can also apply the modification at the half of the process like the experiment in our paper.
>
> > 4. It seems that both the model and the algorithms are deterministic. What happens if the observations are noisy?
>
> Since our paper mainly focuses on the exact global convergence result, we do not consider the noisy setting. The noisy setting induces an additional error term in the gradient.     We believe that when the variance of the noise is small, our result can be extended to the noisy setting, and the gradient descent finally converges to a statistical error dependent on the noise.
>
> > 5. It's claimed on top of page 6 that the results easily extend to the rectangular case. Could the authors state such results formally (even without formal proofs)? I'm curious to see how the results depend on the aspect ratio n_2 / n_1.
>
> Assume that $F \in \mathbb{R}^{n_1\times k}$ and $G \in \mathbb{R}^{n_2\times k}$. Since we do not use the property that $F$ and $G$ are square matrices, we can replace $n$ by $\max\{n_1,n_2\}$ among all the proof and main theorem of Theorem 4.2, 4.3 and 5.1.
>
>
>
> > 6. Lemma G.1 assumes x, y are "random vectors". Are they actually independent and uniform over the sphere? For generic joint distribution, not much can be said about their angle. Please make the statement more precise.
>
> Thanks for catching it. Indeed, we assume every element of the random vector $x,y$ is independent and Gaussian. We have updated our paper accordingly.
>
>
> > 7. typos
>
> Thanks for pointing out the typos. We have fixed them in our revised version.
>
> *References:*
>
> [1]. Tian Ye, Simon S. Du.  Global convergence of gradient descent for asymmetric low-rank matrix factorization. NeurIPS 2021.
>
> [2]. Jiang et al. Algorithmic Regularization in Model-free Overparametrized Asymmetric Matrix Factorization. SIAM 2023.
>
> [3].  Soltanolkotabi et al. Implicit Balancing and Regularization:
> Generalization and Convergence Guarantees for Overparameterized Asymmetric Matrix Sensing. COLT 2023.
>
> [4]. Zhuo et al. On the computational and statistical complexity of over-parameterized matrix sensing. preprint 2021.

---

> > ### Comment · Reviewer_XP68 · 2023-11-21
> > **Response**
> >
> > Thanks to the authors for addressing my questions to a reasonably satisfactory extent.
> > My evaluation remains unchanged.

---

### Official Review · Reviewer_UZwv · 2023-10-31

**Soundness:** 3 good
**Presentation:** 3 good
**Contribution:** 3 good
**Rating:** 8
**Confidence:** 4

**Summary:**

In this work, the authors provided the analysis on the different convergence rates when exact-parameterization or over-parameterization are used. They also proposed a new algorithm to avoid the dependence of the convergence rate on the initialization rate for the asymmetric and over-parameterized case.

**Strengths:**

The results of this paper are novel and should be interesting to audiences in optimization and machine learning fields. The theory provides an explanation for the slow-down of GD in the over-parameterized case, and the paper offered a partial solution to this problem. However, due to the time limit, I cannot check the appendix. So I am not sure about the correctness of the results in this work.

**Weaknesses:**

I can only see a few minor problems with the presentation. For example, the requirement on the sample complexity can be briefly discussed when the informal results are introduced.

**Questions:**

(1) Theorem 1.1: it would be better to say that each entry of X is independently initialized with Gaussian random variable with variance \alpha^2. Similar comment applies to other theorems.

(2) In Section 1, I think the authors did not mention any requirements on the sample size m. It might be better to briefly mention the requirement on the sample complexity or the RIP constant in Section 1.

(3) For the asymmetric case, I think most convergence results require a regularization term \|F^TF - G^TG\|_F^2 to penalize the imbalance between F and G. It would be better to mention the intuition why the regularization term is not required in this work.

(4) After Theorem 1.3: I think it should be "Comparing Theorem 1.3 and Theorem 1.1".

(5) Section 1.3: It might be better to also mention the current state-of-the-art results on landscape analysis:

Zhang, H., Bi, Y., & Lavaei, J. (2021). General low-rank matrix optimization: Geometric analysis and sharper bounds. Advances in Neural Information Processing Systems, 34, 27369-27380.

Bi, Y., Zhang, H., & Lavaei, J. (2022, June). Local and global linear convergence of general low-rank matrix recovery problems. In Proceedings of the AAAI Conference on Artificial Intelligence (Vol. 36, No. 9, pp. 10129-10137).


(6) Section 2: "Asymmetric Matrix Sensing"

(7) Theorem 3.1: it seems that the "ultimate error" does not appear in Section 3.1.

(8) Also, it might be better to mention that the over-parameterization size k depends on \alpha and briefly explain what happens if the size k is smaller than this threshold.

(9) In (3.3a), I think T should be T^{(0)}?

(10) Below Theorem 3.1: For the inequality \|X_tX_t^T - \Sigma\|_F^2 \geq A_t / n, I wonder if it can be improved to \|X_tX_t^T - \Sigma\|_F^2 \geq A_t?

(11) I wonder if there is a reason that initialization scales are chosen as \alpha and \alpha/3? Would it be possible to use, for example, \alpha and \alpha / 10 to achieve a better convergence rate?

---

> ### Author Response · Authors · 2023-11-19
> **Response to the Reviewer UZwv**
>
> Thanks for the detailed response! Please find our response to your questions below.
>
> > 1. Theorem 1.1: it would be better to say that each entry of X is independently initialized with Gaussian random variable with variance \alpha^2. A similar comment applies to other theorems.
>
> Thanks for your reminder. We have updated our paper accordingly.
>
> > 2. In Section 1, I think the authors did not mention any requirements on the sample size m. It might be better to briefly mention the requirement on the sample complexity or the RIP constant in Section 1.
>
> Below the Definition 2.1, we state that $m = \widetilde{\Omega}(nk/\delta^2)$, where $\delta$ is the RIP error parameter. Combining with the constraint of $\delta$ in Equation 4.8, we can get $m = \widetilde{\Omega}(nk^2r)$ is needed, where we ignore the polynomial terms of $\kappa$. We have added this argument in Sections 1 and 2.
>
> > 3. For the asymmetric case, I think most convergence results require a regularization term $\|F^TF - G^TG\|_F^2$ to penalize the imbalance between F and G. It would be better to mention the intuition why the regularization term is not required in this work.
>
>
> Recently, the paper [1] provides the global convergence result for the gradient descent without the regularized term. They mainly focus on the exact-parameterized case, while the over-parameterized case remains open. Some other recent works [2], [3] also provide the convergence guarantee of the gradient descent without the regularization term. However, their convergence is not exact, meaning they only deal with a finite number of iterations.
>
> The intuition why the regularization term is not required is because the change of the  $||F^TF-G^TG||\_F^2$ is actually $O(\eta^2)$, i.e. $||(F_{t+1}^TF_{t+1} - G_{t+1}^TG_{t+1})-(F_t^TF_t-G_t^TG_t)|| = O(\eta^2)$. Thus when $\eta$ is small, we can prove that $F_t^TF_t-G_t^TG_t$ will not diverge. Hence, the two matrices $F$ and $G$ will not diverge and be too imbalanced.
>
> > 4. mention the two current state-of-the-art results on landscape analysis
>
> Thanks for pointing out these works. We have mentioned them in the related work section of our revised version.  These two works provide an analysis of the global geometry of general low-rank minimization problems via the Burer-Monterio factorization approach for the exact-parameterized case ($k=r$). The paper [1] obtains less conservative conditions for guaranteeing the non-existence of spurious second-order critical points and the strict saddle property, for both symmetric and asymmetric low-rank minimization problems.  The paper [2] analyzes the gradient descent for the symmetric case and asymmetric case with a regularized loss. They provide the local convergence result using PL inequality, and show the global convergence for the perturbed gradient descent. Compared to them, our paper tends to focus on the over-parameterized case and the vanilla gradient descent.
>
> > 5. Theorem 3.1: it seems that the "ultimate error" does not appear in Section 3.1.
>
>
> In the previous paper for the symmetric case [4], they prove that the gradient descent will linearly converge to a small error that is dependent on the initialization scale. We want to state that the "ultimate error" refers to this small error. Indeed, note that our lower bound in Equation (3.2) also has a $\alpha^4$. So it is also consistent with the phenomenon that the gradient descent first quickly converges to a small error (which we call "ultimate error") that depends on the initialization scale, with a linear convergence rate. Theorem 3.1 characterizes the behavior of the gradient descent after this quick convergence: The convergence speed becomes $\Omega(1/T^2)$ after the GD converges to a small error linearly.
>
>
>
>
> > 6.  Also, it might be better to mention that the over-parameterization size k depends on $\alpha$ and briefly explain what happens if the size k is smaller than this threshold.
>
> The over-parameterization size $k$ is subject to a lower bound requirement in (3.1) that depends on $\alpha$. However, since $\alpha$ appears in the logarithmic term, this requirement is not overly restrictive. We posit that (3.1) may be a result of a proof artifact, and the lower bound of $\Omega(1/T^2)$ still holds even when $k$ is smaller than this threshold.

---

> ### Author Response · Authors · 2023-11-19
> **Response to the Reviewer UZwv**
>
> > 7. Below Theorem 3.1: For the inequality $\|X_tX_t^T - \Sigma\|_F^2 \geq A_t / n$, I wonder if it can be improved to $\|X_tX_t^T - \Sigma\|_F^2 \geq A_t$?
>
>  We find that the result can be improved. We also need to mention that there is a typo in this paragraph: Equation (3.3) implies $A_t = O(\alpha^2/t)$. Then since $(X_tX_t^T-\Sigma)\_{ii} = \|x_i\|^2$, we can have $||X_tX_t^T-\Sigma||\_F^2 \ge \sum_{i>r}(X_tX_t^T-\Sigma)\_{ii}^2 = \sum_{i>r}\|x_i\|^4\ge A_t^2/n,$ where the last inequality uses the Cauchy's inequality.  Thus the $n^{-2}$ in the final result (Equation (3.2)) can be improved to $n^{-1}$.
>  However, We do not find a way to directly prove $\|X_tX_t^T-\Sigma\|_F^2 \ge A_t^2$.
>
>  > 8.  I wonder if there is a reason that initialization scales are chosen as \alpha and \alpha/3? Would it be possible to use, for example, \alpha and \alpha / 10 to achieve a better convergence rate?
>
> The selection of the more imbalanced initialization $\alpha$ and $\alpha/10$ can make two matrices more imbalanced, thus leading to a faster convergence rate after the gradient descent converges to a local point (Section D.4, D.5 and D.6). However, this might result in a slower convergence rate and impose more stringent requirements on the parameters (up to a constant) at the beginning of the proof, as outlined in Theorem 1 in [3], a pivotal component employed in Section D.3 as the foundation of the proof. To be more specific, a more imbalanced initialization may lead to a looser constant in Lemma B.7 and condition (23) in [3].
>
>
> We also want to highlight that a more imbalanced initialization also gives a guarantee $\|\Delta_0\| = O(\alpha^2)$ where $\Delta_0 = F_0^TF_0 - G_0^TG_0$. Thus all the changes are only up to a constant.
>
> > 9. typos
>
> Thanks for your reminder. We have fixed them in our revised version.
>
>
> *References:*
>
> [1]. Tian Ye, Simon S. Du.  Global convergence of gradient descent for asymmetric low-rank matrix factorization. NeurIPS 2021.
>
> [2]. Jiang et al. Algorithmic Regularization in Model-free Overparametrized Asymmetric Matrix Factorization. SIAM 2023.
>
> [3].  Soltanolkotabi et al. Implicit Balancing and Regularization:
> Generalization and Convergence Guarantees for Overparameterized Asymmetric Matrix Sensing. COLT 2023.
>
> [4]. Stoger and Soltanolkotabi. Small random initialization is akin to spectral learning: Optimization and generalization guarantees for overparameterized low-rank matrix reconstruction. NeurIPS 2021.

---

> > ### Comment · Reviewer_UZwv · 2023-11-21
> > **Response to authors' rebuttal**
> >
> > I would like to thank the authors for their detailed response! I am happy to increase my score.

---

### Official Review · Reviewer_Z4Yg · 2023-10-31

**Soundness:** 3 good
**Presentation:** 4 excellent
**Contribution:** 3 good
**Rating:** 8
**Confidence:** 4

**Summary:**

The reviewed paper is a theoretical investigation of the convergence properties of gradient-descent, and other first-order based methods, for over-parameterized matrix factorization/sensing for symmetric matrices. The specific focus is on the role of using **symmetric** versus **general** Burer-Monteiro factorization as parameterization and how it effects the convergence properties. The unexpected result is that the *symmetricity* versus *imbalance* plays a significant role.

The main "positive" result states that the over-parameterized gradient descent on $FG^T$ factorization is able to achieve linear convergence when the two components are imbalanced in the sense of the spectrum of $\Delta = F^\top F - G^\top G$, and the specific convergence rate depends on this imbalance. The main "negative" result shows that there will always exist a positive measure of cases when symmetric parametrization $FF^\top$ cannot have faster than sublinear convergence.

The work provides simple, but well explained numerical examples of small matrix sizes ($50 \times 50, \mathrm{rank} =3$) that clearly demonstrate this phenomenon.

The proofs take more than 30 pages in the appendices, they are technically involved and not easy to check in their entirety, but at first sight the result seems correct.

**Strengths:**

I believe this paper has several very strong points:
* It presents a novel and surprising result
* It gives rigorous proofs for the two main statements which together describe a very interesting behaviour
* The numerical examples corroborate the proven theory
* The paper is very clearly written, the structure and main message is clear (although the theorems themselves can be a bit complicated to interpret)
* It gives a very good comparison with existing literature

**Weaknesses:**

There is not much that I would consider a weakness to this paper. That said, I would like to know, how much the results of the numerical experiments in terms of the neat convergence rate depend on a specific initialisation of the methods and whether these result would also occur for larger ranks and problem sizes.

**Questions:**

1) In Fig 2 we see that for larger $alpha$ the convergence rate is faster. What is the limit of how large $\alpha$ can be?
2) Do the numerical results hold also for larger ranks of the true matrix and over-parameterized ranks? Also larger imbalance of ranks, lets say k = 20 and r = 5?

---

> ### Author Response · Authors · 2023-11-19
> **Response to the Reviewer Z4Yg**
>
> Thanks for the positive review! We have added more experiments according to your suggestions. Please find our response to your questions below.
> > 1. In Fig 2 we see that for larger $\alpha$ the convergence rate is faster. What is the limit of how large $\alpha$ can be?
>
> In practice, a large initialization may not be able to converge at the initialization phase. We run an experiment to show that a large $\alpha$ fails to converge to a local point. The experiment result is in Appendix I.3
>
> On the theoretical side, the initialization scale $\alpha$ has a constraint in Equation (4.7). A large $\alpha$ does not satisfy the constraint.
>
>
> > 2. Do the numerical results hold also for larger ranks of the true matrix and over-parameterized ranks? Also larger imbalance of ranks, lets say k = 20 and r = 5?
>
> Since we need a relatively small $k$ in Theorem 4.2 $(k<n/8)$, we only run an experiment using $k=10$ and $r=5$. The number of samples is $m=2000$. The experiment results are in Appendix I.4, which shows that a similar phenomenon also holds for a larger rank of the true matrix and a larger over-parameterized rank.

---

> > ### Comment · Reviewer_Z4Yg · 2023-11-21
> > **Response to the rebuttal**
> >
> > I want to thank the authors for addressing all my questions and remarks. My positive score remains unchanged.

---

### Official Review · Reviewer_MU8J · 2023-10-31

**Soundness:** 3 good
**Presentation:** 3 good
**Contribution:** 3 good
**Rating:** 8
**Confidence:** 4

**Summary:**

This paper provides several new results for over-parameterized matrix sensing. First, the authors rigorously prove that with a symmetric parameterization, over-parameterization slows down GD.  In particular, they give a lower bound rate of $\Omega(1/T^2)$. Second, the authors also show that with an asymmetric parameterization, GD converges at an linear rate depending on the initialization scale. This is in contrast with GD with symmetric parameterization, which has a sublinear rate. Finally, the authors extend their algorithm so that the linear convergence rate is independent of the initialization scale.

**Strengths:**

Overall I think this is a good paper. The fact that over-parameterization slows down GD for matrix sensing has been observed by quite a few previous papers. However, this is the first paper that I'm aware of to rigorously establish a lower bound. The authors also show that with asymmetric parameterization, GD converges at an exponential rate that depends on the initialization scale. This is somewhat surprising, given that the asymmetric case has traditionally been considered harder due to potential imbalance of the factors.

**Weaknesses:**

My main concern is with the experiments in this paper. I think the paper could benefit from a more thorough experimental section, perhaps in the appendix.

In the symmetric case, if we use GD with small initialization, then it is often the case that GD goes through an initialization phase where the loss is relatively flat, and then converges rapidly to a small error. However, in the experiments in Figure 2, I do not see this initialization phase in Figure 2b. Instead, linear convergence is observed right from the start, even when a small initialization is used. I wonder why is this the case? For the asymmetric case, is the initialization phase much faster?

Additional experiments which i think should be nice: on the same plot, compare the convergence of asymmetric versus symmetric parameterization, using the same initialization. Also perform the experiment for different initialization scales. I think the authors should also plot convergence for ill-conditioned versus well-conditioned matrices, as GD with small initialization performs differently based on the eigenvalues.

In any case, i would like to see a more detailed comparison of symmetric versus asymmetric parameterization, even just using synthetic experiments.

**Questions:**

In Theorem 1.3, the convergence rate depends on the initialization scale $\alpha$. This is also observed empirically in figure 2b. In practice, does this mean that small initialization has no advantage? One could just set $\alpha$ to be large to ensure rapid convergence?

---

> ### Author Response · Authors · 2023-11-19
> **Response to the Reviewer MU8J**
>
> Thank you for your review and for thinking our paper is good. We have updated our paper, including more experiments according to your suggestions. Please also find our response to your comments below.
>
> > 1. In the symmetric case, if we use GD with small initialization, then it is often the case that GD goes through an initialization phase where the loss is relatively flat, and then converges rapidly to a small error. However, in the experiments in Figure 2, I do not see this initialization phase in Figure 2b. Instead, linear convergence is observed right from the start, even when a small initialization is used. I wonder why is this the case? For the asymmetric case, is the initialization phase much faster?
>
> In Figure 2b, our purpose is to show the exact linear convergence. Hence, we run the algorithm with a large number of iterations, 70000 iterations. Thus,  the initialization phase is not clear in Figure 2b. After zooming into the first 5000 iterations of Figure 2b, we find the existence of the initialization phase. That is, the loss is rather flat during this phase. We can also see that the initialization phase is longer when $\alpha$ is smaller. The experiment results are shown in Appendix I.5.
>
>
> > 2. Additional experiments that I think should be nice: on the same plot, compare the convergence of asymmetric versus symmetric parameterization, using the same initialization. Also, perform the experiment for different initialization scales.
>
> We add the detailed comparison in Appendix I.1. The comparison shows that the asymmetric parameterization performs better than the symmetric parameterization.
>
> > 3. I think the authors should also plot convergence for ill-conditioned versus well-conditioned matrices, as GD with small initialization performs differently based on the eigenvalues.
>
> We plot the curve when $\kappa = 1.5, 3, 10$ with minimal eigenvalue $0.66, 0.33, 0.1$. The experiment results are in Appendix I.2.
> From the experiment results, we can see two phenomena:
> * When the minimum eigenvalue is smaller, the gradient descent will converge to a smaller error at a linear rate. We call this phase as the local convergence phase.
> * After the local convergence phase, the curve first remains flat and then starts to converge at a linear rate again. We can see that the curve remains flat for a longer time when the matrix is ill-conditioned, i.e. $\kappa$ is larger.
>
> This phenomenon has been theoretically identified by the previous work for incremental learning [1,2], in which GD is shown to sequentially recover singular components of the ground truth from the largest singular value to the smallest singular value.
>
>
>
> > 4. In Theorem 1.3, the convergence rate depends on the initialization scale . This is also observed empirically in figure 2b. In practice, does this mean that small initialization has no advantage? One could just set to be large to ensure rapid convergence?
>
> In practice, a large initialization may not be able to converge at the initialization phase. We run an experiment to show that a large $\alpha$ fails to converge to a local point. The experiment result is in Appendix I.3
>
> On the theoretical side, the initialization scale $\alpha$ has a constraint in Equation (4.7). A large $\alpha$ does not satisfy the constraint.
>
> *References:*
>
> [1] Jin et al. Understanding incremental learning of gradient descent: A fine-grained analysis of matrix sensing. ICML 2023.
>
> [2] Jiang et al. Algorithmic Regularization in Model-free Overparametrized Asymmetric Matrix Factorization. SIAM 2023.

---

> > ### Author Response · Authors · 2023-11-21
> > **Follow-Up**
> >
> > Dear Reviewer,
> >
> > Thank you for your time and efforts in reviewing our work. We have added additional experimental results following your suggestions.
> >
> > If you have any additional questions or comments, we would be happy to have further discussions.
> >
> > Thanks,
> >
> > The authors

---

> > > ### Comment · Reviewer_MU8J · 2023-11-21
> > >
> > > I thank the authors for the additional experiments, which resolves my main concern.  I have increased my score.

---

### Author Response · Authors · 2023-11-19
**General Response and A Summary of Changes**

Thanks for the positive and detailed response of all reviewers. We have updated our paper accordingly. All the changes are highlighted in blue. Below is a summary of the changes we made:

* We fixed the typos in Equation (2.2) and Section 3.1, as well as other instances highlighted by the reviewers. We also improved the results in Theorem 3.1 asked by Reviewer UZwv.

* We added additional experiments in Appendix I. To be more specific, we added more comparisons of the asymmetric case and symmetric case in Appendix I.1. We add the comparison of the well-conditioned case and ill-conditioned case in Appendix I.2. We also investigated how large $\alpha$ can be in practice in Appendix I.3. The results for larger true rank $r$ and over-parameterized rank $k$ are in Appendix I.4. The curve of the initialization phase is in Appendix I.5.

* We added a discussion of the requirement of the over-parameterized rank $k$ and the discussion of the small error below Theorem 3.1. We also add the conclusion of the sample complexity $m = \widetilde{O}(nk^2r)$ in Sections 1 and 2.

* We modify the statement of Lemma G.1 to make it clearer.

---

### Meta-Review · Area_Chair_KKTh · 2023-12-06

**Metareview:**

The paper explores the matrix sensing problem, and in particular provides a theoretical analysis of the behavior of gradient descent on various settings of the problem.

Strengths:
 - Reviewers all agree that the results are novel and interesting advances in the field of matrix sensing.

Weaknesses:
 -  Reviewer MU8j's initial concern regarding the experiments was satisfactorily answered in rebuttal.

**Justification For Why Not Higher Score:**

The paper received high numerical scores from the reviewers, and based purely on score, looks like a clear case for ICLR oral.  However, the high score emerges because the reviewers have been carefully chosen to be experts in this area.  However the area itself is a small subfield of machine learning (below than 1% of submissions [*]), and the bar for an oral paper at ICLR is that a large proportion of attendees should be exposed to the paper.

[*] Total hits for "matrix [factorization|completion|sensing]" on all ICLR content covering several years is 115; ICLR 2022-2024 submissions number 15000. https://openreview.net/search?term=%22matrix%20factorization%22&group=ICLR.cc&content=all&source=forum

**Justification For Why Not Lower Score:**

This is an excellent paper within the field of matrix sensing, advances that field, and therefore may be of interest to ICLR attendees in adjacent subfields who might not seek it out as a poster.

---

### Decision · Program_Chairs · 2024-01-16

Accept (spotlight)